# Prediction of Compressive Strength Loss of Normal Concrete after Exposure to High Temperature

Xiaoyu Qin [ID], Qianmin Ma *, Rongxin Guo [ID] and Shaoen Tan

Yunnan Key Laboratory of Disaster Reduction in Civil Engineering, Faculty of Civil Engineering and Mechanics, Kunming University of Science and Technology, Kunming 650500, China
* Correspondence: maqianmin666@163.com

**Abstract:** In recent years, there has been an increasing number of fires in buildings. The methods for detecting residual properties of buildings after fires are commonly destructive and subjective. In this context, property prediction based on mathematical modeling has exhibited its potential. Backpropagation (BP), particle swarm algorithms optimized-BP (PSO-BP) and random forest (RF) models were established in this paper using 1803 sets of data from the literature. Material and relevant heating parameters, as well as compressive strength loss percentage, were used as input and output parameters, respectively. Experimental work was also carried out to evaluate the feasibility of the models for prediction. The accuracy of all the models was sufficiently high, and they were also much more feasible for prediction. Moreover, based on the RF model, the importance of the inputting parameters was ranked as well. Such prediction has provided a new perspective to non-destructively and objectively assess the post-fire properties of concrete. Additionally, this model could be used to guide performance-based design for fire-resistant concrete.

**Keywords:** concrete; compressive strength loss after high temperature; artificial neural network; random forest; prediction model

## 1. Introduction

In recent years, there has been an increasing number of fires in buildings. It is reported that during 2003–2012 there was an annual average of 180,000 fires in China and 1,403,000 in the United States, respectively [1], while in the first three quarters of 2022, 223,800 residential fires were already reported in China [2]. When a fire occurs, temperature increases fast in a short time (see Figure 1). As a result of the decomposition of hydration products, thermal cracking, loosening of matrix structure and expansion and/or decomposition of aggregates, the properties of structural concrete usually deteriorate with the increase in temperature [3]. Currently, the following methods are mostly used to detect the quality of a building after fire [4]: (1) concrete is hammered by experienced personnel to determine its degree of damage according to the sound made; (2) the rebound method is used to determine the residual strength of the concrete, particularly in the near-surface zone, i.e., roughly 30 mm; (3) concrete is cored, and the residual strength of the core sample is determined by its depth of ablation. Although these methods are widely used, they also have shortcomings such as being highly subjective, the number of samples being limited and possible secondary damage being caused. In this context, based on the data in the literature published, this paper proposes statistical models to predict compressive strength loss of normal concrete after exposure to high temperatures, attempting to implement such examination effectively and non-destructively. Parallel experimental work is also to be carried out to evaluate the feasibility of the prediction. In addition, weighting of the influencing factors is to be analyzed, which could be useful to better understand the thermal behavior of the concrete and to provide possible guidance for thermal performance-based concrete design.

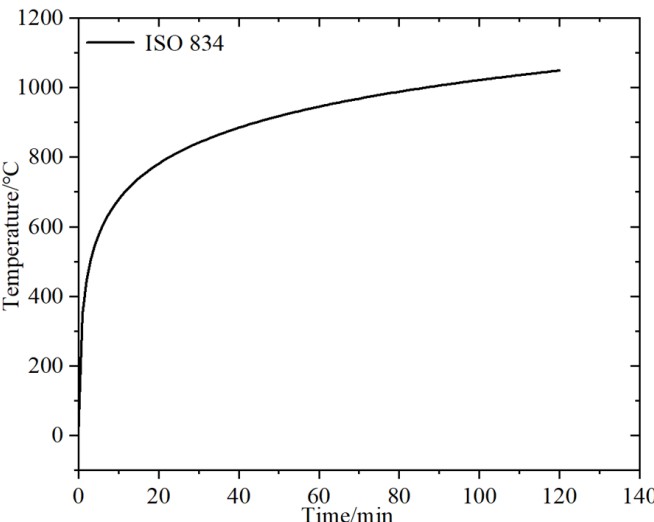

**Figure 1.** ISO 834 temperature–time curve.

## 2. Modeling

Backpropagation (BP) neural network, particle swarm optimization BP (PSO-BP) neural network and random forest (RF) modeling methods are conducted in this paper to implement the prediction work. In addition, an importance ranking of influencing parameters is also carried out based on the RF model.

### 2.1. Parameter Determination

Compressive strength loss of normal concrete after high temperatures is mainly influenced by material parameters, such as water binder ratio (W/B) and high-temperature operating mechanisms, such as heating temperature (T), heating velocity (V), maintaining duration at target temperature (MD), cooling method (C) and resting duration after cooling (RD). Therefore, in this paper, the W/B, T, V, MD, C and RD were used as input parameters to establish the models. The cooling method (water cooling/natural cooling) was converted into a digital signal (0/1). To eliminate the influence of initial strength at room temperature (i.e., before exposure to high temperature), strength loss percentage after high temperatures (P) was selected as the output parameter in this paper. The P is determined according to Equation (1).

$$P = \frac{f_{cu,\ high\ temperature} - f_{cu,\ room\ temperature}}{f_{cu,\ room\ temperature}} \times 100\%, \tag{1}$$

where $f_{cu,\ high\ temperature}$ is the compressive strength of concrete after high temperature and $f_{cu,\ room\ temperature}$ is the compressive strength of concrete at room temperature.

### 2.2. Data Collection

The data, with a total of 1803 sets, for building the models in this paper was obtained from the literature [5,88], as detailed in Appendix A, and its statistics are provided in Table 1. Training used 70% of the data, and the rest of the data was evenly used for validation and testing.

**Table 1.** Data statistics.

| | W/B | T (°C) | V (°C/min) | MD (h) | C | RD (day) | P (%) |
|---|---|---|---|---|---|---|---|
| Range | [0.18–0.77] | [50–1200] | [0.10–500] | [0–48] | 0/1 | [0–112] | [−99.46–66.88] |
| Average | 0.44 | 479.50 | 18.82 | 2.43 | - | 3.55 | −31.22 |
| Standard deviation | 0.12 | 240.29 | 57.48 | 3.09 | - | 14.07 | 29.17 |

*2.3. Models*

2.3.1. BP Neural Network

The BP neural network is the most commonly used artificial neural network, which is a method of processing information imitating a human neural network. The BP neural network continuously approximates the function by adjusting the weights and thresholds, and if the error does not meet the requirements, the signal is fed backwards [89]. It is widely used in the field of civil engineering because of its strong robustness. Its network usually contains a single input layer, single or multi-hidden layer(s) and a single output layer, where many neurons are involved. In this paper, the number of neurons in the input and output layers was 6 and 1, respectively. After trial-and-error and comparison (see Table 2, where RMSE and MAE are root mean square error and mean absolute error, respectively, which are to be defined in Section 2.4), a network structure of 6-7-1 was finally determined, as shown in Figure 2. The Levenberg–Marquardt algorithm was used for training, where the training frequency, learning rate and minimum error of the training target were set to 1000, 0.01 and 0.00001, respectively. The optimum solution after 50 training sessions was used as the target model. Furthermore, in order to minimize the effect of data on the results, the data were normalized to $[-1, 1]$.

**Table 2.** Comparison between different network structures.

| Error (%) | 6-5-1 | | 6-7-1 | | 6-10-1 | | 6-(5,5)-1 | |
|---|---|---|---|---|---|---|---|---|
| | **Training** | **Test** | **Training** | **Test** | **Training** | **Test** | **Training** | **Test** |
| RMSE | 6.27 | 6.09 | 4.17 | 3.14 | 5.98 | 4.27 | 4.92 | 3.97 |
| MAE | 5.74 | 5.01 | 3.12 | 2.78 | 4.73 | 3.91 | 4.03 | 3.19 |

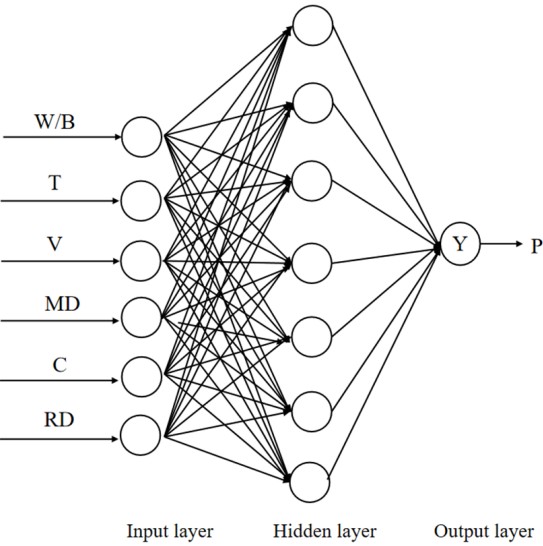

**Figure 2.** BP neural network structure.

2.3.2. PSO-BP Neural Network

Due to the algorithm of the BP neural network easily resulting in the issue of local extreme small values, the particle swarm algorithm was used in this paper to further optimize weights and thresholds (PSO), developing the PSO-BP neural network. The PSO algorithm is inspired by the predatory behavior of bird colonies, treating each individual as a particle in different spaces to build a search model in terms of velocity (V) and position (X). If there are n particles in an A dimensional space, i.e., $X = (X_1, X_2, X_3 \ldots \ldots X_n)$, the position of particle $X_i$ in the A dimensional space, i.e., potential solution, is denoted as $X_i = (X_{i1}, X_{i2}, X_{i3} \ldots \ldots X_{iA})^T$. The particle is then updated by each iteration of the individual and extreme global values, with the velocity and position defined in Equations (2) and (3) [90].

$$V_{iA}{}^{k+1} = \omega V_{iA}{}^{k} + c_1 r_1 (P_{iA} - X_{iA}) + c_2 r_2 (P_{gA} - X_{iA}), \tag{2}$$

$$X_{iA}{}^{k+1} = X_{iA}{}^{k} + V_{iA}{}^{k+1}, \tag{3}$$

where

→ $V_i$: particle velocity, denoted as $(V_{i1}, V_{i2}, V_{i3}......V_{iA})^T$;
→ $P_i$: individual extreme value, denoted as $(P_{i1}, P_{i2}, P_{i3}......P_{iA})^T$;
→ $P_g$: global extreme value, denoted as $(P_{g1}, P_{g2}, P_{g3}......P_{gA})^T$;
→ $X_i$: particle position, denoted as $(X_{i1}, X_{i2}, X_{i3}......X_{iA})^T$;
→ $\omega$: inertia weights;
→ k: number of current iterations;
→ $c_1$ and $c_2$: learning factors which are non-negative constants;
→ $r_1$ and $r_2$: momentum coefficients which are random numbers between [0,1].

In this paper, particle number n was set as 20, learning factors $c_1$ and $c_2$ were kept the same as 2 and momentum coefficients $r_1$ and $r_2$ were kept the same as 0.8 [90]. Inertia weight $\omega$ was assigned to 0.8 after several trials. In addition, in order to avoid blind searching of particles, the position and velocity of particles were limited in ranges of $[-1, 1]$ and $[0, 1]$, respectively [90]. The model iterated and updated 100 times, which ensured that the model was sufficiently convergent and could be highly reliable because, after 60 iterations, the adaptability of the model was sufficiently stable, as shown in Figure 3. The training times, network structure, target minimum error, learning efficiency and normalization range of the PSO-BP modeling were kept the same as BP modeling, as shown in the previous section.

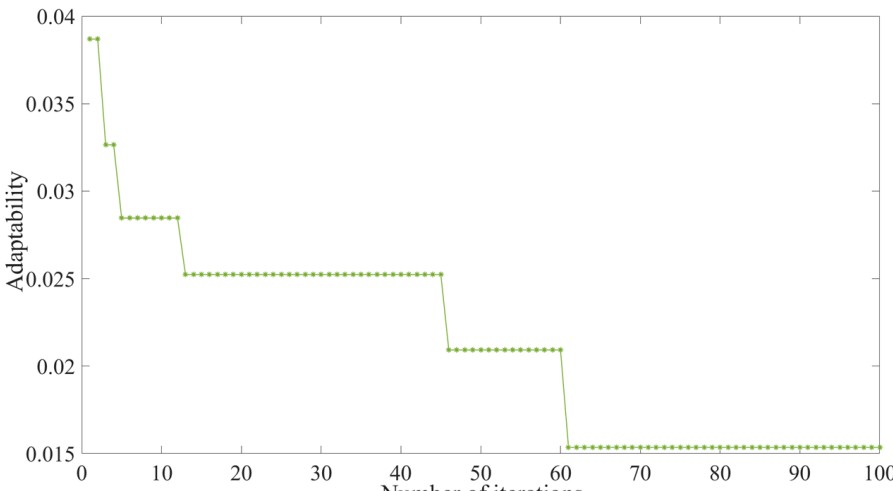

**Figure 3.** Adaptation curve of the PSO-BP model.

### 2.3.3. RF

RF is a method of taking data samples from a database randomly, where bootstrap re-sampling is usually used, and then repeatedly dichotomizing the data to eventually determine the optimum solution by voting [91]. In RF, "forest" refers to an integration of decision trees which consists of nodes and directed edges. The structure of the model is shown in Figure 4. This method can highly tolerate outliers of data, compute fast and yield a prediction with high accuracy. It was found that the determination coefficient $R^2$ tends to be constant when the number of decision trees is up to 500, which was set to be the number of decision trees used in this paper.

In addition, RF was also used in this paper to perform a variable importance measure (VIM), where the contribution of each variable to each tree in the RF is averaged and then ranked. The Gini index (*GI*) was used in this paper to evaluate the VIM [92]. Assuming that there were J variables $X_1, X_2, X_3..., X_J$, I decision trees and C categories of variables

(C = 2 when dichotomization is used to process data), the GI of node q of the ith tree is provided in Equation (4).

$$GI_q^{(i)} = \sum_{C=1}^{C} p_{qc}^{(i)} \left(1 - p_{qc}^{(i)}\right), \tag{4}$$

where $p_{qc}$ is the proportion of category C in node q.

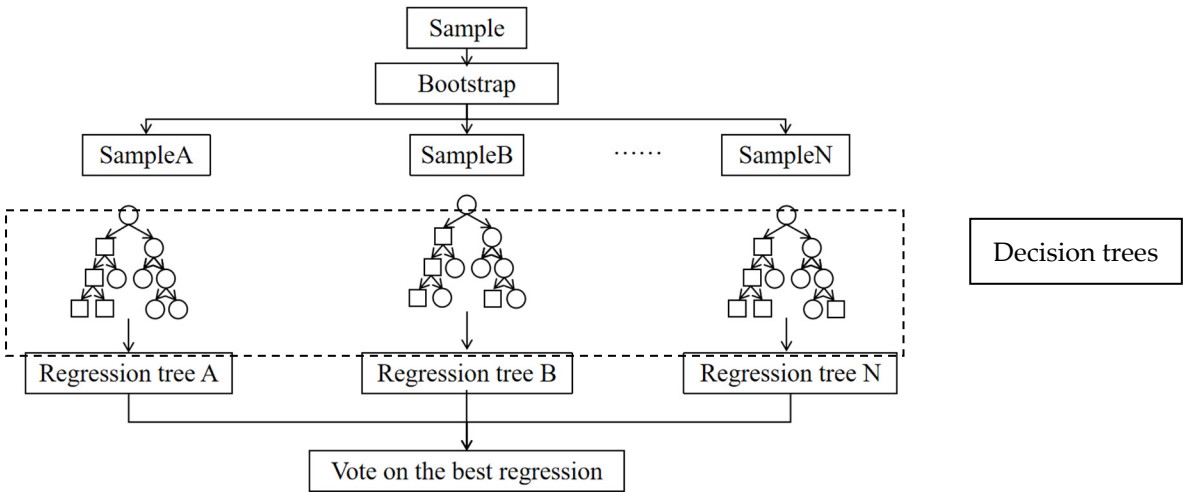

**Figure 4.** RF model structure.

The importance of variable $X_J$ at node q of the ith tree, i.e., the difference in the Gini index before and after the branching of node q, is provided in Equation (5).

$$VIM_{jq}^{(Gini)(i)} = GI_q^{(i)} - GI_l^{(i)} - GI_r^{(i)}, \tag{5}$$

where $GI_l^{(i)}$ and $GI_r^{(i)}$ are the Gini index of the two new nodes after branching, respectively.

If variable $X_J$ appears C times, the importance of $X_J$ in the ith tree is provided in Equation (6).

$$VIM_{ij}^{(Gini)(i)} = \sum_{q=1}^{C} VIM_{jq}^{(Gini)(i)}. \tag{6}$$

Finally, the importance score of RF is provided in Equation (7).

$$VIM_j^{(Gini)} = \frac{1}{I} \sum_{i=1}^{I} VIM_{ij}^{(Gini)(i)}. \tag{7}$$

*2.4. Error Evaluation*

Root mean square error (RMSE) and mean absolute error (MAE) evaluations were applied in this paper to evaluate the accuracy of the models mentioned previously. Formulae of the error evaluations are provided in Equations (8) and (9), where $y_i$ ($y_1, y_2 \ldots \ldots y_n$) is the value measured value, $\widehat{y}_i$ ($\widehat{y}_1, \widehat{y}_2 \ldots \ldots \widehat{y}_n$) is the value predicted and n is the number of data samples. The closer the RMSE and MAE values to zero, the smaller the error between the data samples and the more accurate the model.

$$RMSE = \left( \sum_{i=1}^{n} \left(y_i - \widehat{y}_i\right)^2 / n \right)^{1/2}, \tag{8}$$

$$MAE = \sum_{i=1}^{n} \left| y_i - \widehat{y}_i \right| / n. \tag{9}$$

## 3. Experimental Program

Experimental work was also carried out to validate the prediction based on the mathematic models. To manufacture the concrete specimens, 42.5-grade ordinary Portland cement produced by the Yunnan Kunming Huaxin Cement Factory was used. Crushed stone with 5–25 mm continuous grade was used as coarse aggregate. Machined sand with a fineness of 2.82 was used as fine aggregate, and a sand ratio of 40% was applied. Water–binder ratios (W/B) of 0.3, 0.4 and 0.5 were used for comparison. Mix proportions of concrete specimens are provided in Table 3.

**Table 3.** Mix proportions of concrete specimens (kg/m$^3$).

| No. | W/B | Water | Cement | Coarse Aggregates | Fine Aggregates |
|---|---|---|---|---|---|
| 1 | 0.3 | 380 | 114 | 1113 | 743 |
| 2,4–7 | 0.4 | 380 | 152 | 1091 | 727 |
| 3 | 0.5 | 380 | 190 | 1068 | 712 |

Concrete specimens with a size of 100 mm × 100 mm × 100 mm were manufactured in accordance with the Chinese national standard GB/T 50081-2019. The concrete mixture was cast into the mold in two layers. After each layer-casting, the mixture was vibrated for 10–20 s on a vibration table to eliminate any possible voids. The hardened concrete specimens were de-molded after standing for 1 day in an ambient environment and then put in a curing room with a temperature of 20 ± 1 °C and relative humidity of 100%. After curing for another 27 days, the specimens were extracted and placed in an electrical muffle furnace for heating with the operation mechanism provided in Table 4 (illustrated graphically in Figure 5). Afterwards, the specimens were crushed using a WE-300 hydraulic universal testing machine to test compressive strength. Three duplicated specimens were produced for each mix at each heating temperature, and the strength reported is an average of the three results. The strength loss percentage was calculated using Equation (1).

**Table 4.** High temperature operation mechanism.

| No. | T (°C) | V (°C/min) | MD (Hour) | C | RD (Day) |
|---|---|---|---|---|---|
| 1 | | 10 | 2 | Nature | 0 |
| 2 | | 10 | 2 | Nature | 0 |
| 3 | 200, 400, 600, 800 | 10 | 2 | Nature | 0 |
| 4 | | 5 | 2 | Nature | 0 |
| 5 | | 10 | 1 | Nature | 0 |
| 6 | | 10 | 2 | Water | 0 |
| 7 | | 10 | 2 | Nature | 1 |

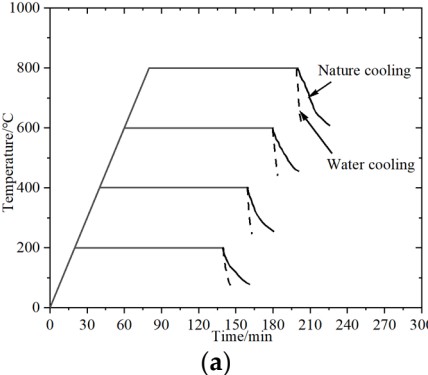

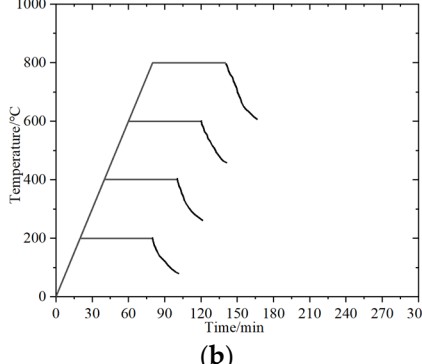

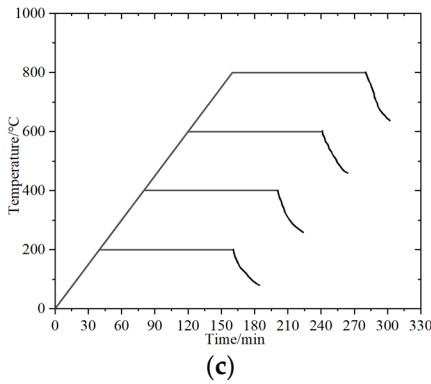

**Figure 5.** High temperature operating mechanisms. (**a**) 10 °C/min-2 h. (**b**) 10 °C/min-1 h. (**c**) 5 °C/min-2 h.



## 4. Results and Discussion

### 4.1. Modeling

Regressions of the BP, PSO-BP and RF models are shown in Figures 6–8, respectively. In the figures, the X (Target) and Y (Output) axis represent the data (P) reported in the literature and those obtained via modeling, respectively. The dotted line refers to the output value equal to the target value, and the correlation coefficient R equals 1. The real line is the regression of the real relation between output value and target value. The closer the real line is to the dotted line, the higher the R-value. From Figure 6, it can be seen that there is a good correlation between the data samples no matter whether training, testing, validation or the whole stage of the BP modeling is considered, as the R-value is greater than 0.86. PSO processing improved the correlation further as the minimum R-value increased to 0.87 (see Figure 7). The correlation was significantly improved when RF modeling was applied, as all the R values were higher than 0.92 (see Figure 8). Error evaluation of the models is provided in Table 5. Both RMSE and MAE values of the models were at a very low level. This was particularly true when the PSO-BP and the RF models were considered. Both correlation and error evaluation indicated that the accuracy of the models was sufficiently high. Compared to the BP model, the PSO-BP and the RF models were more accurate.

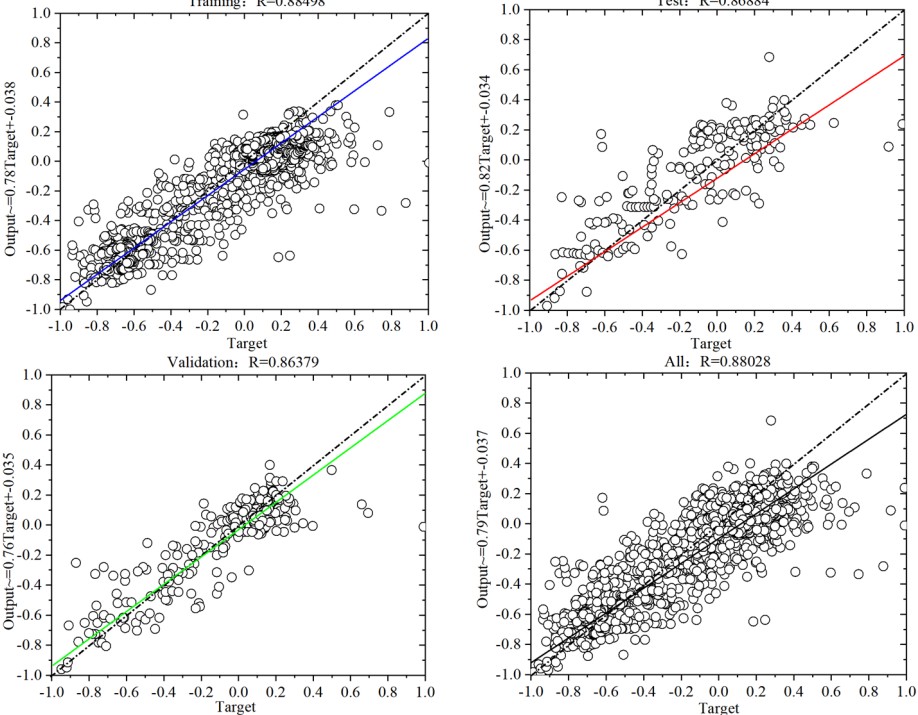

**Figure 6.** Correlation of BP modeling.

### 4.2. Experimental Validation

Later, experimental work was carried out to validate the feasibility of the modeling prediction. The compressive strength of the seven concrete mixes was experimentally tested to calculate the strength loss percentage; the results are reported in Table 6 and Figure 9. The appearance of the concrete specimens before and after high-temperature exposures is shown in Figure 10. From the results, it can be seen that the compressive strength of all seven mixes reduced with the increase in temperature. After 800 °C, only 24–36% of the strength remained. With the increase of W/B from 0.3 to 0.5 (mixes 1, 2 and 3), concrete specimens were to be less dense [3], resulting in a lower strength at room temperature and a higher strength loss after high-temperature exposures. When a lower heating velocity was applied (comparison between mixes two and four), the heating duration was prolonged to achieve the target temperature causing higher strength loss [93].

A shorter maintenance duration at the high temperatures (comparison between mixes two and five) could protect the specimens from worse thermal damage [94]; therefore, lower strength loss was observed. It is usually considered that water cooling will cause thermal stress distributed in concrete to lower its strength [45]. It was found that the shrinkage of the cement matrix with temperature could compensate for such stress [95]; therefore, in this paper, it was observed that water cooling had a less significant influence on the strength loss (comparison between mixes two and six). After cooling, resting for a certain duration before crushing allowed parts of the products of concrete, which were decomposed thermally, to rehydrate to make the concrete dense [96]. Consequently, it was found that the specimens with a one-day-resting duration showed less strength loss.

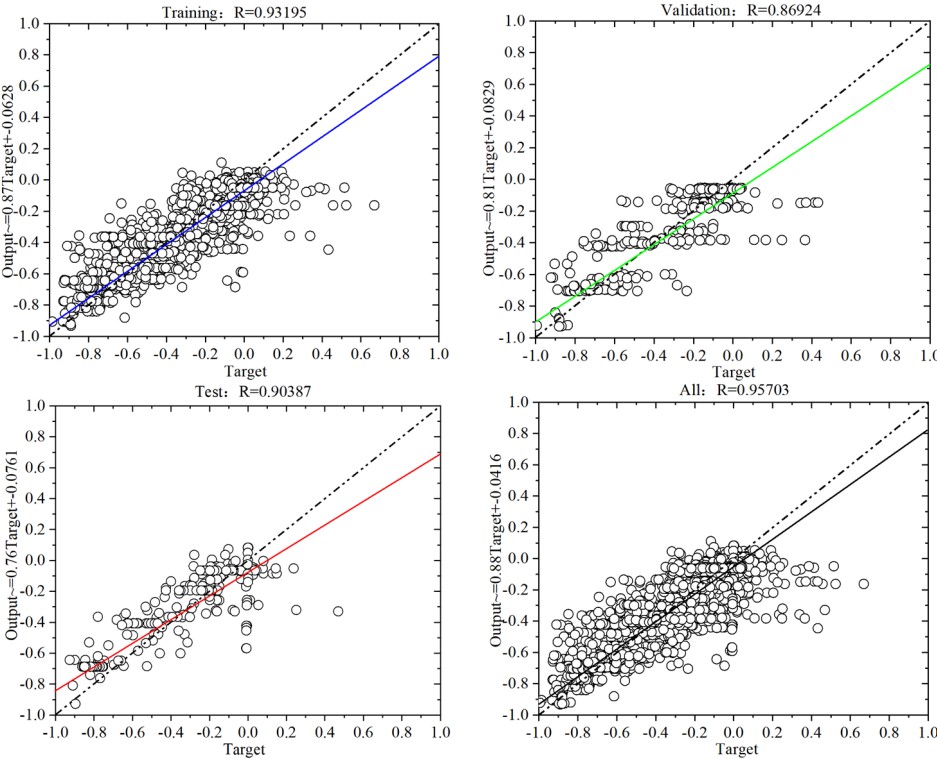

**Figure 7.** Correlation of PSO-BP modeling.

Simultaneously, the material and heating-related parameters used in the experimental work were put in the models to run a prediction, and the results predicted are reported in Table 6 as well. The error between the results measured and predicted (100% × | predicted value-measured value | /measured value) is illustrated in Figure 11, and the statistic of the errors is summarized in Table 7. From the results, it can be seen that although in all cases the maximum error values were relatively high, i.e., more than 30%, the mean values of less than 10% were still at a low level, which could meet the requirement of engineering practice. Furthermore, concrete mix one always yielded the largest error value in all three predictions. A possible reason for that could be due to a relatively low W/B of 0.3 being used in mix one, and the temperature was not high enough, i.e., 200 °C; therefore, the data under such circumstances is not adequate in the literature to run sufficient training during modeling. Nevertheless, compared to the BP model, the PSO-BP and the RF models indeed improved the feasibility of the prediction as both the error range and the average error were reduced dramatically.

Due to the sufficient feasibility of the prediction, it is suggested that in practice, engineers could input relevant parameters into the models to yield a residual strength instantly, which could also avoid secondary damage to the post-fire building caused by destructive testing, and the results would be objective and more reliable.

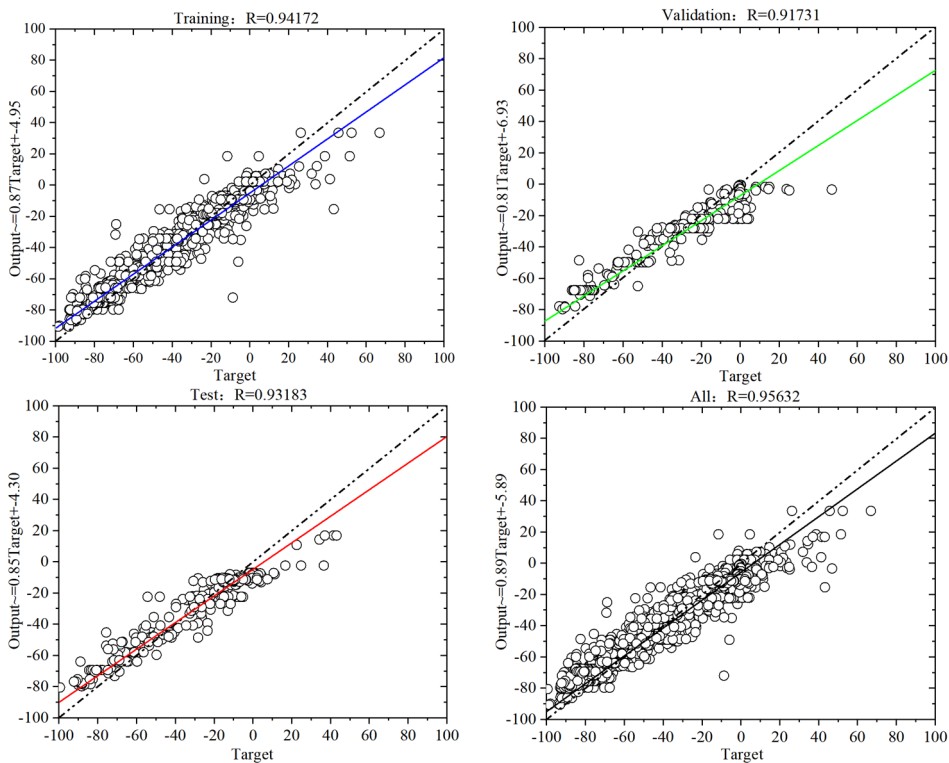

**Figure 8.** Correlation of RF modeling.

**Table 5.** Error evaluation.

| | BP | | PSO-BP | | RF | |
|---|---|---|---|---|---|---|
| | **Training** | **Test** | **Training** | **Test** | **Training** | **Test** |
| RMSE (%) | 4.17 | 3.14 | 2.60 | 2.12 | 2.56 | 2.23 |
| MAE (%) | 3.12 | 2.78 | 3.17 | 2.63 | 2.57 | 1.79 |

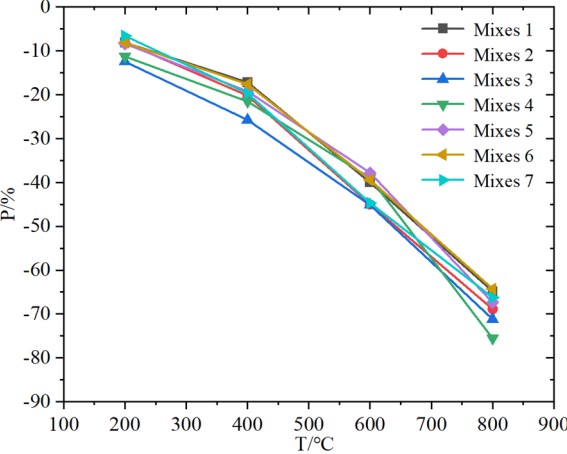

**Figure 9.** Relationship between temperature and loss of compressive strength of concrete after high temperature.

### 4.3. VIM

In this paper, a VIM was implemented using the RF model to rank the importance of the inputting parameters on compressive strength loss of normal concrete after high temperature, and the result is illustrated in Figure 12. The importance ranking is in the sequence of T > W/B > V > MD > RD > C. Heating temperature had a crucial influence,

accounting for 67.3%. Influences contributed by W/B and heating velocity were quite similar, accounting for 11.6% and 10.3%, respectively, followed by the duration at target temperature and duration after cooling, accounting for 5.6% and 4.2%, respectively; the influence of the cooling method was insignificant, accounting for only 1.0%.

**Table 6.** Compressive strength loss percentages measured and predicted.

| No. | T (°C) | Compressive Strength (MPa) | P (Measured, %) | P (Predicted by BP, %) | P (Predicted by PSO-BP, %) | P (Predicted by RF, %) |
|---|---|---|---|---|---|---|
| 1 | 20 | 75.61 | – | – | – | – |
| | 200 | 69.45 | −8.14 | −4.61 | −5.57 | −5.13 |
| | 400 | 62.59 | −17.22 | −14.37 | −15.83 | −16.98 |
| | 600 | 45.36 | −40.00 | −37.42 | −41.23 | −39.61 |
| | 800 | 26.61 | −64.81 | −64.14 | −65.03 | −62.03 |
| 2 | 20 | 48.16 | – | – | – | – |
| | 200 | 44.24 | −8.14 | −8.27 | −7.99 | −9.01 |
| | 400 | 38.42 | −20.22 | −16.68 | −21.07 | −20.66 |
| | 600 | 26.48 | −45.02 | −37.69 | −47.51 | −47.22 |
| | 800 | 14.99 | −68.88 | −65.36 | −69.32 | −70.23 |
| 3 | 20 | 40.42 | – | – | – | – |
| | 200 | 35.38 | −12.47 | −13.15 | −11.73 | −12.40 |
| | 400 | 30.01 | −25.76 | −27.39 | −25.82 | −24.92 |
| | 600 | 22.17 | −45.15 | −47.86 | −46.22 | −42.79 |
| | 800 | 11.64 | −71.19 | −70.37 | −69.61 | −68.11 |
| 4 | 20 | 48.16 | – | – | – | – |
| | 200 | 42.72 | −11.31 | −10.96 | −12.62 | −11.80 |
| | 400 | 37.78 | −21.56 | −22.92 | −22.56 | −23.19 |
| | 600 | 29.37 | −39.01 | −43.83 | −38.91 | −38.93 |
| | 800 | 11.76 | −75.57 | −77.17 | −77.14 | −74.32 |
| 5 | 20 | 48.16 | – | – | – | – |
| | 200 | 44.11 | −8.41 | −7.59 | −10.23 | −6.99 |
| | 400 | 38.89 | −19.25 | −16.00 | −23.23 | −20.20 |
| | 600 | 29.94 | −37.83 | −36.47 | −39.67 | −38.90 |
| | 800 | 15.70 | −67.39 | −64.00 | −68.22 | −66.39 |
| 6 | 20 | 48.16 | – | – | – | – |
| | 200 | 44.24 | −8.14 | −8.14 | −8.19 | −9.13 |
| | 400 | 39.67 | −17.63 | −16.10 | −18.11 | −15.19 |
| | 600 | 29.15 | −39.46 | −34.44 | −40.34 | −38.92 |
| | 800 | 17.21 | −64.27 | −61.42 | −63.28 | −64.02 |
| 7 | 20 | 48.16 | – | – | – | – |
| | 200 | 44.96 | −6.63 | −8.21 | −7.91 | −7.24 |
| | 400 | 38.67 | −19.70 | −21.03 | −21.19 | −18.93 |
| | 600 | 26.63 | −44.69 | −46.11 | −47.91 | −44.93 |
| | 800 | 15.29 | −66.24 | −68.31 | −69.01 | −67.29 |

It is clear that, for normal concrete, the decomposition of hydration products, cracking, loosening of matrix structure and expansion and/or decomposition of aggregates caused the loss of compressive strength with the increase in temperature [3]. Consequently, the temperature should be the most important factor influencing the compressive strength loss of concrete. As discussed in Section 4.2, it is usually considered that water cooling will result in thermal stress in concrete to dramatically reduce its strength. At the same time, it is also found that shrinkage of cement matrix with temperature would compensate for such stress to lower the influence on the strength loss. Therefore, the importance of the cooling method is less significant. Furthermore, a smaller amount of literature has discussed the influence of the cooling method on the compressive strength of normal concrete; therefore, a smaller data size would also have an influence on the ranking of the cooling method.

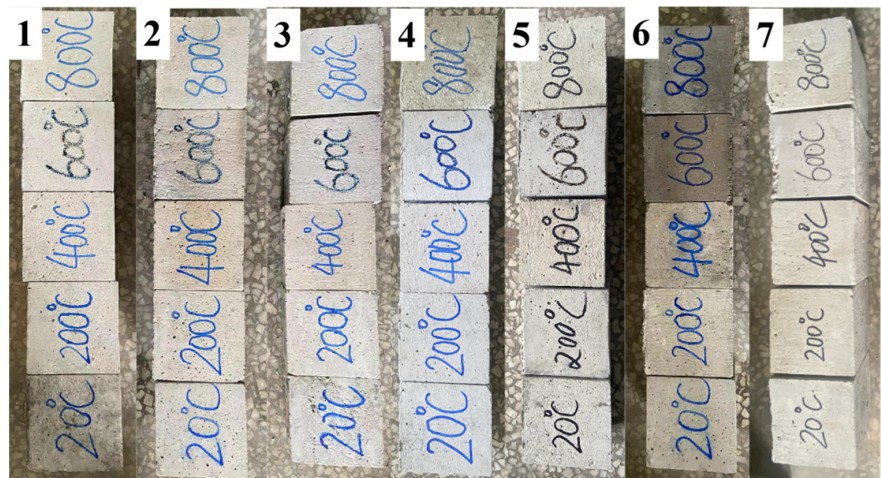

**Figure 10.** Appearance of concrete specimens before and after high-temperature exposures.

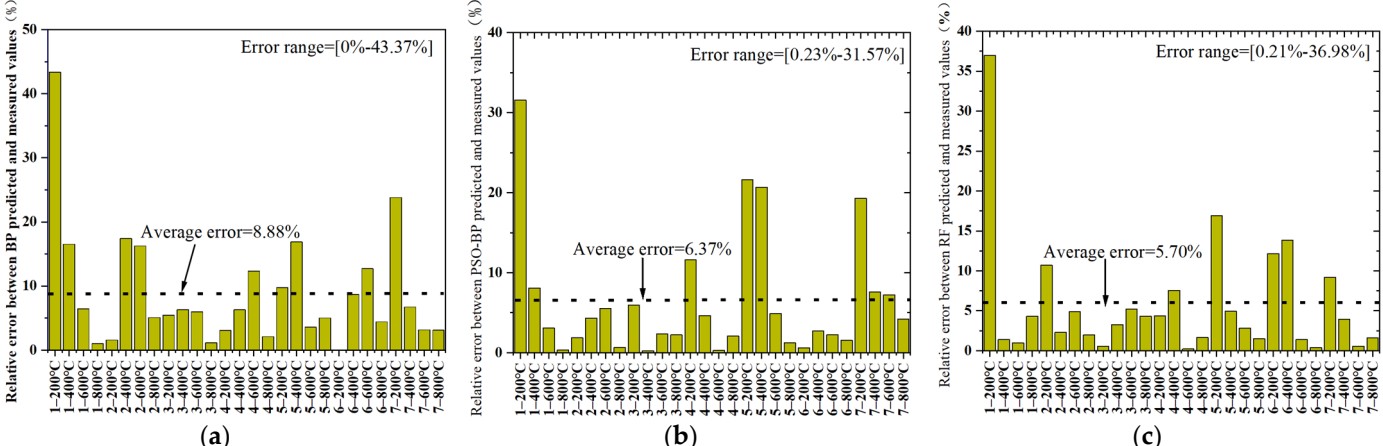

**Figure 11.** Error between the values predicted and measured. (**a**) BP model. (**b**) PSO-BP model. (**c**) RF model.

**Table 7.** Statistic of the error between the results measured and predicted.

|  | BP | PSO-BP | RF |
|---|---|---|---|
| Range (%) | [0.00–43.37] | [0.23–31.57] | [0.20–36.98] |
| Average (%) | 8.88 | 6.37 | 5.70 |

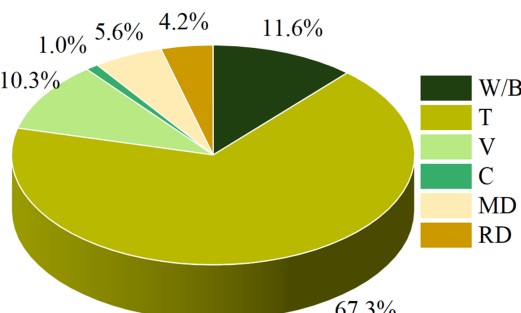

**Figure 12.** Importance of input parameters based on the RF model.

## 5. Conclusions

Fires in buildings cause serious damage to structural concrete. Post-fire assessments currently used are usually subjective and destructive. Therefore, BP, PSO-BP and RF models

were established in this paper to predict the compressive strength loss of normal concrete after high temperatures.

(1) To establish the models, 1803 sets of data from the publicly published literature were used, W/B, T, V, MD, C and RD were determined as input parameters and P was applied as an output parameter. Based on RMSE and MAE error evaluation, the accuracy of all three models was sufficiently high. Compared to the BP model, both the PSO-BP and the RF models were more accurate.

(2) Parallel experimental work was carried out with modeling prediction using the same parameters. An error value between the results measured and predicted of less than 10% proved that all three models had sufficient feasibility to complement the prediction. Compared to the other two models, RF one was much more feasible.

(3) Based on the RF model, the importance of the input parameters was ranked in the sequence of T > W/B > V > MD > RD > C.

(4) With the continuous expansion of data size, the accuracy of the models could be improved further. Such prediction work has provided a new perspective to assess the post-fire properties of concrete non-destructively and objectively. Additionally, it could be used to guide performance-based design for fire-resistant concrete.

**Author Contributions:** Conceptualization, X.Q. and Q.M.; methodology, Q.M.; software, X.Q.; validation, R.G. and S.T.; formal analysis, X.Q.; investigation, Q.M.; resources, R.G.; data collection, S.T.; writing original draft preparation, X.Q.; writing—review and editing, Q.M.; visualization, S.T.; supervision, R.G.; project administration, Q.M.; funding acquisition, Q.M. All authors have read and agreed to the published version of the manuscript.

**Funding:** This research was funded by the National Natural Science Foundation of China grant number [52068038] and the Yunnan Provincial Department of Science and Technology grant number [202101AT070089].

**Institutional Review Board Statement:** Not applicable.

**Informed Consent Statement:** Not applicable.

**Data Availability Statement:** All the data used to support the finding of this study have been included in the article.

**Conflicts of Interest:** No conflicts of interest.

**Appendix A**

**Table A1.** Data for modeling.

| REF. | Input Parameters | | | | | | Output Parameters |
|---|---|---|---|---|---|---|---|
| | W/B | T (°C) | V (°C/min) | MD (h) | C | RD (day) | P (%) |
| | 0.49 | 200 | 10 | 2 | 1 | 14 | −4.28 |
| | 0.49 | 300 | 10 | 2 | 1 | 14 | 8.06 |
| | 0.49 | 400 | 10 | 2 | 1 | 14 | 11.83 |
| | 0.49 | 500 | 10 | 2 | 1 | 14 | 2.00 |
| | 0.49 | 600 | 10 | 2 | 1 | 14 | −23.50 |
| | 0.49 | 700 | 10 | 2 | 1 | 14 | −50.06 |
| | 0.49 | 800 | 10 | 2 | 1 | 14 | −71.17 |
| [5] | 0.49 | 200 | 10 | 2 | 0 | 14 | −1.33 |
| | 0.49 | 300 | 10 | 2 | 0 | 14 | 0.94 |
| | 0.49 | 400 | 10 | 2 | 0 | 14 | −1.75 |
| | 0.49 | 500 | 10 | 2 | 0 | 14 | 1.81 |
| | 0.49 | 600 | 10 | 2 | 0 | 14 | −17.64 |
| | 0.49 | 700 | 10 | 2 | 0 | 14 | −42.75 |
| | 0.49 | 800 | 10 | 2 | 0 | 14 | −63.03 |

**Table A1.** *Cont.*

| REF. | Input Parameters | | | | | | Output Parameters |
|---|---|---|---|---|---|---|---|
| | W/B | T (°C) | V (°C/min) | MD (h) | C | RD (day) | P (%) |
| [6] | 0.18 | 100 | 2 | 2 | 1 | 3 | −1.05 |
| | 0.18 | 200 | 2 | 2 | 1 | 3 | 14.65 |
| | 0.18 | 300 | 2 | 2 | 1 | 3 | 25.42 |
| | 0.18 | 400 | 2 | 2 | 1 | 3 | 34.56 |
| | 0.18 | 500 | 2 | 2 | 1 | 3 | 8.61 |
| | 0.18 | 600 | 2 | 2 | 1 | 3 | 4.22 |
| | 0.18 | 800 | 2 | 2 | 1 | 3 | −68.85 |
| [7] | 0.52 | 200 | 5 | 6 | 1 | 0 | −1.05 |
| | 0.52 | 400 | 5 | 6 | 1 | 0 | −9.41 |
| | 0.52 | 600 | 5 | 6 | 1 | 0 | −27.74 |
| | 0.52 | 800 | 5 | 6 | 1 | 0 | −43.71 |
| | 0.52 | 200 | 5 | 6 | 0 | 2 | 2.09 |
| | 0.52 | 400 | 5 | 6 | 0 | 2 | −11.00 |
| | 0.52 | 600 | 5 | 6 | 0 | 2 | −33.50 |
| | 0.52 | 800 | 5 | 6 | 0 | 2 | −52.36 |
| [8] | 0.33 | 105 | 10 | 3 | 1 | 1 | −7.06 |
| | 0.33 | 200 | 10 | 3 | 1 | 1 | −12.86 |
| | 0.33 | 400 | 10 | 3 | 1 | 1 | −36.64 |
| | 0.33 | 600 | 10 | 3 | 1 | 1 | −63.91 |
| | 0.33 | 800 | 10 | 3 | 1 | 1 | −85.36 |
| [9] | 0.49 | 105 | 10 | 6 | 1 | 1 | −10.00 |
| | 0.49 | 300 | 10 | 6 | 1 | 1 | −2.80 |
| | 0.49 | 500 | 10 | 6 | 1 | 1 | −13.34 |
| | 0.49 | 700 | 10 | 6 | 1 | 1 | −51.94 |
| | 0.6 | 105 | 10 | 6 | 1 | 0 | −11.67 |
| | 0.6 | 300 | 10 | 6 | 1 | 0 | −7.00 |
| | 0.6 | 500 | 10 | 6 | 1 | 0 | −19.63 |
| | 0.6 | 700 | 10 | 6 | 1 | 0 | −44.23 |
| | 0.49 | 105 | 10 | 6 | 1 | 0 | −10.01 |
| | 0.49 | 300 | 10 | 6 | 1 | 0 | −3.42 |
| | 0.49 | 500 | 10 | 6 | 1 | 0 | −14.23 |
| | 0.49 | 700 | 10 | 6 | 1 | 0 | −41.75 |
| | 0.4 | 105 | 10 | 6 | 1 | 0 | −12.74 |
| | 0.4 | 300 | 10 | 6 | 1 | 0 | −9.07 |
| | 0.4 | 500 | 10 | 6 | 1 | 0 | −10.81 |
| | 0.4 | 700 | 10 | 6 | 1 | 0 | −46.70 |
| [10] | 0.5 | 100 | 10 | 1 | 1 | 1 | −8.28 |
| | 0.5 | 100 | 10 | 2 | 1 | 1 | −8.77 |
| | 0.5 | 100 | 10 | 3 | 1 | 1 | −6.16 |
| | 0.5 | 100 | 10 | 1 | 1 | 1 | −6.62 |
| | 0.5 | 100 | 10 | 2 | 1 | 1 | −8.48 |
| | 0.5 | 100 | 10 | 3 | 1 | 1 | −1.86 |
| | 0.5 | 100 | 10 | 1 | 1 | 1 | −6.85 |
| | 0.5 | 100 | 10 | 2 | 1 | 1 | −8.95 |
| | 0.5 | 100 | 10 | 3 | 1 | 1 | −4.09 |
| | 0.5 | 300 | 10 | 1 | 1 | 1 | −7.35 |
| | 0.5 | 300 | 10 | 2 | 1 | 1 | −5.64 |
| | 0.5 | 300 | 10 | 3 | 1 | 1 | −6.67 |
| | 0.5 | 300 | 10 | 1 | 1 | 1 | −4.23 |
| | 0.5 | 300 | 10 | 2 | 1 | 1 | −6.11 |
| | 0.5 | 300 | 10 | 3 | 1 | 1 | −7.73 |
| | 0.5 | 300 | 10 | 1 | 1 | 1 | −3.16 |
| | 0.5 | 300 | 10 | 2 | 1 | 1 | −4.66 |
| | 0.5 | 300 | 10 | 3 | 1 | 1 | −7.77 |
| | 0.5 | 500 | 10 | 1 | 1 | 1 | −38.09 |

**Table A1.** *Cont.*

| REF. | Input Parameters | | | | | | Output Parameters |
|---|---|---|---|---|---|---|---|
| | W/B | T (°C) | V (°C/min) | MD (h) | C | RD (day) | P (%) |
| | 0.5 | 500 | 10 | 2 | 1 | 1 | −41.99 |
| | 0.5 | 500 | 10 | 3 | 1 | 1 | −46.16 |
| | 0.5 | 500 | 10 | 1 | 1 | 1 | −36.27 |
| | 0.5 | 500 | 10 | 2 | 1 | 1 | −41.93 |
| | 0.5 | 500 | 10 | 3 | 1 | 1 | −47.90 |
| | 0.5 | 500 | 10 | 1 | 1 | 1 | −39.07 |
| | 0.5 | 500 | 10 | 2 | 1 | 1 | −41.79 |
| | 0.5 | 500 | 10 | 3 | 1 | 1 | −48.22 |
| [10] | 0.5 | 700 | 10 | 1 | 1 | 1 | −62.35 |
| | 0.5 | 700 | 10 | 2 | 1 | 1 | −67.53 |
| | 0.5 | 700 | 10 | 3 | 1 | 1 | −68.91 |
| | 0.5 | 700 | 10 | 1 | 1 | 1 | −63.46 |
| | 0.5 | 700 | 10 | 2 | 1 | 1 | −66.48 |
| | 0.5 | 700 | 10 | 3 | 1 | 1 | −69.65 |
| | 0.5 | 700 | 10 | 1 | 1 | 1 | −66.18 |
| | 0.5 | 700 | 10 | 2 | 1 | 1 | −68.90 |
| | 0.5 | 700 | 10 | 3 | 1 | 1 | −70.65 |
| | 0.57 | 200 | 16 | 1.5 | 1 | 2 | −6.75 |
| [11] | 0.57 | 400 | 10 | 1.5 | 1 | 2 | −13.00 |
| | 0.57 | 600 | 3 | 1.5 | 1 | 2 | −48.75 |
| | 0.57 | 200 | 16 | 1.5 | 0 | 2 | −2.25 |
| | 0.57 | 400 | 10 | 1.5 | 0 | 2 | −20.00 |
| | 0.57 | 600 | 3 | 1.5 | 0 | 2 | −38.50 |
| | 0.57 | 200 | 16 | 1.5 | 1 | 2 | −17.02 |
| | 0.57 | 400 | 10 | 1.5 | 1 | 2 | −26.71 |
| | 0.57 | 600 | 3 | 1.5 | 1 | 2 | −55.79 |
| | 0.57 | 200 | 16 | 1.5 | 0 | 2 | −15.60 |
| | 0.57 | 400 | 10 | 1.5 | 0 | 2 | −29.08 |
| | 0.57 | 600 | 3 | 1.5 | 0 | 2 | −47.04 |
| | 0.57 | 200 | 16 | 1.5 | 1 | 2 | −11.33 |
| | 0.57 | 400 | 10 | 1.5 | 1 | 2 | −23.65 |
| | 0.57 | 600 | 3 | 1.5 | 1 | 2 | −61.08 |
| | 0.57 | 200 | 16 | 1.5 | 0 | 2 | −10.10 |
| | 0.57 | 400 | 10 | 1.5 | 0 | 2 | −32.51 |
| | 0.57 | 600 | 3 | 1.5 | 0 | 2 | −48.77 |
| | 0.57 | 200 | 16 | 1.5 | 1 | 2 | −7.73 |
| | 0.57 | 400 | 10 | 1.5 | 1 | 2 | −27.93 |
| | 0.57 | 600 | 3 | 1.5 | 1 | 2 | −57.61 |
| | 0.57 | 200 | 16 | 1.5 | 0 | 2 | −3.24 |
| | 0.57 | 400 | 10 | 1.5 | 0 | 2 | −32.67 |
| | 0.57 | 600 | 3 | 1.5 | 0 | 2 | −41.15 |
| | 0.57 | 200 | 16 | 1.5 | 1 | 2 | −5.97 |
| | 0.57 | 400 | 10 | 1.5 | 1 | 2 | −28.05 |
| | 0.57 | 600 | 3 | 1.5 | 1 | 2 | −62.08 |
| | 0.57 | 800 | 2.29 | 1.5 | 1 | 2 | −83.64 |
| | 0.57 | 200 | 16 | 1.5 | 0 | 2 | −3.90 |
| | 0.57 | 400 | 10 | 1.5 | 0 | 2 | −37.66 |
| | 0.57 | 600 | 3 | 1.5 | 0 | 2 | −48.05 |
| | 0.57 | 800 | 2.29 | 1.5 | 0 | 2 | −88.31 |
| | 0.42 | 300 | 30 | 1.5 | 1 | 90 | 1.37 |
| | 0.42 | 500 | 25 | 1 | 1 | 90 | 1.31 |
| | 0.42 | 700 | 17.5 | 1 | 1 | 90 | 1.78 |
| [12] | 0.42 | 900 | 11.25 | 1 | 1 | 90 | −1.75 |
| | 0.42 | 300 | 30 | 1.67 | 1 | 90 | 0.03 |
| | 0.42 | 500 | 25 | 1.67 | 1 | 90 | 4.24 |
| | 0.42 | 700 | 17.5 | 1.67 | 1 | 90 | −15.57 |

<div align="center"><b>Table A1.</b> <i>Cont.</i></div>

| REF. | Input Parameters | | | | | | Output Parameters |
|------|------|------|----------|--------|------|----------|--------|
| | W/B | T (°C) | V (°C/min) | MD (h) | C | RD (day) | P (%) |
| [12] | 0.42 | 900 | 11.25 | 1.67 | 1 | 90 | −27.11 |
| | 0.42 | 300 | 30 | 2.33 | 1 | 90 | 2.17 |
| | 0.42 | 500 | 25 | 2.33 | 1 | 90 | −5.90 |
| | 0.42 | 700 | 17.5 | 2.33 | 1 | 90 | −50.55 |
| | 0.42 | 900 | 11.25 | 2.33 | 1 | 90 | −70.10 |
| | 0.42 | 300 | 30 | 3 | 1 | 90 | 4.57 |
| | 0.42 | 500 | 25 | 3 | 1 | 90 | −4.57 |
| | 0.42 | 700 | 17.5 | 3 | 1 | 90 | −58.82 |
| | 0.42 | 900 | 11.25 | 3 | 1 | 90 | −73.57 |
| [13] | 0.42 | 300 | 30 | 1 | 0 | 90 | −2.33 |
| | 0.42 | 500 | 25 | 1 | 0 | 90 | −10.62 |
| | 0.42 | 700 | 17.5 | 1 | 0 | 90 | −15.43 |
| | 0.42 | 900 | 11.25 | 1 | 0 | 90 | −21.31 |
| | 0.42 | 300 | 30 | 1.67 | 0 | 90 | −5.22 |
| | 0.42 | 500 | 25 | 1.67 | 0 | 90 | −1.21 |
| | 0.42 | 700 | 17.5 | 1.67 | 0 | 90 | −13.78 |
| | 0.42 | 900 | 11.25 | 1.67 | 0 | 90 | −47.30 |
| | 0.42 | 300 | 30 | 2.33 | 0 | 90 | −1.42 |
| | 0.42 | 500 | 25 | 2.33 | 0 | 90 | −17.73 |
| | 0.42 | 700 | 17.5 | 2.33 | 0 | 90 | −48.21 |
| | 0.42 | 900 | 11.25 | 2.33 | 0 | 90 | −58.64 |
| | 0.42 | 300 | 30 | 3 | 0 | 90 | 2.91 |
| | 0.42 | 500 | 25 | 3 | 0 | 90 | −9.12 |
| | 0.42 | 700 | 17.5 | 3 | 0 | 90 | −63.13 |
| | 0.42 | 900 | 11.25 | 3 | 0 | 90 | −72.22 |
| [14] | 0.52 | 200 | 6.67 | 2 | 1 | 2 | −14.58 |
| | 0.52 | 200 | 6.67 | 6 | 1 | 2 | −17.17 |
| | 0.52 | 200 | 6.67 | 24 | 1 | 2 | −24.02 |
| | 0.52 | 200 | 6.67 | 48 | 1 | 2 | −26.80 |
| | 0.52 | 400 | 13.33 | 2 | 1 | 2 | −45.23 |
| | 0.52 | 400 | 13.33 | 6 | 1 | 2 | −52.54 |
| | 0.52 | 400 | 13.33 | 24 | 1 | 2 | −57.13 |
| | 0.52 | 400 | 13.33 | 48 | 1 | 2 | −60.30 |
| | 0.52 | 600 | 20 | 2 | 1 | 2 | −85.16 |
| | 0.52 | 600 | 20 | 6 | 1 | 2 | −88.72 |
| | 0.52 | 600 | 20 | 24 | 1 | 2 | −89.46 |
| | 0.52 | 600 | 20 | 48 | 1 | 2 | −90.56 |
| | 0.52 | 800 | 26.67 | 2 | 1 | 2 | −92.43 |
| | 0.52 | 800 | 26.67 | 6 | 1 | 2 | −92.79 |
| | 0.52 | 800 | 26.67 | 24 | 1 | 2 | −93.18 |
| | 0.52 | 800 | 26.67 | 48 | 1 | 2 | −93.95 |
| [15] | 0.58 | 150 | 150 | 3 | 1 | 27 | −3.25 |
| | 0.58 | 150 | 150 | 3 | 1 | 25 | 3.25 |
| | 0.58 | 150 | 150 | 3 | 1 | 21 | −15.52 |
| | 0.58 | 150 | 150 | 3 | 1 | 14 | −17.33 |
| | 0.58 | 150 | 150 | 3 | 1 | 0 | −22.74 |
| | 0.58 | 300 | 300 | 3 | 1 | 27 | −13.36 |
| | 0.58 | 300 | 300 | 3 | 1 | 25 | −8.30 |
| | 0.58 | 300 | 300 | 3 | 1 | 21 | −20.58 |
| | 0.58 | 300 | 300 | 3 | 1 | 14 | −16.35 |
| | 0.58 | 300 | 300 | 3 | 1 | 0 | −29.96 |
| | 0.58 | 400 | 200 | 3 | 1 | 27 | −19.49 |
| | 0.58 | 400 | 200 | 3 | 1 | 25 | −9.75 |
| | 0.58 | 400 | 200 | 3 | 1 | 21 | −13.00 |
| | 0.58 | 400 | 200 | 3 | 1 | 14 | −22.02 |

Table A1. *Cont.*

| REF. | Input Parameters | | | | | | Output Parameters |
|------|-----|--------|-----------|--------|---|----------|-------|
| | W/B | T (°C) | V (°C/min) | MD (h) | C | RD (day) | P (%) |
| | 0.58 | 400 | 200 | 3 | 1 | 0 | −16.97 |
| | 0.58 | 600 | 75 | 3 | 1 | 27 | −37.55 |
| | 0.58 | 600 | 75 | 3 | 1 | 25 | −23.47 |
| | 0.58 | 600 | 75 | 3 | 1 | 21 | −29.97 |
| | 0.58 | 600 | 75 | 3 | 1 | 14 | −47.29 |
| | 0.58 | 600 | 75 | 3 | 1 | 0 | −35.74 |
| | 0.58 | 800 | 35.5 | 3 | 1 | 27 | −62.82 |
| | 0.58 | 800 | 35.5 | 3 | 1 | 25 | −47.29 |
| | 0.58 | 800 | 35.5 | 3 | 1 | 21 | −51.99 |
| | 0.58 | 800 | 35.5 | 3 | 1 | 14 | −73.65 |
| | 0.58 | 800 | 35.5 | 3 | 1 | 0 | −59.93 |
| | 0.58 | 900 | 20.2 | 3 | 1 | 27 | −79.78 |
| | 0.58 | 900 | 20.2 | 3 | 1 | 25 | −69.68 |
| | 0.58 | 900 | 20.2 | 3 | 1 | 21 | −75.09 |
| | 0.58 | 900 | 20.2 | 3 | 1 | 14 | −84.12 |
| | 0.58 | 900 | 20.2 | 3 | 1 | 0 | −75.09 |
| | 0.58 | 150 | 150 | 3 | 0 | 27 | 5.12 |
| | 0.58 | 150 | 150 | 3 | 0 | 25 | 1.89 |
| | 0.58 | 150 | 150 | 3 | 0 | 21 | −1.41 |
| | 0.58 | 150 | 150 | 3 | 0 | 14 | −5.45 |
| | 0.58 | 150 | 150 | 3 | 0 | 0 | −27.17 |
| | 0.58 | 300 | 300 | 3 | 0 | 27 | 10.91 |
| | 0.58 | 300 | 300 | 3 | 0 | 25 | 19.22 |
| [15] | 0.58 | 300 | 300 | 3 | 0 | 21 | 1.89 |
| | 0.58 | 300 | 300 | 3 | 0 | 14 | −9.17 |
| | 0.58 | 300 | 300 | 3 | 0 | 0 | −31.73 |
| | 0.58 | 400 | 200 | 3 | 0 | 27 | 1.81 |
| | 0.58 | 400 | 200 | 3 | 0 | 25 | 6.39 |
| | 0.58 | 400 | 200 | 3 | 0 | 21 | 3.55 |
| | 0.58 | 400 | 200 | 3 | 0 | 14 | −2.55 |
| | 0.58 | 400 | 200 | 3 | 0 | 0 | −18.90 |
| | 0.58 | 600 | 75 | 3 | 0 | 27 | −25.92 |
| | 0.58 | 600 | 75 | 3 | 0 | 25 | −11.40 |
| | 0.58 | 600 | 75 | 3 | 0 | 21 | −24.59 |
| | 0.58 | 600 | 75 | 3 | 0 | 14 | −59.04 |
| | 0.58 | 600 | 75 | 3 | 0 | 0 | −76.83 |
| | 0.58 | 800 | 35.5 | 3 | 0 | 27 | −42.47 |
| | 0.58 | 800 | 35.5 | 3 | 0 | 25 | −30.85 |
| | 0.58 | 800 | 35.5 | 3 | 0 | 21 | −38.24 |
| | 0.58 | 800 | 35.5 | 3 | 0 | 14 | −57.59 |
| | 0.58 | 800 | 35.5 | 3 | 0 | 0 | −76.83 |
| | 0.58 | 900 | 20.2 | 3 | 0 | 27 | −65.64 |
| | 0.58 | 900 | 20.2 | 3 | 0 | 25 | −51.54 |
| | 0.58 | 900 | 20.2 | 3 | 0 | 21 | −56.03 |
| | 0.58 | 900 | 20.2 | 3 | 0 | 14 | −76.21 |
| | 0.58 | 900 | 20.2 | 3 | 0 | 0 | −87.17 |
| | 0.34 | 200 | 5 | 3 | 1 | 0 | −3.57 |
| | 0.34 | 400 | 5 | 3 | 1 | 0 | −27.26 |
| | 0.34 | 600 | 5 | 3 | 1 | 0 | −59.08 |
| | 0.4 | 200 | 5 | 3 | 1 | 0 | −8.24 |
| [16] | 0.4 | 400 | 5 | 3 | 1 | 0 | −28.92 |
| | 0.4 | 600 | 5 | 3 | 1 | 0 | −57.47 |
| | 0.5 | 200 | 5 | 3 | 1 | 0 | −10.54 |
| | 0.5 | 400 | 5 | 3 | 1 | 0 | −23.49 |
| | 0.5 | 600 | 5 | 3 | 1 | 0 | −53.37 |

**Table A1.** *Cont.*

| REF. | Input Parameters | | | | | | Output Parameters |
|------|-----|-------|----------|--------|---|----------|-------|
| | W/B | T (°C) | V (°C/min) | MD (h) | C | RD (day) | P (%) |
| [17] | 0.49 | 200 | 2.5 | 8 | 1 | 0 | 41.30 |
| | 0.49 | 400 | 2.5 | 8 | 1 | 0 | 7.60 |
| | 0.49 | 600 | 2.5 | 8 | 1 | 0 | −33.20 |
| | 0.49 | 700 | 2.5 | 8 | 1 | 0 | −46.70 |
| | 0.49 | 800 | 2.5 | 8 | 1 | 0 | −66.30 |
| | 0.49 | 200 | 2.5 | 8 | 1 | 0 | −23.40 |
| | 0.49 | 400 | 2.5 | 8 | 1 | 0 | −46.50 |
| | 0.49 | 600 | 2.5 | 8 | 1 | 0 | −55.50 |
| | 0.49 | 700 | 2.5 | 8 | 1 | 0 | −65.50 |
| | 0.49 | 800 | 2.5 | 8 | 1 | 0 | −67.60 |
| | 0.49 | 900 | 2.5 | 8 | 1 | 0 | −82.50 |
| [18] | 0.42 | 300 | 10 | 6 | 1 | 0 | −27.00 |
| | 0.42 | 500 | 10 | 6 | 1 | 0 | −44.00 |
| | 0.42 | 700 | 10 | 6 | 1 | 0 | −64.00 |
| | 0.42 | 300 | 10 | 6 | 1 | 0 | −14.00 |
| | 0.42 | 500 | 10 | 6 | 1 | 0 | −33.00 |
| | 0.42 | 700 | 10 | 6 | 1 | 0 | −67.00 |
| [19] | 0.25 | 200 | 90 | 0.5 | 1 | 1 | −4.59 |
| | 0.25 | 400 | 90 | 0.5 | 1 | 1 | −13.33 |
| | 0.25 | 600 | 54 | 0.5 | 1 | 1 | −32.43 |
| | 0.25 | 800 | 28.42 | 0.5 | 1 | 1 | −44.40 |
| | 0.3 | 200 | 90 | 0.5 | 1 | 1 | −4.52 |
| | 0.3 | 400 | 90 | 0.5 | 1 | 1 | −12.36 |
| | 0.3 | 600 | 54 | 0.5 | 1 | 1 | −31.62 |
| | 0.3 | 800 | 28.42 | 0.5 | 1 | 1 | −46.36 |
| [20] | 0.31 | 200 | 10 | 3 | 1 | 0 | 4.63 |
| | 0.31 | 400 | 10 | 3 | 1 | 0 | 45.66 |
| | 0.31 | 600 | 10 | 3 | 1 | 0 | 7.07 |
| | 0.31 | 800 | 10 | 3 | 1 | 0 | −56.30 |
| | 0.31 | 200 | 10 | 3 | 1 | 0 | 38.68 |
| | 0.31 | 400 | 10 | 3 | 1 | 0 | 52.44 |
| | 0.31 | 600 | 10 | 3 | 1 | 0 | 23.40 |
| | 0.31 | 800 | 10 | 3 | 1 | 0 | −49.33 |
| [21] | 0.3 | 200 | 16 | 2 | 1 | 0 | −10.55 |
| | 0.3 | 400 | 20.5 | 2 | 1 | 0 | −29.61 |
| | 0.3 | 600 | 17.5 | 2 | 1 | 0 | −47.08 |
| | 0.3 | 800 | 13.4 | 2 | 1 | 0 | −75.64 |
| | 0.3 | 200 | 16 | 2 | 1 | 0 | −12.78 |
| | 0.3 | 400 | 20.5 | 2 | 1 | 0 | −31.86 |
| | 0.3 | 600 | 17.5 | 2 | 1 | 0 | −48.60 |
| | 0.3 | 800 | 13.4 | 2 | 1 | 0 | −78.01 |
| | 0.3 | 200 | 16 | 2 | 1 | 0 | −15.41 |
| | 0.3 | 400 | 20.5 | 2 | 1 | 0 | −37.21 |
| | 0.3 | 600 | 17.5 | 2 | 1 | 0 | −51.17 |
| | 0.3 | 800 | 13.4 | 2 | 1 | 0 | −78.30 |
| | 0.3 | 200 | 16 | 2 | 1 | 0 | −10.35 |
| | 0.3 | 400 | 20.5 | 2 | 1 | 0 | −35.24 |
| | 0.3 | 600 | 17.5 | 2 | 1 | 0 | −49.20 |
| | 0.3 | 800 | 13.4 | 2 | 1 | 0 | −79.57 |
| [22] | 0.48 | 200 | 10 | 1.5 | 1 | 0 | −16.21 |
| | 0.48 | 400 | 10 | 1.5 | 1 | 0 | −10.48 |
| | 0.48 | 800 | 10 | 1.5 | 1 | 0 | −4.68 |
| | 0.48 | 200 | 10 | 1.5 | 1 | 0 | 4.29 |
| | 0.48 | 400 | 10 | 1.5 | 1 | 0 | −69.21 |
| | 0.48 | 800 | 10 | 1.5 | 1 | 0 | −31.90 |

**Table A1.** *Cont.*

| REF. | Input Parameters | | | | | | Output Parameters |
|---|---|---|---|---|---|---|---|
| | W/B | T (°C) | V (°C/min) | MD (h) | C | RD (day) | P (%) |
| [23] | 0.35 | 200 | 12 | 2 | 1 | 0 | −18.16 |
| | 0.35 | 300 | 12 | 2 | 1 | 0 | −28.18 |
| | 0.35 | 400 | 12 | 2 | 1 | 0 | −22.24 |
| | 0.35 | 600 | 12 | 2 | 1 | 0 | −41.64 |
| | 0.35 | 800 | 12 | 2 | 1 | 0 | −69.66 |
| | 0.35 | 200 | 12 | 2 | 1 | 0 | −8.54 |
| | 0.35 | 300 | 12 | 2 | 1 | 0 | −17.24 |
| | 0.35 | 400 | 12 | 2 | 1 | 0 | −15.05 |
| | 0.35 | 600 | 12 | 2 | 1 | 0 | −37.74 |
| | 0.35 | 800 | 12 | 2 | 1 | 0 | −72.94 |
| | 0.35 | 200 | 12 | 2 | 1 | 0 | −7.32 |
| | 0.35 | 300 | 12 | 2 | 1 | 0 | −16.47 |
| | 0.35 | 400 | 12 | 2 | 1 | 0 | −14.25 |
| | 0.35 | 600 | 12 | 2 | 1 | 0 | −36.24 |
| | 0.35 | 800 | 12 | 2 | 1 | 0 | −66.57 |
| | 0.35 | 200 | 12 | 2 | 1 | 0 | −14.66 |
| | 0.35 | 300 | 12 | 2 | 1 | 0 | −23.60 |
| | 0.35 | 400 | 12 | 2 | 1 | 0 | −21.76 |
| | 0.35 | 600 | 12 | 2 | 1 | 0 | −41.94 |
| | 0.35 | 800 | 12 | 2 | 1 | 0 | −71.91 |
| [24] | 0.33 | 200 | 15 | 2 | 1 | 1 | −8.24 |
| | 0.33 | 400 | 15 | 2 | 1 | 1 | −19.06 |
| | 0.33 | 600 | 15 | 2 | 1 | 1 | −40.14 |
| | 0.33 | 800 | 15 | 2 | 1 | 1 | −65.84 |
| | 0.33 | 200 | 15 | 2 | 1 | 1 | −5.55 |
| | 0.33 | 400 | 15 | 2 | 1 | 1 | −14.62 |
| | 0.33 | 600 | 15 | 2 | 1 | 1 | −32.71 |
| | 0.33 | 800 | 15 | 2 | 1 | 1 | −44.80 |
| | 0.33 | 200 | 15 | 2 | 1 | 1 | −9.10 |
| | 0.33 | 400 | 15 | 2 | 1 | 1 | −16.28 |
| | 0.33 | 600 | 15 | 2 | 1 | 1 | −32.99 |
| | 0.33 | 800 | 15 | 2 | 1 | 1 | −45.90 |
| | 0.33 | 200 | 15 | 2 | 1 | 1 | −9.07 |
| | 0.33 | 400 | 15 | 2 | 1 | 1 | −20.40 |
| | 0.33 | 600 | 15 | 2 | 1 | 1 | −33.99 |
| | 0.33 | 800 | 15 | 2 | 1 | 1 | −48.03 |
| [25] | 0.27 | 200 | 10 | 1.5 | 1 | 0 | −0.13 |
| | 0.27 | 400 | 10 | 1.5 | 1 | 0 | −7.02 |
| | 0.27 | 600 | 10 | 1.5 | 1 | 0 | −28.61 |
| | 0.27 | 800 | 10 | 1.5 | 1 | 0 | −71.52 |
| | 0.27 | 200 | 10 | 1.5 | 1 | 0 | −19.42 |
| | 0.27 | 400 | 10 | 1.5 | 1 | 0 | −30.87 |
| | 0.27 | 600 | 10 | 1.5 | 1 | 0 | −37.39 |
| | 0.27 | 800 | 10 | 1.5 | 1 | 0 | −70.14 |
| [26] | 0.49 | 800 | 6 | 1 | 1 | 0 | −74.41 |
| | 0.49 | 1100 | 6 | 1 | 1 | 0 | −91.88 |
| | 0.49 | 800 | 6 | 1 | 0 | 0 | −78.62 |
| | 0.49 | 1100 | 6 | 1 | 0 | 0 | −91.57 |
| | 0.49 | 800 | 6 | 1 | 1 | 0 | −65.90 |
| | 0.49 | 1100 | 6 | 1 | 1 | 0 | −87.75 |
| | 0.49 | 800 | 6 | 1 | 0 | 0 | −71.63 |
| | 0.49 | 1100 | 6 | 1 | 0 | 0 | −89.60 |
| [27] | 0.49 | 250 | 10 | 2 | 0 | 7 | −2.03 |
| | 0.49 | 450 | 10 | 2 | 0 | 7 | −3.36 |
| | 0.49 | 250 | 10 | 2 | 0 | 7 | −7.90 |

Table A1. *Cont.*

| REF. | Input Parameters | | | | | | Output Parameters |
|---|---|---|---|---|---|---|---|
| | W/B | T (°C) | V (°C/min) | MD (h) | C | RD (day) | P (%) |
| [27] | 0.49 | 450 | 10 | 2 | 0 | 7 | −21.94 |
| | 0.49 | 250 | 10 | 2 | 0 | 7 | −17.92 |
| | 0.49 | 450 | 10 | 2 | 0 | 7 | −28.29 |
| | 0.49 | 250 | 10 | 2 | 0 | 7 | −23.79 |
| | 0.49 | 450 | 10 | 2 | 0 | 7 | −31.72 |
| | 0.49 | 550 | 10 | 2 | 0 | 7 | −46.32 |
| | 0.49 | 650 | 10 | 2 | 0 | 7 | −66.78 |
| [28] | 0.55 | 100 | 5 | 1 | 1 | 2 | −9.01 |
| | 0.55 | 300 | 5 | 1 | 1 | 2 | −26.12 |
| | 0.55 | 500 | 5 | 1 | 1 | 2 | −36.94 |
| | 0.55 | 700 | 5 | 1 | 1 | 2 | −47.75 |
| | 0.33 | 100 | 5 | 1 | 1 | 2 | −8.74 |
| | 0.33 | 300 | 5 | 1 | 1 | 2 | −33.01 |
| | 0.33 | 500 | 5 | 1 | 1 | 2 | −45.63 |
| | 0.33 | 700 | 5 | 1 | 1 | 2 | −69.90 |
| [29] | 0.36 | 200 | 20 | 1.5 | 0 | 2 | −4.30 |
| | 0.36 | 300 | 20 | 1.5 | 0 | 2 | −4.70 |
| | 0.36 | 400 | 20 | 2 | 0 | 2 | −15.30 |
| | 0.36 | 500 | 20 | 2 | 0 | 2 | −35.60 |
| | 0.36 | 600 | 20 | 2.5 | 0 | 2 | −40.50 |
| | 0.36 | 700 | 20 | 2.5 | 0 | 2 | −44.20 |
| | 0.36 | 800 | 20 | 2.5 | 0 | 2 | −69.20 |
| [30] | 0.44 | 100 | 100 | 2 | 1 | 0.25 | 19.38 |
| | 0.44 | 100 | 100 | 2 | 1 | 0.25 | 0.32 |
| | 0.44 | 100 | 100 | 2 | 1 | 0.25 | −0.66 |
| | 0.44 | 200 | 200 | 2 | 1 | 0.25 | 1.94 |
| | 0.44 | 200 | 200 | 2 | 1 | 0.25 | −7.05 |
| | 0.44 | 200 | 200 | 2 | 1 | 0.25 | 4.32 |
| | 0.44 | 300 | 300 | 2 | 1 | 0.25 | 6.20 |
| | 0.44 | 300 | 300 | 2 | 1 | 0.25 | −16.99 |
| | 0.44 | 300 | 300 | 2 | 1 | 0.25 | −9.97 |
| | 0.44 | 400 | 400 | 2 | 1 | 0.25 | 1.94 |
| | 0.44 | 400 | 400 | 2 | 1 | 0.25 | −14.10 |
| | 0.44 | 400 | 400 | 2 | 1 | 0.25 | −18.27 |
| | 0.44 | 500 | 500 | 2 | 1 | 0.25 | −7.75 |
| | 0.44 | 500 | 500 | 2 | 1 | 0.25 | −19.87 |
| | 0.44 | 500 | 500 | 2 | 1 | 0.25 | −15.61 |
| | 0.44 | 600 | 2.5 | 1 | 1 | 0.25 | −24.81 |
| | 0.44 | 600 | 2.5 | 1 | 1 | 0.25 | −32.69 |
| | 0.44 | 600 | 2.5 | 1 | 1 | 0.25 | −31.89 |
| | 0.44 | 700 | 2.5 | 1 | 1 | 0.25 | −47.67 |
| | 0.44 | 700 | 2.5 | 1 | 1 | 0.25 | −50.64 |
| | 0.44 | 700 | 2.5 | 1 | 1 | 0.25 | −57.81 |
| | 0.44 | 800 | 2.5 | 1 | 1 | 0.25 | −54.26 |
| | 0.44 | 800 | 2.5 | 1 | 1 | 0.25 | −60.90 |
| | 0.44 | 800 | 2.5 | 1 | 1 | 0.25 | −72.09 |
| | 0.44 | 900 | 2.5 | 1 | 1 | 0.25 | −74.03 |
| | 0.44 | 900 | 2.5 | 1 | 1 | 0.25 | −77.88 |
| | 0.44 | 900 | 2.5 | 1 | 1 | 0.25 | −74.42 |
| | 0.35 | 100 | 100 | 2 | 1 | 0.25 | −1.48 |
| | 0.35 | 100 | 100 | 2 | 1 | 0.25 | 2.08 |
| | 0.35 | 100 | 100 | 2 | 1 | 0.25 | 6.04 |
| | 0.35 | 200 | 200 | 2 | 1 | 0.25 | −8.62 |
| | 0.35 | 200 | 200 | 2 | 1 | 0.25 | −7.79 |
| | 0.35 | 200 | 200 | 2 | 1 | 0.25 | 0.00 |
| | 0.35 | 300 | 300 | 2 | 1 | 0.25 | −12.56 |

**Table A1.** *Cont.*

| REF. | Input Parameters | | | | | | Output Parameters |
|---|---|---|---|---|---|---|---|
| | W/B | T (°C) | V (°C/min) | MD (h) | C | RD (day) | P (%) |
| | 0.35 | 300 | 300 | 2 | 1 | 0.25 | −3.64 |
| | 0.35 | 300 | 300 | 2 | 1 | 0.25 | −12.09 |
| | 0.35 | 400 | 400 | 2 | 1 | 0.25 | −20.69 |
| | 0.35 | 400 | 400 | 2 | 1 | 0.25 | −16.88 |
| | 0.35 | 400 | 400 | 2 | 1 | 0.25 | −7.69 |
| | 0.35 | 500 | 500 | 2 | 1 | 0.25 | −26.60 |
| | 0.35 | 500 | 500 | 2 | 1 | 0.25 | −24.68 |
| | 0.35 | 500 | 500 | 2 | 1 | 0.25 | −22.53 |
| | 0.35 | 600 | 2.5 | 1 | 1 | 0.25 | −40.89 |
| | 0.35 | 600 | 2.5 | 1 | 1 | 0.25 | −38.18 |
| | 0.35 | 600 | 2.5 | 1 | 1 | 0.25 | −32.69 |
| | 0.35 | 700 | 2.5 | 1 | 1 | 0.25 | −43.60 |
| | 0.35 | 700 | 2.5 | 1 | 1 | 0.25 | −45.19 |
| | 0.35 | 700 | 2.5 | 1 | 1 | 0.25 | −32.97 |
| | 0.35 | 800 | 2.5 | 1 | 1 | 0.25 | −60.59 |
| | 0.35 | 800 | 2.5 | 1 | 1 | 0.25 | −60.00 |
| | 0.35 | 800 | 2.5 | 1 | 1 | 0.25 | −52.75 |
| | 0.35 | 900 | 2.5 | 1 | 1 | 0.25 | −78.08 |
| | 0.35 | 900 | 2.5 | 1 | 1 | 0.25 | −80.52 |
| | 0.35 | 900 | 2.5 | 1 | 1 | 0.25 | −75.82 |
| | 0.35 | 100 | 100 | 2 | 1 | 0.25 | 2.90 |
| | 0.35 | 100 | 100 | 2 | 1 | 0.25 | 2.31 |
| [30] | 0.35 | 100 | 100 | 2 | 1 | 0.25 | 4.78 |
| | 0.35 | 200 | 200 | 2 | 1 | 0.25 | −9.54 |
| | 0.35 | 200 | 200 | 2 | 1 | 0.25 | −9.64 |
| | 0.35 | 200 | 200 | 2 | 1 | 0.25 | −9.56 |
| | 0.35 | 300 | 300 | 2 | 1 | 0.25 | −16.80 |
| | 0.35 | 300 | 300 | 2 | 1 | 0.25 | −18.03 |
| | 0.35 | 300 | 300 | 2 | 1 | 0.25 | −17.67 |
| | 0.35 | 400 | 400 | 2 | 1 | 0.25 | −18.67 |
| | 0.35 | 400 | 400 | 2 | 1 | 0.25 | −25.37 |
| | 0.35 | 400 | 400 | 2 | 1 | 0.25 | −22.25 |
| | 0.35 | 500 | 500 | 2 | 1 | 0.25 | −27.59 |
| | 0.35 | 500 | 500 | 2 | 1 | 0.25 | −27.67 |
| | 0.35 | 500 | 500 | 2 | 1 | 0.25 | −27.86 |
| | 0.35 | 600 | 2.5 | 1 | 1 | 0.25 | −31.33 |
| | 0.35 | 600 | 2.5 | 1 | 1 | 0.25 | −33.33 |
| | 0.35 | 600 | 2.5 | 1 | 1 | 0.25 | −32.85 |
| | 0.35 | 700 | 2.5 | 1 | 1 | 0.25 | −37.97 |
| | 0.35 | 700 | 2.5 | 1 | 1 | 0.25 | −41.72 |
| | 0.35 | 700 | 2.5 | 1 | 1 | 0.25 | −40.33 |
| | 0.35 | 800 | 2.5 | 1 | 1 | 0.25 | −61.83 |
| | 0.35 | 800 | 2.5 | 1 | 1 | 0.25 | −58.07 |
| | 0.35 | 800 | 2.5 | 1 | 1 | 0.25 | −61.33 |
| | 0.35 | 900 | 2.5 | 1 | 1 | 0.25 | −88.80 |
| | 0.46 | 100 | 11 | 6 | 1 | 1 | −4.00 |
| | 0.46 | 300 | 11 | 6 | 1 | 1 | −15.00 |
| | 0.46 | 500 | 11 | 6 | 1 | 1 | −45.00 |
| | 0.46 | 700 | 11 | 6 | 1 | 1 | −73.00 |
| | 0.46 | 100 | 11 | 6 | 1 | 7 | −14.00 |
| | 0.46 | 300 | 11 | 6 | 1 | 7 | −18.00 |
| [31] | 0.46 | 500 | 11 | 6 | 1 | 7 | −58.00 |
| | 0.46 | 700 | 11 | 6 | 1 | 7 | −79.00 |
| | 0.46 | 100 | 11 | 6 | 1 | 14 | 4.00 |
| | 0.46 | 300 | 11 | 6 | 1 | 14 | −21.00 |
| | 0.46 | 500 | 11 | 6 | 1 | 14 | −60.00 |
| | 0.46 | 700 | 11 | 6 | 1 | 14 | −78.00 |

Table A1. *Cont.*

| REF. | Input Parameters | | | | | | Output Parameters |
|------|-----|--------|-----------|--------|---|----------|-------|
|      | W/B | T (°C) | V (°C/min) | MD (h) | C | RD (day) | P (%) |
| [31] | 0.46 | 100 | 11 | 6 | 1 | 28 | 0.00 |
|      | 0.46 | 300 | 11 | 6 | 1 | 28 | −23.00 |
|      | 0.46 | 500 | 11 | 6 | 1 | 28 | −63.00 |
|      | 0.46 | 700 | 11 | 6 | 1 | 28 | −80.00 |
|      | 0.46 | 100 | 11 | 6 | 1 | 56 | −9.00 |
|      | 0.46 | 300 | 11 | 6 | 1 | 56 | −17.00 |
|      | 0.46 | 500 | 11 | 6 | 1 | 56 | −58.00 |
|      | 0.46 | 700 | 11 | 6 | 1 | 56 | −75.00 |
|      | 0.46 | 100 | 11 | 6 | 1 | 77 | −16.00 |
|      | 0.46 | 300 | 11 | 6 | 1 | 77 | −26.00 |
|      | 0.46 | 500 | 11 | 6 | 1 | 77 | −52.00 |
|      | 0.46 | 700 | 11 | 6 | 1 | 77 | −77.00 |
|      | 0.46 | 100 | 11 | 6 | 1 | 112 | −5.00 |
|      | 0.46 | 300 | 11 | 6 | 1 | 112 | −23.00 |
|      | 0.46 | 500 | 11 | 6 | 1 | 112 | −57.00 |
|      | 0.46 | 700 | 11 | 6 | 1 | 112 | −76.00 |
| [32] | 0.43 | 200 | 20 | 1 | 1 | 0 | −24.30 |
|      | 0.43 | 300 | 17.5 | 1 | 1 | 0 | −18.66 |
|      | 0.43 | 400 | 2.24 | 1 | 1 | 0 | −24.66 |
|      | 0.43 | 500 | 1.84 | 1 | 1 | 0 | −28.99 |
|      | 0.43 | 600 | 1.16 | 1 | 1 | 0 | −41.69 |
|      | 0.43 | 700 | 1.04 | 1 | 1 | 0 | −62.96 |
|      | 0.43 | 800 | 1.02 | 1 | 1 | 0 | −70.94 |
| [33] | 0.56 | 150 | 10 | 1.5 | 1 | 0 | −3.80 |
|      | 0.56 | 250 | 10 | 1.5 | 1 | 0 | −8.70 |
|      | 0.56 | 350 | 10 | 1.5 | 1 | 0 | −13.60 |
|      | 0.56 | 450 | 10 | 1.5 | 1 | 0 | −21.50 |
|      | 0.56 | 550 | 10 | 1.5 | 1 | 0 | −37.70 |
|      | 0.56 | 650 | 10 | 1.5 | 1 | 0 | −53.60 |
|      | 0.56 | 150 | 10 | 1.5 | 0 | 0 | −11.30 |
|      | 0.56 | 250 | 10 | 1.5 | 0 | 0 | −18.00 |
|      | 0.56 | 350 | 10 | 1.5 | 0 | 0 | −22.60 |
|      | 0.56 | 450 | 10 | 1.5 | 0 | 0 | −25.70 |
|      | 0.56 | 550 | 10 | 1.5 | 0 | 0 | −29.80 |
|      | 0.56 | 650 | 10 | 1.5 | 0 | 0 | −35.10 |
|      | 0.53 | 150 | 10 | 1.5 | 1 | 0 | −4.20 |
|      | 0.53 | 250 | 10 | 1.5 | 1 | 0 | −11.70 |
|      | 0.53 | 350 | 10 | 1.5 | 1 | 0 | −15.90 |
|      | 0.53 | 450 | 10 | 1.5 | 1 | 0 | −18.40 |
|      | 0.53 | 550 | 10 | 1.5 | 1 | 0 | −33.30 |
|      | 0.53 | 650 | 10 | 1.5 | 1 | 0 | −51.50 |
|      | 0.53 | 150 | 10 | 1.5 | 0 | 0 | −7.40 |
|      | 0.53 | 250 | 10 | 1.5 | 0 | 0 | −14.60 |
|      | 0.53 | 350 | 10 | 1.5 | 0 | 0 | −21.00 |
|      | 0.53 | 450 | 10 | 1.5 | 0 | 0 | −23.00 |
|      | 0.53 | 550 | 10 | 1.5 | 0 | 0 | −24.30 |
|      | 0.53 | 650 | 10 | 1.5 | 0 | 0 | −31.40 |
|      | 0.5 | 150 | 10 | 1.5 | 1 | 0 | −6.10 |
|      | 0.5 | 250 | 10 | 1.5 | 1 | 0 | −13.00 |
|      | 0.5 | 350 | 10 | 1.5 | 1 | 0 | −18.60 |
|      | 0.5 | 450 | 10 | 1.5 | 1 | 0 | −23.50 |
|      | 0.5 | 550 | 10 | 1.5 | 1 | 0 | −35.40 |
|      | 0.5 | 650 | 10 | 1.5 | 1 | 0 | −56.20 |
|      | 0.5 | 150 | 10 | 1.5 | 0 | 0 | −7.80 |
|      | 0.5 | 250 | 10 | 1.5 | 0 | 0 | −15.70 |
|      | 0.5 | 350 | 10 | 1.5 | 0 | 0 | −21.70 |

Table A1. *Cont.*

| REF. | Input Parameters | | | | | | Output Parameters |
|---|---|---|---|---|---|---|---|
| | W/B | T (°C) | V (°C/min) | MD (h) | C | RD (day) | P (%) |
| [33] | 0.5 | 450 | 10 | 1.5 | 0 | 0 | −29.00 |
| | 0.5 | 550 | 10 | 1.5 | 0 | 0 | −25.80 |
| | 0.5 | 650 | 10 | 1.5 | 0 | 0 | −31.30 |
| [34] | 0.77 | 200 | 2.5 | 1 | 1 | 1 | −8.40 |
| | 0.77 | 400 | 2.5 | 1 | 1 | 1 | −12.38 |
| | 0.77 | 800 | 2.5 | 1 | 1 | 1 | −84.51 |
| | 0.77 | 200 | 2.5 | 1 | 1 | 1 | −4.23 |
| | 0.77 | 400 | 2.5 | 1 | 1 | 1 | −13.93 |
| | 0.77 | 800 | 2.5 | 1 | 1 | 1 | −70.04 |
| | 0.77 | 200 | 2.5 | 1 | 1 | 1 | −7.01 |
| | 0.77 | 400 | 2.5 | 1 | 1 | 1 | −14.02 |
| | 0.77 | 800 | 2.5 | 1 | 1 | 1 | −68.49 |
| | 0.77 | 200 | 2.5 | 1 | 1 | 1 | −3.59 |
| | 0.77 | 400 | 2.5 | 1 | 1 | 1 | −25.01 |
| | 0.77 | 800 | 2.5 | 1 | 1 | 1 | −77.04 |
| | 0.77 | 200 | 2.5 | 1 | 1 | 1 | −1.51 |
| | 0.77 | 400 | 2.5 | 1 | 1 | 1 | −6.82 |
| | 0.77 | 800 | 2.5 | 1 | 1 | 1 | −64.01 |
| | 0.77 | 200 | 2.5 | 1 | 1 | 1 | −7.97 |
| | 0.77 | 400 | 2.5 | 1 | 1 | 1 | −16.88 |
| | 0.77 | 800 | 2.5 | 1 | 1 | 1 | −56.69 |
| | 0.77 | 200 | 2.5 | 1 | 1 | 1 | −11.30 |
| | 0.77 | 400 | 2.5 | 1 | 1 | 1 | −14.78 |
| | 0.77 | 800 | 2.5 | 1 | 1 | 1 | −56.23 |
| | 0.77 | 200 | 2.5 | 1 | 1 | 1 | −11.69 |
| | 0.77 | 400 | 2.5 | 1 | 1 | 1 | −35.50 |
| | 0.77 | 800 | 2.5 | 1 | 1 | 1 | −56.28 |
| [35] | 0.29 | 851 | 2 | 4 | 1 | 0 | −78.80 |
| | 0.29 | 851 | 10 | 4 | 1 | 0 | −57.60 |
| | 0.29 | 851 | 100 | 4 | 1 | 0 | −8.00 |
| | 0.25 | 851 | 2 | 4 | 1 | 0 | −78.30 |
| | 0.25 | 851 | 10 | 4 | 1 | 0 | −68.90 |
| | 0.25 | 851 | 100 | 4 | 1 | 0 | −16.20 |
| | 0.27 | 851 | 2 | 4 | 1 | 0 | −74.20 |
| | 0.27 | 851 | 10 | 4 | 1 | 0 | −73.70 |
| | 0.27 | 851 | 100 | 4 | 1 | 0 | −7.00 |
| | 0.24 | 851 | 2 | 4 | 1 | 0 | −75.70 |
| | 0.24 | 851 | 10 | 4 | 1 | 0 | −78.00 |
| | 0.24 | 851 | 100 | 4 | 1 | 0 | −11.50 |
| [36] | 0.43 | 200 | 10 | 6 | 1 | 1 | −0.97 |
| | 0.43 | 300 | 10 | 6 | 1 | 1 | 6.31 |
| | 0.43 | 400 | 10 | 6 | 1 | 1 | −13.56 |
| | 0.43 | 500 | 10 | 6 | 1 | 1 | −21.81 |
| | 0.37 | 200 | 10 | 6 | 1 | 1 | −2.46 |
| | 0.37 | 300 | 10 | 6 | 1 | 1 | −18.60 |
| | 0.37 | 400 | 10 | 6 | 1 | 1 | −13.31 |
| | 0.37 | 500 | 10 | 6 | 1 | 1 | −28.04 |
| [37] | 0.4 | 100 | 3 | 3 | 1 | 0 | −13.32 |
| | 0.4 | 200 | 3 | 3 | 1 | 0 | −10.05 |
| | 0.4 | 300 | 3 | 3 | 1 | 0 | −24.80 |
| | 0.4 | 600 | 3 | 3 | 1 | 0 | −66.47 |
| | 0.35 | 100 | 3 | 3 | 1 | 0 | −15.04 |
| | 0.35 | 200 | 3 | 3 | 1 | 0 | −12.75 |
| | 0.35 | 300 | 3 | 3 | 1 | 0 | −23.80 |
| | 0.35 | 600 | 3 | 3 | 1 | 0 | −70.16 |

**Table A1.** *Cont.*

| REF. | Input Parameters | | | | | | Output Parameters |
|---|---|---|---|---|---|---|---|
| | W/B | T (°C) | V (°C/min) | MD (h) | C | RD (day) | P (%) |
| [37] | 0.3 | 100 | 3 | 3 | 1 | 0 | −15.52 |
| | 0.3 | 200 | 3 | 3 | 1 | 0 | −14.72 |
| | 0.3 | 300 | 3 | 3 | 1 | 0 | −30.95 |
| | 0.3 | 600 | 3 | 3 | 1 | 0 | −73.35 |
| | 0.3 | 100 | 3 | 3 | 1 | 0 | −14.52 |
| | 0.3 | 200 | 3 | 3 | 1 | 0 | −11.99 |
| | 0.3 | 300 | 3 | 3 | 1 | 0 | −27.48 |
| | 0.3 | 600 | 3 | 3 | 1 | 0 | −69.15 |
| [38] | 0.29 | 600 | 2.5 | 0 | 1 | 0 | −52.55 |
| | 0.29 | 800 | 2.5 | 0 | 1 | 0 | −74.47 |
| | 0.29 | 600 | 2.5 | 0 | 1 | 0 | −45.45 |
| | 0.29 | 800 | 2.5 | 0 | 1 | 0 | −66.67 |
| | 0.29 | 600 | 2.5 | 0 | 1 | 0 | −49.91 |
| | 0.29 | 800 | 2.5 | 0 | 1 | 0 | −75.08 |
| | 0.29 | 600 | 2.5 | 0 | 1 | 0 | −48.00 |
| | 0.29 | 800 | 2.5 | 0 | 1 | 0 | −67.63 |
| | 0.29 | 600 | 2.5 | 0 | 1 | 0 | −61.35 |
| | 0.29 | 800 | 2.5 | 0 | 1 | 0 | −81.70 |
| | 0.29 | 600 | 2.5 | 0 | 1 | 0 | −55.75 |
| | 0.29 | 800 | 2.5 | 0 | 1 | 0 | −75.53 |
| | 0.29 | 600 | 2.5 | 0 | 1 | 0 | −59.23 |
| | 0.29 | 800 | 2.5 | 0 | 1 | 0 | −82.60 |
| | 0.29 | 600 | 2.5 | 0 | 1 | 0 | −62.54 |
| | 0.29 | 800 | 2.5 | 0 | 1 | 0 | −82.87 |
| | 0.29 | 600 | 2.5 | 0 | 1 | 0 | −55.06 |
| | 0.29 | 800 | 2.5 | 0 | 1 | 0 | −79.29 |
| | 0.29 | 600 | 2.5 | 0 | 1 | 0 | −54.30 |
| | 0.29 | 800 | 2.5 | 0 | 1 | 0 | −75.18 |
| | 0.29 | 600 | 2.5 | 0 | 1 | 0 | −53.18 |
| | 0.29 | 800 | 2.5 | 0 | 1 | 0 | −71.97 |
| | 0.29 | 600 | 2.5 | 0 | 1 | 0 | −54.95 |
| | 0.29 | 800 | 2.5 | 0 | 1 | 0 | −80.99 |
| | 0.29 | 600 | 2.5 | 0 | 1 | 0 | −58.42 |
| | 0.29 | 800 | 2.5 | 0 | 1 | 0 | −82.65 |
| | 0.29 | 600 | 2.5 | 0 | 1 | 0 | −54.54 |
| | 0.29 | 800 | 2.5 | 0 | 1 | 0 | −76.78 |
| [39] | 0.2 | 120 | 4 | 2 | 1 | 3 | 0.04 |
| | 0.2 | 200 | 4 | 2 | 1 | 3 | 0.07 |
| | 0.2 | 300 | 4 | 2 | 1 | 3 | 0.14 |
| | 0.2 | 400 | 4 | 2 | 1 | 3 | 0.17 |
| | 0.2 | 500 | 4 | 2 | 1 | 3 | 0.00 |
| | 0.2 | 600 | 4 | 2 | 1 | 3 | −0.16 |
| | 0.2 | 700 | 4 | 2 | 1 | 3 | −0.44 |
| | 0.2 | 800 | 4 | 2 | 1 | 3 | −0.81 |
| | 0.2 | 900 | 4 | 2 | 1 | 3 | −0.80 |
| | 0.2 | 120 | 4 | 2 | 1 | 3 | 0.05 |
| | 0.2 | 200 | 4 | 2 | 1 | 3 | 0.06 |
| | 0.2 | 300 | 4 | 2 | 1 | 3 | 0.12 |
| | 0.2 | 400 | 4 | 2 | 1 | 3 | 0.15 |
| | 0.2 | 500 | 4 | 2 | 1 | 3 | 0.01 |
| | 0.2 | 600 | 4 | 2 | 1 | 3 | −0.20 |
| | 0.2 | 700 | 4 | 2 | 1 | 3 | −0.41 |
| | 0.2 | 800 | 4 | 2 | 1 | 3 | −0.72 |
| | 0.2 | 900 | 4 | 2 | 1 | 3 | −0.69 |
| | 0.2 | 120 | 4 | 2 | 1 | 3 | 0.06 |
| | 0.2 | 200 | 4 | 2 | 1 | 3 | 0.08 |

**Table A1.** *Cont.*

| REF. | Input Parameters | | | | | | Output Parameters |
|------|------|--------|-----------|--------|---|----------|-------|
| | W/B | T (°C) | V (°C/min) | MD (h) | C | RD (day) | P (%) |
| [39] | 0.2 | 300 | 4 | 2 | 1 | 3 | 0.13 |
| | 0.2 | 400 | 4 | 2 | 1 | 3 | 0.16 |
| | 0.2 | 500 | 4 | 2 | 1 | 3 | 0.09 |
| | 0.2 | 600 | 4 | 2 | 1 | 3 | −0.15 |
| | 0.2 | 700 | 4 | 2 | 1 | 3 | −0.40 |
| | 0.2 | 800 | 4 | 2 | 1 | 3 | −0.72 |
| | 0.2 | 900 | 4 | 2 | 1 | 3 | −0.68 |
| [40] | 0.28 | 200 | 5 | 0 | 0 | 0 | −30.95 |
| | 0.28 | 400 | 5 | 0 | 0 | 0 | −36.11 |
| | 0.41 | 200 | 5 | 0 | 0 | 0 | −11.29 |
| | 0.41 | 400 | 5 | 0 | 0 | 0 | −9.57 |
| | 0.41 | 200 | 5 | 0 | 0 | 0 | −9.23 |
| | 0.41 | 400 | 5 | 0 | 0 | 0 | −4.62 |
| | 0.41 | 600 | 5 | 0 | 0 | 0 | −33.46 |
| | 0.41 | 800 | 5 | 0 | 0 | 0 | −68.65 |
| | 0.64 | 200 | 5 | 0 | 0 | 0 | −10.41 |
| | 0.64 | 400 | 5 | 0 | 0 | 0 | −19.46 |
| | 0.64 | 600 | 5 | 0 | 0 | 0 | −35.75 |
| | 0.64 | 800 | 5 | 0 | 0 | 0 | −70.14 |
| | 0.64 | 200 | 5 | 0 | 0 | 0 | −16.97 |
| | 0.64 | 400 | 5 | 0 | 0 | 0 | −49.54 |
| | 0.64 | 600 | 5 | 0 | 0 | 0 | −66.51 |
| | 0.64 | 800 | 5 | 0 | 0 | 0 | −76.15 |
| [41] | 0.34 | 100 | 10 | 3 | 1 | 0 | 2.54 |
| | 0.34 | 200 | 9.09 | 3 | 1 | 0 | −11.38 |
| | 0.34 | 300 | 10.89 | 3 | 1 | 0 | −5.84 |
| | 0.34 | 400 | 8.7 | 3 | 1 | 0 | −14.82 |
| | 0.34 | 500 | 9.09 | 3 | 1 | 0 | −23.80 |
| | 0.34 | 600 | 8.82 | 3 | 1 | 0 | −40.42 |
| | 0.34 | 700 | 8.92 | 3 | 1 | 0 | −51.65 |
| | 0.34 | 800 | 8.73 | 3 | 1 | 0 | −67.81 |
| | 0.34 | 900 | 8.91 | 3 | 1 | 0 | −83.83 |
| | 0.3 | 100 | 10 | 3 | 1 | 0 | 10.74 |
| | 0.3 | 200 | 9.09 | 3 | 1 | 0 | −1.67 |
| | 0.3 | 300 | 10.89 | 3 | 1 | 0 | 8.95 |
| | 0.3 | 400 | 8.7 | 3 | 1 | 0 | −7.40 |
| | 0.3 | 600 | 8.82 | 3 | 1 | 0 | −52.03 |
| | 0.3 | 800 | 8.73 | 3 | 1 | 0 | −63.60 |
| | 0.3 | 900 | 8.91 | 3 | 1 | 0 | −76.37 |
| | 0.25 | 100 | 10 | 3 | 1 | 0 | −6.34 |
| | 0.25 | 200 | 9.09 | 3 | 1 | 0 | −9.09 |
| | 0.25 | 300 | 10.89 | 3 | 1 | 0 | −11.85 |
| | 0.25 | 400 | 8.7 | 3 | 1 | 0 | −9.09 |
| | 0.25 | 500 | 9.09 | 3 | 1 | 0 | −30.67 |
| | 0.25 | 600 | 8.82 | 3 | 1 | 0 | −41.23 |
| | 0.25 | 700 | 8.92 | 3 | 1 | 0 | −56.38 |
| | 0.34 | 100 | 10 | 3 | 1 | 0 | −0.15 |
| | 0.34 | 200 | 9.09 | 3 | 1 | 0 | −2.26 |
| | 0.34 | 300 | 10.89 | 3 | 1 | 0 | 3.77 |
| | 0.34 | 400 | 8.7 | 3 | 1 | 0 | −15.51 |
| | 0.34 | 500 | 9.09 | 3 | 1 | 0 | −22.59 |
| | 0.34 | 600 | 8.82 | 3 | 1 | 0 | −39.61 |
| | 0.34 | 700 | 8.92 | 3 | 1 | 0 | −50.75 |
| | 0.34 | 800 | 8.73 | 3 | 1 | 0 | −72.44 |
| | 0.34 | 900 | 8.91 | 3 | 1 | 0 | −81.33 |
| | 0.3 | 100 | 10 | 3 | 1 | 0 | 6.99 |

**Table A1.** *Cont.*

| REF. | Input Parameters | | | | | | Output Parameters |
|------|-----|--------|-----------|--------|---|----------|--------|
| | W/B | T (°C) | V (°C/min) | MD (h) | C | RD (day) | P (%) |
| | 0.3 | 200 | 9.09 | 3 | 1 | 0 | 10.29 |
| | 0.3 | 300 | 10.89 | 3 | 1 | 0 | 3.96 |
| | 0.3 | 400 | 8.7 | 3 | 1 | 0 | −4.88 |
| | 0.3 | 500 | 9.09 | 3 | 1 | 0 | −21.77 |
| | 0.3 | 600 | 8.82 | 3 | 1 | 0 | −22.96 |
| | 0.3 | 700 | 8.92 | 3 | 1 | 0 | −33.51 |
| | 0.3 | 800 | 8.73 | 3 | 1 | 0 | −61.48 |
| | 0.3 | 900 | 8.91 | 3 | 1 | 0 | −71.24 |
| [41] | 0.25 | 100 | 10 | 3 | 1 | 0 | 0.10 |
| | 0.25 | 200 | 9.09 | 3 | 1 | 0 | −7.86 |
| | 0.25 | 300 | 10.89 | 3 | 1 | 0 | −3.49 |
| | 0.25 | 400 | 8.7 | 3 | 1 | 0 | −0.29 |
| | 0.25 | 500 | 9.09 | 3 | 1 | 0 | −31.81 |
| | 0.25 | 600 | 8.82 | 3 | 1 | 0 | −46.56 |
| | 0.25 | 700 | 8.92 | 3 | 1 | 0 | −64.79 |
| | 0.25 | 800 | 8.73 | 3 | 1 | 0 | −74.01 |
| | 0.25 | 900 | 8.91 | 3 | 1 | 0 | −80.60 |
| | 0.5 | 400 | 2.5 | 3 | 1 | 7 | −1.77 |
| | 0.5 | 600 | 2.5 | 3 | 1 | 7 | −23.92 |
| | 0.5 | 800 | 2.5 | 3 | 1 | 7 | −69.93 |
| | 0.5 | 400 | 2.5 | 3 | 1 | 7 | 2.85 |
| | 0.5 | 600 | 2.5 | 3 | 1 | 7 | −11.71 |
| | 0.5 | 800 | 2.5 | 3 | 1 | 7 | −66.18 |
| | 0.5 | 400 | 2.5 | 3 | 1 | 7 | 17.12 |
| | 0.5 | 600 | 2.5 | 3 | 1 | 7 | −5.30 |
| | 0.5 | 800 | 2.5 | 3 | 1 | 7 | −57.75 |
| | 0.5 | 400 | 2.5 | 3 | 1 | 7 | 7.84 |
| | 0.5 | 600 | 2.5 | 3 | 1 | 7 | −4.73 |
| | 0.5 | 800 | 2.5 | 3 | 1 | 7 | −55.76 |
| | 0.5 | 400 | 2.5 | 3 | 1 | 7 | 12.74 |
| [42] | 0.5 | 600 | 2.5 | 3 | 1 | 7 | −5.75 |
| | 0.5 | 800 | 2.5 | 3 | 1 | 7 | −63.26 |
| | 0.5 | 400 | 2.5 | 3 | 1 | 7 | 20.40 |
| | 0.5 | 600 | 2.5 | 3 | 1 | 7 | −26.41 |
| | 0.5 | 800 | 2.5 | 3 | 1 | 7 | −76.98 |
| | 0.5 | 400 | 2.5 | 3 | 1 | 7 | 4.62 |
| | 0.5 | 600 | 2.5 | 3 | 1 | 7 | −19.48 |
| | 0.5 | 800 | 2.5 | 3 | 1 | 7 | −75.39 |
| | 0.5 | 400 | 2.5 | 3 | 1 | 7 | 2.86 |
| | 0.5 | 600 | 2.5 | 3 | 1 | 7 | −19.80 |
| | 0.5 | 800 | 2.5 | 3 | 1 | 7 | −77.81 |
| | 0.5 | 400 | 2.5 | 3 | 1 | 7 | 3.29 |
| | 0.5 | 600 | 2.5 | 3 | 1 | 7 | −25.48 |
| | 0.5 | 800 | 2.5 | 3 | 1 | 7 | −77.02 |
| | 0.53 | 50 | 16 | 3 | 0 | 0 | 9.06 |
| | 0.53 | 100 | 16 | 3 | 0 | 0 | −3.92 |
| | 0.53 | 150 | 16 | 3 | 0 | 0 | −5.34 |
| | 0.53 | 200 | 16 | 3 | 0 | 0 | −23.00 |
| | 0.53 | 250 | 16 | 3 | 0 | 0 | −26.59 |
| | 0.53 | 300 | 16 | 3 | 0 | 0 | −30.54 |
| [43] | 0.53 | 350 | 16 | 3 | 0 | 0 | −26.90 |
| | 0.53 | 400 | 16 | 3 | 0 | 0 | −57.57 |
| | 0.53 | 450 | 16 | 3 | 0 | 0 | −47.44 |
| | 0.53 | 500 | 16 | 3 | 0 | 0 | −55.71 |
| | 0.53 | 600 | 16 | 3 | 0 | 0 | −59.28 |
| | 0.53 | 700 | 16 | 3 | 0 | 0 | −67.17 |

**Table A1.** *Cont.*

| REF. | Input Parameters | | | | | | Output Parameters |
|---|---|---|---|---|---|---|---|
| | W/B | T (°C) | V (°C/min) | MD (h) | C | RD (day) | P (%) |
| [43] | 0.53 | 50 | 16 | 3 | 1 | 0 | 13.39 |
| | 0.53 | 100 | 16 | 3 | 1 | 0 | 20.27 |
| | 0.53 | 150 | 16 | 3 | 1 | 0 | −1.01 |
| | 0.53 | 200 | 16 | 3 | 1 | 0 | −16.14 |
| | 0.53 | 250 | 16 | 3 | 1 | 0 | −19.01 |
| | 0.53 | 300 | 16 | 3 | 1 | 0 | −23.68 |
| | 0.53 | 350 | 16 | 3 | 1 | 0 | −18.60 |
| | 0.53 | 400 | 16 | 3 | 1 | 0 | −30.49 |
| | 0.53 | 450 | 16 | 3 | 1 | 0 | −33.71 |
| | 0.53 | 500 | 16 | 3 | 1 | 0 | −48.13 |
| | 0.53 | 600 | 16 | 3 | 1 | 0 | −49.89 |
| | 0.53 | 700 | 16 | 3 | 1 | 0 | −61.39 |
| | 0.53 | 50 | 16 | 3 | 0 | 0 | −6.01 |
| | 0.53 | 100 | 16 | 3 | 0 | 0 | −13.34 |
| | 0.53 | 150 | 16 | 3 | 0 | 0 | −18.00 |
| | 0.53 | 200 | 16 | 3 | 0 | 0 | −20.66 |
| | 0.53 | 250 | 16 | 3 | 0 | 0 | −32.67 |
| | 0.53 | 300 | 16 | 3 | 0 | 0 | −21.27 |
| | 0.53 | 350 | 16 | 3 | 0 | 0 | −29.95 |
| | 0.53 | 400 | 16 | 3 | 0 | 0 | −42.63 |
| | 0.53 | 450 | 16 | 3 | 0 | 0 | −37.92 |
| | 0.53 | 500 | 16 | 3 | 0 | 0 | −45.59 |
| | 0.53 | 600 | 16 | 3 | 0 | 0 | −62.93 |
| | 0.53 | 700 | 16 | 3 | 0 | 0 | −68.23 |
| | 0.53 | 50 | 16 | 3 | 1 | 0 | −4.01 |
| | 0.53 | 100 | 16 | 3 | 1 | 0 | −9.33 |
| | 0.53 | 150 | 16 | 3 | 1 | 0 | −8.64 |
| | 0.53 | 200 | 16 | 3 | 1 | 0 | −17.98 |
| | 0.53 | 250 | 16 | 3 | 1 | 0 | −19.96 |
| | 0.53 | 300 | 16 | 3 | 1 | 0 | −16.26 |
| | 0.53 | 350 | 16 | 3 | 1 | 0 | −19.24 |
| | 0.53 | 400 | 16 | 3 | 1 | 0 | −20.56 |
| | 0.53 | 450 | 16 | 3 | 1 | 0 | −23.54 |
| | 0.53 | 500 | 16 | 3 | 1 | 0 | −31.88 |
| | 0.53 | 600 | 16 | 3 | 1 | 0 | −42.87 |
| | 0.53 | 700 | 16 | 3 | 1 | 0 | −54.52 |
| [44] | 0.43 | 100 | 1 | 0 | 1 | 0 | −4.72 |
| | 0.43 | 200 | 1 | 0 | 1 | 0 | −25.65 |
| | 0.43 | 400 | 1 | 0 | 1 | 0 | −28.44 |
| | 0.43 | 600 | 1 | 0 | 1 | 0 | −35.62 |
| | 0.43 | 800 | 1 | 0 | 1 | 0 | −56.00 |
| | 0.43 | 100 | 1 | 0 | 1 | 0 | −13.26 |
| | 0.43 | 200 | 1 | 0 | 1 | 0 | −20.74 |
| | 0.43 | 400 | 1 | 0 | 1 | 0 | −20.93 |
| | 0.43 | 600 | 1 | 0 | 1 | 0 | −33.55 |
| | 0.43 | 800 | 1 | 0 | 1 | 0 | −70.24 |
| | 0.43 | 100 | 1 | 0 | 1 | 0 | 1.75 |
| | 0.43 | 200 | 1 | 0 | 1 | 0 | −15.82 |
| | 0.43 | 400 | 1 | 0 | 1 | 0 | −15.24 |
| | 0.43 | 600 | 1 | 0 | 1 | 0 | −32.51 |
| | 0.43 | 800 | 1 | 0 | 1 | 0 | −69.72 |
| [45] | 0.3 | 200 | 5.5 | 0 | 1 | 0 | −7.41 |
| | 0.3 | 400 | 5.5 | 0 | 1 | 0 | −12.12 |
| | 0.3 | 600 | 5.5 | 0 | 1 | 0 | −26.94 |
| | 0.3 | 800 | 5.5 | 0 | 1 | 0 | −46.97 |
| | 0.3 | 1000 | 5.5 | 0 | 1 | 0 | −89.06 |

**Table A1.** *Cont.*

| REF. | Input Parameters | | | | | | Output Parameters |
|---|---|---|---|---|---|---|---|
| | W/B | T (°C) | V (°C/min) | MD (h) | C | RD (day) | P (%) |
| [45] | 0.3 | 200 | 5.5 | 0 | 0 | 0 | −27.27 |
| | 0.3 | 400 | 5.5 | 0 | 0 | 0 | −29.12 |
| | 0.3 | 600 | 5.5 | 0 | 0 | 0 | −43.43 |
| | 0.5 | 200 | 6.67 | 0 | 1 | 0 | −31.84 |
| | 0.5 | 400 | 6.67 | 0 | 1 | 0 | −23.15 |
| | 0.5 | 600 | 6.67 | 0 | 1 | 0 | −26.09 |
| | 0.5 | 800 | 6.67 | 0 | 1 | 0 | −51.47 |
| | 0.5 | 1000 | 6.67 | 0 | 1 | 0 | −75.32 |
| | 0.5 | 200 | 6.67 | 0 | 0 | 0 | −32.90 |
| | 0.5 | 400 | 6.67 | 0 | 0 | 0 | −29.38 |
| | 0.5 | 600 | 6.67 | 0 | 0 | 0 | −33.96 |
| | 0.5 | 800 | 6.67 | 0 | 0 | 0 | −55.82 |
| [46] | 0.43 | 110 | 0.1 | 0.5 | 1 | 0 | −0.82 |
| | 0.43 | 210 | 0.1 | 0.5 | 1 | 0 | −7.01 |
| | 0.43 | 310 | 0.1 | 0.5 | 1 | 0 | −34.65 |
| [47] | 0.5 | 200 | 10 | 2 | 1 | 0 | −16.93 |
| | 0.5 | 200 | 10 | 4 | 1 | 0 | −27.46 |
| | 0.5 | 200 | 10 | 6 | 1 | 0 | −34.93 |
| | 0.5 | 400 | 10 | 2 | 1 | 0 | −18.43 |
| | 0.5 | 400 | 10 | 4 | 1 | 0 | −27.09 |
| | 0.5 | 400 | 10 | 6 | 1 | 0 | −33.84 |
| | 0.5 | 600 | 10 | 2 | 1 | 0 | −35.75 |
| | 0.5 | 600 | 10 | 4 | 1 | 0 | −38.31 |
| | 0.5 | 600 | 10 | 6 | 1 | 0 | −41.34 |
| | 0.5 | 200 | 10 | 2 | 1 | 0 | −16.08 |
| | 0.5 | 200 | 10 | 4 | 1 | 0 | −16.08 |
| | 0.5 | 200 | 10 | 6 | 1 | 0 | −8.08 |
| | 0.5 | 400 | 10 | 2 | 1 | 0 | −9.86 |
| | 0.5 | 400 | 10 | 4 | 1 | 0 | −13.11 |
| | 0.5 | 400 | 10 | 6 | 1 | 0 | −17.52 |
| | 0.5 | 600 | 10 | 2 | 1 | 0 | −15.78 |
| | 0.5 | 600 | 10 | 4 | 1 | 0 | −20.41 |
| | 0.5 | 600 | 10 | 6 | 1 | 0 | −19.03 |
| | 0.5 | 200 | 10 | 2 | 1 | 0 | −25.15 |
| | 0.5 | 200 | 10 | 4 | 1 | 0 | −31.57 |
| | 0.5 | 200 | 10 | 6 | 1 | 0 | −28.94 |
| | 0.5 | 400 | 10 | 2 | 1 | 0 | −20.57 |
| | 0.5 | 400 | 10 | 4 | 1 | 0 | −25.76 |
| | 0.5 | 400 | 10 | 6 | 1 | 0 | −34.81 |
| | 0.5 | 600 | 10 | 2 | 1 | 0 | −36.31 |
| | 0.5 | 600 | 10 | 4 | 1 | 0 | −41.71 |
| | 0.5 | 600 | 10 | 6 | 1 | 0 | 43.26 |
| [48] | 0.5 | 200 | 16.98 | 0 | 1 | 0 | 20.74 |
| | 0.5 | 400 | 15.75 | 0 | 1 | 0 | −13.41 |
| | 0.5 | 600 | 12.7 | 0 | 1 | 0 | −45.12 |
| | 0.5 | 800 | 11.28 | 0 | 1 | 0 | −70.73 |
| | 0.5 | 200 | 16.98 | 0 | 0 | 0 | −1.62 |
| | 0.5 | 400 | 15.75 | 0 | 0 | 0 | −27.64 |
| | 0.5 | 600 | 12.7 | 0 | 0 | 0 | −59.35 |
| | 0.5 | 800 | 11.28 | 0 | 0 | 0 | −70.32 |
| | 0.5 | 200 | 16.98 | 0 | 0 | 0 | −13.00 |
| | 0.5 | 400 | 15.75 | 0 | 0 | 0 | −40.65 |
| | 0.5 | 600 | 12.7 | 0 | 0 | 0 | −69.51 |
| | 0.5 | 800 | 11.28 | 0 | 0 | 0 | −85.37 |
| | 0.5 | 200 | 16.98 | 0 | 0 | 0 | −15.04 |
| | 0.5 | 400 | 15.75 | 0 | 0 | 0 | −45.53 |

**Table A1.** *Cont.*

| REF. | Input Parameters | | | | | | Output Parameters |
|------|-----|-------|----------|--------|---|----------|--------|
| | W/B | T (°C) | V (°C/min) | MD (h) | C | RD (day) | P (%) |
| | 0.5 | 600 | 12.7 | 0 | 0 | 0 | −67.07 |
| | 0.5 | 800 | 11.28 | 0 | 0 | 0 | −80.89 |
| | 0.5 | 200 | 16.98 | 0 | 0 | 0 | −12.19 |
| | 0.5 | 400 | 15.75 | 0 | 0 | 0 | −33.74 |
| | 0.5 | 600 | 12.7 | 0 | 0 | 0 | −66.26 |
| | 0.5 | 800 | 11.28 | 0 | 0 | 0 | −82.11 |
| | 0.5 | 200 | 16.98 | 0 | 1 | 0 | 8.12 |
| | 0.5 | 400 | 15.75 | 0 | 1 | 0 | −21.40 |
| | 0.5 | 600 | 12.7 | 0 | 1 | 0 | −47.97 |
| | 0.5 | 800 | 11.28 | 0 | 1 | 0 | −72.32 |
| | 0.5 | 200 | 16.98 | 0 | 0 | 0 | 6.64 |
| | 0.5 | 400 | 15.75 | 0 | 0 | 0 | −27.31 |
| | 0.5 | 600 | 12.7 | 0 | 0 | 0 | −53.87 |
| | 0.5 | 800 | 11.28 | 0 | 0 | 0 | −67.53 |
| | 0.5 | 200 | 16.98 | 0 | 0 | 0 | −8.49 |
| | 0.5 | 400 | 15.75 | 0 | 0 | 0 | −42.07 |
| | 0.5 | 600 | 12.7 | 0 | 0 | 0 | −64.21 |
| | 0.5 | 800 | 11.28 | 0 | 0 | 0 | −92.99 |
| | 0.5 | 200 | 16.98 | 0 | 0 | 0 | −10.33 |
| | 0.5 | 400 | 15.75 | 0 | 0 | 0 | −41.33 |
| | 0.5 | 600 | 12.7 | 0 | 0 | 0 | −62.73 |
| | 0.5 | 200 | 16.98 | 0 | 0 | 0 | −6.64 |
| | 0.5 | 400 | 15.75 | 0 | 0 | 0 | −34.32 |
| | 0.5 | 600 | 12.7 | 0 | 0 | 0 | −61.62 |
| | 0.5 | 200 | 16.98 | 0 | 1 | 0 | 9.95 |
| | 0.5 | 400 | 15.75 | 0 | 1 | 0 | −10.38 |
| | 0.5 | 600 | 12.7 | 0 | 1 | 0 | −47.88 |
| | 0.5 | 800 | 11.28 | 0 | 1 | 0 | −84.65 |
| [48] | 0.5 | 200 | 16.98 | 0 | 0 | 0 | 7.10 |
| | 0.5 | 400 | 15.75 | 0 | 0 | 0 | −24.68 |
| | 0.5 | 600 | 12.7 | 0 | 0 | 0 | −60.01 |
| | 0.5 | 800 | 11.28 | 0 | 0 | 0 | −87.14 |
| | 0.5 | 200 | 16.98 | 0 | 0 | 0 | −11.46 |
| | 0.5 | 400 | 15.75 | 0 | 0 | 0 | −34.31 |
| | 0.5 | 600 | 12.7 | 0 | 0 | 0 | −69.66 |
| | 0.5 | 800 | 11.28 | 0 | 0 | 0 | −92.49 |
| | 0.5 | 200 | 16.98 | 0 | 0 | 0 | −13.60 |
| | 0.5 | 400 | 15.75 | 0 | 0 | 0 | −37.16 |
| | 0.5 | 600 | 12.7 | 0 | 0 | 0 | −67.15 |
| | 0.5 | 200 | 16.98 | 0 | 0 | 0 | −8.60 |
| | 0.5 | 400 | 15.75 | 0 | 0 | 0 | −35.37 |
| | 0.5 | 600 | 12.7 | 0 | 0 | 0 | −63.58 |
| | 0.5 | 200 | 16.98 | 0 | 1 | 0 | 31.91 |
| | 0.5 | 400 | 15.75 | 0 | 1 | 0 | −7.44 |
| | 0.5 | 600 | 12.7 | 0 | 1 | 0 | −38.87 |
| | 0.5 | 800 | 11.28 | 0 | 1 | 0 | −74.27 |
| | 0.5 | 200 | 16.98 | 0 | 0 | 0 | 3.20 |
| | 0.5 | 400 | 15.75 | 0 | 0 | 0 | −13.39 |
| | 0.5 | 600 | 12.7 | 0 | 0 | 0 | −48.78 |
| | 0.5 | 800 | 11.28 | 0 | 0 | 0 | −75.75 |
| | 0.5 | 200 | 16.98 | 0 | 0 | 0 | −7.20 |
| | 0.5 | 400 | 15.75 | 0 | 0 | 0 | −32.19 |
| | 0.5 | 600 | 12.7 | 0 | 0 | 0 | −64.61 |
| | 0.5 | 800 | 11.28 | 0 | 0 | 0 | −86.64 |
| | 0.5 | 200 | 16.98 | 0 | 0 | 0 | −4.72 |
| | 0.5 | 400 | 15.75 | 0 | 0 | 0 | −28.23 |
| | 0.5 | 600 | 12.7 | 0 | 0 | 0 | −60.65 |

**Table A1.** *Cont.*

| REF. | Input Parameters | | | | | | Output Parameters |
|---|---|---|---|---|---|---|---|
| | W/B | T (°C) | V (°C/min) | MD (h) | C | RD (day) | P (%) |
| | 0.5 | 800 | 11.28 | 0 | 0 | 0 | −85.16 |
| | 0.5 | 200 | 16.98 | 0 | 0 | 0 | −2.74 |
| | 0.5 | 400 | 15.75 | 0 | 0 | 0 | −21.80 |
| | 0.5 | 600 | 12.7 | 0 | 0 | 0 | −51.75 |
| | 0.5 | 800 | 11.28 | 0 | 0 | 0 | −84.17 |
| | 0.5 | 200 | 16.98 | 0 | 1 | 0 | 8.03 |
| | 0.5 | 400 | 15.75 | 0 | 1 | 0 | −23.89 |
| | 0.5 | 600 | 12.7 | 0 | 1 | 0 | −47.18 |
| | 0.5 | 800 | 11.28 | 0 | 1 | 0 | −75.93 |
| | 0.5 | 200 | 16.98 | 0 | 0 | 0 | −6.06 |
| | 0.5 | 400 | 15.75 | 0 | 0 | 0 | −42.98 |
| | 0.5 | 600 | 12.7 | 0 | 0 | 0 | −60.82 |
| [48] | 0.5 | 800 | 11.28 | 0 | 0 | 0 | −78.65 |
| | 0.5 | 200 | 16.98 | 0 | 0 | 0 | −12.87 |
| | 0.5 | 400 | 15.75 | 0 | 0 | 0 | −54.80 |
| | 0.5 | 600 | 12.7 | 0 | 0 | 0 | −71.27 |
| | 0.5 | 800 | 11.28 | 0 | 0 | 0 | −88.65 |
| | 0.5 | 200 | 16.98 | 0 | 0 | 0 | −9.69 |
| | 0.5 | 400 | 15.75 | 0 | 0 | 0 | −49.80 |
| | 0.5 | 600 | 12.7 | 0 | 0 | 0 | −67.63 |
| | 0.5 | 800 | 11.28 | 0 | 0 | 0 | −85.01 |
| | 0.5 | 200 | 16.98 | 0 | 0 | 0 | −7.87 |
| | 0.5 | 400 | 15.75 | 0 | 0 | 0 | −46.62 |
| | 0.5 | 600 | 12.7 | 0 | 0 | 0 | −64.91 |
| | 0.5 | 800 | 11.28 | 0 | 0 | 0 | −85.93 |
| | 0.5 | 200 | 20 | 2 | 1 | 0.08 | 7.58 |
| | 0.5 | 300 | 20 | 2 | 1 | 0.08 | −0.26 |
| | 0.5 | 400 | 20 | 2 | 1 | 0.08 | −3.18 |
| | 0.5 | 500 | 20 | 2 | 1 | 0.08 | −5.45 |
| | 0.5 | 600 | 20 | 2 | 1 | 0.08 | −8.70 |
| | 0.5 | 700 | 20 | 2 | 1 | 0.08 | −25.06 |
| | 0.5 | 800 | 20 | 2 | 1 | 0.08 | −71.92 |
| | 0.5 | 1000 | 20 | 2 | 1 | 0.08 | −86.94 |
| | 0.5 | 1200 | 20 | 2 | 1 | 0.08 | −93.12 |
| | 0.6 | 200 | 20 | 2 | 1 | 0.08 | −2.25 |
| | 0.6 | 400 | 20 | 2 | 1 | 0.08 | −0.56 |
| | 0.6 | 600 | 20 | 2 | 1 | 0.08 | −12.96 |
| | 0.6 | 700 | 20 | 2 | 1 | 0.08 | −27.69 |
| | 0.6 | 800 | 20 | 2 | 1 | 0.08 | −62.41 |
| | 0.6 | 1000 | 20 | 2 | 1 | 0.08 | −85.30 |
| [49] | 0.6 | 1200 | 20 | 2 | 1 | 0.08 | −92.13 |
| | 0.4 | 200 | 20 | 2 | 1 | 0.08 | −4.87 |
| | 0.4 | 400 | 20 | 2 | 1 | 0.08 | −15.97 |
| | 0.4 | 600 | 20 | 2 | 1 | 0.08 | −25.12 |
| | 0.4 | 700 | 20 | 2 | 1 | 0.08 | −49.68 |
| | 0.4 | 800 | 20 | 2 | 1 | 0.08 | −78.81 |
| | 0.4 | 1000 | 20 | 2 | 1 | 0.08 | −89.26 |
| | 0.4 | 1200 | 20 | 2 | 1 | 0.08 | −98.03 |
| | 0.5 | 200 | 20 | 2 | 1 | 0.08 | −0.61 |
| | 0.5 | 400 | 20 | 2 | 1 | 0.08 | −27.77 |
| | 0.5 | 500 | 20 | 2 | 1 | 0.08 | −34.62 |
| | 0.5 | 600 | 20 | 2 | 1 | 0.08 | −56.89 |
| | 0.5 | 700 | 20 | 2 | 1 | 0.08 | −66.69 |
| | 0.5 | 900 | 20 | 2 | 1 | 0.08 | −83.69 |
| | 0.5 | 1200 | 20 | 2 | 1 | 0.08 | −98.69 |

**Table A1.** *Cont.*

| REF. | Input Parameters | | | | | | Output Parameters |
|---|---|---|---|---|---|---|---|
| | **W/B** | **T (°C)** | **V (°C/min)** | **MD (h)** | **C** | **RD (day)** | **P (%)** |
| [50] | 0.41 | 150 | 5 | 5 | 1 | 0 | −18.56 |
| | 0.41 | 300 | 5 | 5 | 1 | 0 | −24.08 |
| | 0.41 | 400 | 5 | 5 | 1 | 0 | −35.49 |
| | 0.41 | 500 | 5 | 5 | 1 | 0 | −39.87 |
| | 0.41 | 600 | 5 | 5 | 1 | 0 | −60.80 |
| | 0.41 | 700 | 5 | 5 | 1 | 0 | −75.40 |
| | 0.41 | 150 | 5 | 5 | 1 | 0 | −17.85 |
| | 0.41 | 300 | 5 | 5 | 1 | 0 | −24.73 |
| | 0.41 | 400 | 5 | 5 | 1 | 0 | −34.56 |
| | 0.41 | 500 | 5 | 5 | 1 | 0 | −37.13 |
| | 0.41 | 600 | 5 | 5 | 1 | 0 | −61.01 |
| | 0.41 | 700 | 5 | 5 | 1 | 0 | −73.10 |
| | 0.41 | 150 | 5 | 5 | 1 | 0 | −18.27 |
| | 0.41 | 300 | 5 | 5 | 1 | 0 | −24.47 |
| | 0.41 | 400 | 5 | 5 | 1 | 0 | −36.56 |
| | 0.41 | 500 | 5 | 5 | 1 | 0 | −55.23 |
| | 0.41 | 600 | 5 | 5 | 1 | 0 | −66.19 |
| | 0.41 | 700 | 5 | 5 | 1 | 0 | −75.55 |
| | 0.41 | 150 | 5 | 5 | 1 | 0 | −18.70 |
| | 0.41 | 300 | 5 | 5 | 1 | 0 | −32.15 |
| | 0.41 | 400 | 5 | 5 | 1 | 0 | −49.91 |
| | 0.41 | 500 | 5 | 5 | 1 | 0 | −58.15 |
| | 0.41 | 600 | 5 | 5 | 1 | 0 | −66.16 |
| | 0.41 | 700 | 5 | 5 | 1 | 0 | −77.80 |
| [51] | 0.4 | 100 | 3 | 3 | 1 | 0 | −13.70 |
| | 0.4 | 200 | 3 | 3 | 1 | 0 | −10.20 |
| | 0.4 | 300 | 3 | 3 | 1 | 0 | −24.70 |
| | 0.4 | 600 | 3 | 3 | 1 | 0 | −66.60 |
| | 0.35 | 100 | 3 | 3 | 1 | 0 | −15.00 |
| | 0.35 | 200 | 3 | 3 | 1 | 0 | −12.50 |
| | 0.35 | 300 | 3 | 3 | 1 | 0 | −23.50 |
| | 0.35 | 600 | 3 | 3 | 1 | 0 | −70.50 |
| | 0.3 | 100 | 3 | 3 | 1 | 0 | −14.60 |
| | 0.3 | 200 | 3 | 3 | 1 | 0 | 11.40 |
| | 0.3 | 300 | 3 | 3 | 1 | 0 | −27.30 |
| | 0.3 | 600 | 3 | 3 | 1 | 0 | −68.80 |
| | 0.3 | 100 | 3 | 3 | 1 | 0 | −15.30 |
| | 0.3 | 200 | 3 | 3 | 1 | 0 | −14.10 |
| | 0.3 | 300 | 3 | 3 | 1 | 0 | −29.60 |
| | 0.3 | 600 | 3 | 3 | 1 | 0 | −70.90 |
| | 0.3 | 100 | 3 | 3 | 1 | 0 | −15.90 |
| | 0.3 | 200 | 3 | 3 | 1 | 0 | −14.80 |
| | 0.3 | 300 | 3 | 3 | 1 | 0 | −31.30 |
| | 0.3 | 600 | 3 | 3 | 1 | 0 | −73.20 |
| [52] | 0.42 | 105 | 3 | 16 | 1 | 0 | −17.71 |
| | 0.42 | 150 | 3 | 4 | 1 | 0 | −14.07 |
| | 0.42 | 150 | 3 | 8 | 1 | 0 | −9.83 |
| | 0.42 | 150 | 3 | 16 | 1 | 0 | −7.71 |
| | 0.42 | 200 | 3 | 4 | 1 | 0 | −0.54 |
| | 0.42 | 200 | 3 | 8 | 1 | 0 | −3.16 |
| | 0.42 | 200 | 3 | 16 | 1 | 0 | −5.05 |
| | 0.42 | 250 | 3 | 4 | 1 | 0 | −1.28 |
| | 0.42 | 250 | 3 | 8 | 1 | 0 | −4.25 |
| | 0.42 | 250 | 3 | 16 | 1 | 0 | −6.29 |
| | 0.42 | 300 | 3 | 4 | 1 | 0 | −4.06 |
| | 0.42 | 300 | 3 | 8 | 1 | 0 | −7.13 |

**Table A1.** *Cont.*

| REF. | Input Parameters | | | | | | Output Parameters |
|---|---|---|---|---|---|---|---|
| | W/B | T (°C) | V (°C/min) | MD (h) | C | RD (day) | P (%) |
| [52] | 0.42 | 300 | 3 | 16 | 1 | 0 | −9.21 |
| | 0.42 | 350 | 3 | 4 | 1 | 0 | −7.34 |
| | 0.42 | 350 | 3 | 8 | 1 | 0 | −10.38 |
| | 0.42 | 350 | 3 | 16 | 1 | 0 | −12.33 |
| | 0.42 | 400 | 3 | 4 | 1 | 0 | −11.56 |
| | 0.42 | 400 | 3 | 8 | 1 | 0 | −14.15 |
| | 0.42 | 400 | 3 | 16 | 1 | 0 | −15.86 |
| | 0.42 | 450 | 3 | 4 | 1 | 0 | −16.00 |
| | 0.42 | 450 | 3 | 8 | 1 | 0 | −18.37 |
| | 0.42 | 450 | 3 | 16 | 1 | 0 | −19.86 |
| [53] | 0.54 | 100 | 15.6 | 0 | 1 | 0 | −3.65 |
| | 0.54 | 300 | 15.6 | 0 | 1 | 0 | −21.40 |
| | 0.54 | 500 | 15.6 | 0 | 1 | 0 | −29.07 |
| | 0.54 | 800 | 15.6 | 0 | 1 | 0 | −37.90 |
| | 0.54 | 100 | 15.6 | 0 | 1 | 0 | −5.37 |
| | 0.54 | 300 | 15.6 | 0 | 1 | 0 | −22.55 |
| | 0.54 | 500 | 15.6 | 0 | 1 | 0 | −36.69 |
| | 0.54 | 800 | 15.6 | 0 | 1 | 0 | −50.67 |
| | 0.54 | 100 | 15.6 | 0 | 1 | 0 | −8.42 |
| | 0.54 | 300 | 15.6 | 0 | 1 | 0 | −23.88 |
| | 0.54 | 500 | 15.6 | 0 | 1 | 0 | −37.07 |
| | 0.54 | 800 | 15.6 | 0 | 1 | 0 | −55.24 |
| | 0.54 | 100 | 15.6 | 0 | 1 | 0 | −9.94 |
| | 0.54 | 300 | 15.6 | 0 | 1 | 0 | −23.88 |
| | 0.54 | 500 | 15.6 | 0 | 1 | 0 | −38.21 |
| | 0.54 | 800 | 15.6 | 0 | 1 | 0 | −58.09 |
| [54] | 0.25 | 200 | 6.67 | 2 | 0 | 0 | 21.61 |
| | 0.25 | 400 | 6.67 | 2 | 0 | 0 | 1.26 |
| | 0.25 | 600 | 6.67 | 2 | 0 | 0 | −29.58 |
| | 0.25 | 800 | 6.67 | 2 | 0 | 0 | −70.33 |
| | 0.25 | 1000 | 6.67 | 2 | 0 | 0 | −88.92 |
| [55] | 0.26 | 400 | 10 | 1 | 1 | 2 | −0.48 |
| | 0.26 | 600 | 10 | 1 | 1 | 2 | −21.53 |
| | 0.26 | 800 | 10 | 1 | 1 | 2 | −70.56 |
| [56] | 0.4 | 200 | 10 | 3 | 1 | 28 | 4.29 |
| | 0.4 | 400 | 10 | 3 | 1 | 28 | −15.71 |
| | 0.4 | 600 | 10 | 3 | 1 | 28 | −24.29 |
| | 0.4 | 800 | 10 | 3 | 1 | 28 | −52.86 |
| | 0.4 | 1000 | 10 | 3 | 1 | 28 | −61.43 |
| | 0.4 | 200 | 10 | 3 | 0 | 28 | −1.43 |
| | 0.4 | 400 | 10 | 3 | 0 | 28 | −24.29 |
| | 0.4 | 600 | 10 | 3 | 0 | 28 | −34.29 |
| | 0.4 | 800 | 10 | 3 | 0 | 28 | −57.14 |
| | 0.4 | 1000 | 10 | 3 | 0 | 28 | −74.29 |
| [57] | 0.62 | 150 | 1 | 0 | 1 | 0 | −2.55 |
| | 0.62 | 300 | 1 | 0 | 1 | 0 | −5.71 |
| | 0.62 | 450 | 1 | 0 | 1 | 0 | −42.01 |
| | 0.62 | 600 | 1 | 0 | 1 | 0 | −90.96 |
| | 0.55 | 150 | 1 | 0 | 1 | 0 | −8.44 |
| | 0.55 | 300 | 1 | 0 | 1 | 0 | −0.76 |
| | 0.55 | 450 | 1 | 0 | 1 | 0 | −50.38 |
| | 0.55 | 600 | 1 | 0 | 1 | 0 | −83.85 |
| | 0.44 | 150 | 1 | 0 | 1 | 0 | −9.39 |
| | 0.44 | 300 | 1 | 0 | 1 | 0 | 2.45 |
| | 0.44 | 450 | 1 | 0 | 1 | 0 | −58.29 |

**Table A1.** *Cont.*

| REF. | Input Parameters | | | | | | Output Parameters |
|---|---|---|---|---|---|---|---|
| | W/B | T (°C) | V (°C/min) | MD (h) | C | RD (day) | P (%) |
| [57] | 0.44 | 600 | 1 | 0 | 1 | 0 | −87.17 |
| | 0.36 | 150 | 1 | 0 | 1 | 0 | 4.66 |
| | 0.36 | 300 | 1 | 0 | 1 | 0 | 11.67 |
| | 0.36 | 450 | 1 | 0 | 1 | 0 | −47.72 |
| | 0.36 | 600 | 1 | 0 | 1 | 0 | −84.58 |
| | 0.29 | 150 | 1 | 0 | 1 | 0 | −1.72 |
| | 0.29 | 300 | 1 | 0 | 1 | 0 | 7.41 |
| | 0.29 | 350 | 1 | 0 | 1 | 0 | 20.54 |
| | 0.29 | 600 | 1 | 0 | 1 | 0 | −84.87 |
| [58] | 0.33 | 200 | 1 | 3 | 1 | 0 | 3.86 |
| | 0.33 | 400 | 1 | 3 | 1 | 0 | −17.66 |
| | 0.33 | 600 | 1 | 3 | 1 | 0 | −49.04 |
| | 0.33 | 800 | 1 | 3 | 1 | 0 | −75.02 |
| | 0.33 | 200 | 1 | 3 | 1 | 0 | 3.21 |
| | 0.33 | 400 | 1 | 3 | 1 | 0 | −15.91 |
| | 0.33 | 600 | 1 | 3 | 1 | 0 | −49.41 |
| | 0.33 | 800 | 1 | 3 | 1 | 0 | −76.35 |
| | 0.33 | 200 | 1 | 3 | 1 | 0 | −2.29 |
| | 0.33 | 400 | 1 | 3 | 1 | 0 | −17.49 |
| | 0.33 | 600 | 1 | 3 | 1 | 0 | −53.97 |
| | 0.33 | 800 | 1 | 3 | 1 | 0 | −76.66 |
| | 0.33 | 200 | 1 | 3 | 1 | 0 | −2.18 |
| | 0.33 | 400 | 1 | 3 | 1 | 0 | −16.53 |
| | 0.33 | 600 | 1 | 3 | 1 | 0 | −54.68 |
| | 0.33 | 800 | 1 | 3 | 1 | 0 | −78.94 |
| | 0.33 | 200 | 1 | 3 | 1 | 0 | −0.53 |
| | 0.33 | 400 | 1 | 3 | 1 | 0 | −18.59 |
| | 0.33 | 600 | 1 | 3 | 1 | 0 | −57.16 |
| | 0.33 | 800 | 1 | 3 | 1 | 0 | −77.98 |
| | 0.33 | 200 | 1 | 3 | 1 | 0 | 3.58 |
| | 0.33 | 400 | 1 | 3 | 1 | 0 | −19.79 |
| | 0.33 | 600 | 1 | 3 | 1 | 0 | −57.00 |
| | 0.33 | 800 | 1 | 3 | 1 | 0 | −80.14 |
| | 0.33 | 200 | 1 | 3 | 1 | 0 | −5.91 |
| | 0.33 | 400 | 1 | 3 | 1 | 0 | −20.10 |
| | 0.33 | 600 | 1 | 3 | 1 | 0 | −54.58 |
| | 0.33 | 800 | 1 | 3 | 1 | 0 | −81.95 |
| | 0.33 | 200 | 1 | 3 | 1 | 0 | 1.91 |
| | 0.33 | 400 | 1 | 3 | 1 | 0 | −23.97 |
| | 0.33 | 600 | 1 | 3 | 1 | 0 | −50.66 |
| | 0.33 | 800 | 1 | 3 | 1 | 0 | −76.39 |
| | 0.33 | 200 | 1 | 3 | 1 | 0 | −1.37 |
| | 0.33 | 400 | 1 | 3 | 1 | 0 | −16.77 |
| | 0.33 | 600 | 1 | 3 | 1 | 0 | −52.98 |
| | 0.33 | 800 | 1 | 3 | 1 | 0 | −77.25 |
| | 0.33 | 200 | 1 | 3 | 1 | 0 | −1.61 |
| | 0.33 | 400 | 1 | 3 | 1 | 0 | −21.09 |
| | 0.33 | 600 | 1 | 3 | 1 | 0 | −48.67 |
| | 0.33 | 800 | 1 | 3 | 1 | 0 | −78.99 |
| | 0.33 | 200 | 1 | 3 | 1 | 0 | −3.44 |
| | 0.33 | 400 | 1 | 3 | 1 | 0 | −22.70 |
| | 0.33 | 600 | 1 | 3 | 1 | 0 | −51.69 |
| | 0.33 | 800 | 1 | 3 | 1 | 0 | −80.00 |
| | 0.33 | 200 | 1 | 3 | 1 | 0 | 1.10 |
| | 0.33 | 400 | 1 | 3 | 1 | 0 | −18.71 |
| | 0.33 | 600 | 1 | 3 | 1 | 0 | −49.54 |
| | 0.33 | 800 | 1 | 3 | 1 | 0 | −76.41 |

Table A1. *Cont.*

| REF. | Input Parameters | | | | | | Output Parameters |
|------|-----|--------|----------|--------|---|----------|--------|
| | W/B | T (°C) | V (°C/min) | MD (h) | C | RD (day) | P (%) |
| [58] | 0.33 | 200 | 1 | 3 | 1 | 0 | 0.17 |
| | 0.33 | 400 | 1 | 3 | 1 | 0 | −15.91 |
| | 0.33 | 600 | 1 | 3 | 1 | 0 | −48.17 |
| | 0.33 | 800 | 1 | 3 | 1 | 0 | −75.55 |
| | 0.33 | 200 | 1 | 3 | 1 | 0 | 3.66 |
| | 0.33 | 400 | 1 | 3 | 1 | 0 | −18.71 |
| | 0.33 | 600 | 1 | 3 | 1 | 0 | −47.05 |
| | 0.33 | 800 | 1 | 3 | 1 | 0 | −77.40 |
| | 0.33 | 200 | 1 | 3 | 1 | 0 | −4.54 |
| | 0.33 | 400 | 1 | 3 | 1 | 0 | −21.01 |
| | 0.33 | 600 | 1 | 3 | 1 | 0 | −50.58 |
| | 0.33 | 800 | 1 | 3 | 1 | 0 | −78.00 |
| | 0.33 | 200 | 1 | 3 | 1 | 0 | −7.56 |
| | 0.33 | 400 | 1 | 3 | 1 | 0 | −21.89 |
| | 0.33 | 600 | 1 | 3 | 1 | 0 | −48.55 |
| | 0.33 | 800 | 1 | 3 | 1 | 0 | −78.97 |
| | 0.33 | 200 | 1 | 3 | 1 | 0 | −8.06 |
| | 0.33 | 400 | 1 | 3 | 1 | 0 | −20.73 |
| | 0.33 | 600 | 1 | 3 | 1 | 0 | −47.87 |
| | 0.33 | 800 | 1 | 3 | 1 | 0 | −80.42 |
| [59] | 0.56 | 105 | 3 | 16 | 1 | 0 | −17.73 |
| | 0.56 | 150 | 3 | 16 | 1 | 0 | −7.75 |
| | 0.56 | 200 | 3 | 16 | 1 | 0 | −5.07 |
| | 0.56 | 250 | 3 | 16 | 1 | 0 | −6.26 |
| | 0.56 | 300 | 3 | 16 | 1 | 0 | −9.24 |
| | 0.56 | 350 | 3 | 16 | 1 | 0 | −12.37 |
| | 0.56 | 400 | 3 | 16 | 1 | 0 | −15.95 |
| | 0.56 | 450 | 3 | 16 | 1 | 0 | −19.82 |
| [60] | 0.3 | 200 | 2.5 | 1 | 1 | 0 | −3.90 |
| | 0.3 | 400 | 2.5 | 1 | 1 | 0 | −12.78 |
| | 0.3 | 600 | 2.5 | 1 | 1 | 0 | −42.78 |
| | 0.3 | 800 | 2.5 | 1 | 1 | 0 | −76.59 |
| | 0.3 | 200 | 2.5 | 1 | 1 | 0 | 6.36 |
| | 0.3 | 400 | 2.5 | 1 | 1 | 0 | −11.03 |
| | 0.3 | 600 | 2.5 | 1 | 1 | 0 | −42.83 |
| | 0.3 | 800 | 2.5 | 1 | 1 | 0 | −75.86 |
| | 0.3 | 200 | 2.5 | 1 | 1 | 0 | 10.81 |
| | 0.3 | 400 | 2.5 | 1 | 1 | 0 | −7.33 |
| | 0.3 | 600 | 2.5 | 1 | 1 | 0 | −39.31 |
| | 0.3 | 800 | 2.5 | 1 | 1 | 0 | −73.22 |
| | 0.3 | 200 | 2.5 | 1 | 1 | 0 | 14.90 |
| | 0.3 | 400 | 2.5 | 1 | 1 | 0 | −6.87 |
| | 0.3 | 600 | 2.5 | 1 | 1 | 0 | −35.44 |
| | 0.3 | 800 | 2.5 | 1 | 1 | 0 | −69.50 |
| | 0.3 | 200 | 2.5 | 1 | 1 | 0 | 19.08 |
| | 0.3 | 400 | 2.5 | 1 | 1 | 0 | 0.14 |
| | 0.3 | 600 | 2.5 | 1 | 1 | 0 | −44.28 |
| | 0.3 | 800 | 2.5 | 1 | 1 | 0 | −71.78 |
| [61] | 0.31 | 200 | 10 | 3 | 1 | 0 | −11.55 |
| | 0.31 | 400 | 10 | 3 | 1 | 0 | 26.27 |
| | 0.31 | 600 | 10 | 3 | 1 | 0 | −5.35 |
| | 0.31 | 800 | 10 | 3 | 1 | 0 | −59.94 |
| | 0.31 | 200 | 10 | 3 | 1 | 0 | 51.47 |
| | 0.31 | 400 | 10 | 3 | 1 | 0 | 66.88 |
| | 0.31 | 600 | 10 | 3 | 1 | 0 | 33.89 |
| | 0.31 | 800 | 10 | 3 | 1 | 0 | −42.68 |

| REF. | Input Parameters | | | | | | Output Parameters |
|---|---|---|---|---|---|---|---|
| | W/B | T (°C) | V (°C/min) | MD (h) | C | RD (day) | P (%) |
| [62] | 0.55 | 400 | 7.5 | 1 | 1 | 4 | −18.89 |
| | 0.55 | 600 | 7.5 | 1 | 1 | 4 | −20.94 |
| | 0.55 | 800 | 7.5 | 1 | 1 | 4 | −50.72 |
| | 0.55 | 400 | 7.5 | 1 | 1 | 4 | −17.03 |
| | 0.55 | 600 | 7.5 | 1 | 1 | 4 | −21.62 |
| | 0.55 | 800 | 7.5 | 1 | 1 | 4 | −59.19 |
| | 0.55 | 400 | 7.5 | 1 | 1 | 4 | −14.33 |
| | 0.55 | 600 | 7.5 | 1 | 1 | 4 | −21.50 |
| | 0.55 | 800 | 7.5 | 1 | 1 | 4 | −63.48 |
| | 0.55 | 400 | 7.5 | 1 | 1 | 4 | −19.55 |
| | 0.55 | 600 | 7.5 | 1 | 1 | 4 | −27.73 |
| | 0.55 | 800 | 7.5 | 1 | 1 | 4 | −66.82 |
| [63] | 0.5 | 100 | 6.5 | 1 | 1 | 0 | −6.45 |
| | 0.5 | 200 | 6.5 | 1 | 1 | 0 | −1.80 |
| | 0.5 | 300 | 6.5 | 1 | 1 | 0 | −12.15 |
| | 0.5 | 100 | 6.5 | 1 | 1 | 0 | −19.43 |
| | 0.5 | 200 | 6.5 | 1 | 1 | 0 | −2.69 |
| | 0.5 | 300 | 6.5 | 1 | 1 | 0 | −17.59 |
| | 0.5 | 400 | 6.5 | 1 | 1 | 0 | −32.77 |
| | 0.5 | 500 | 6.5 | 1 | 1 | 0 | −37.51 |
| | 0.5 | 600 | 6.5 | 1 | 1 | 0 | −79.81 |
| | 0.5 | 700 | 6.5 | 1 | 1 | 0 | −63.91 |
| | 0.5 | 800 | 5 | 1 | 1 | 0 | −69.21 |
| | 0.5 | 100 | 6.5 | 1 | 1 | 0 | −14.36 |
| | 0.5 | 200 | 6.5 | 1 | 1 | 0 | −0.17 |
| | 0.5 | 300 | 6.5 | 1 | 1 | 0 | −11.14 |
| | 0.5 | 400 | 6.5 | 1 | 1 | 0 | −21.18 |
| | 0.5 | 500 | 6.5 | 1 | 1 | 0 | −37.36 |
| | 0.5 | 600 | 6.5 | 1 | 1 | 0 | −55.38 |
| | 0.5 | 700 | 6.5 | 1 | 1 | 0 | −59.90 |
| | 0.5 | 800 | 5 | 1 | 1 | 0 | −68.10 |
| | 0.5 | 100 | 6.5 | 1 | 1 | 0 | 10.56 |
| | 0.5 | 200 | 6.5 | 1 | 1 | 0 | 0.19 |
| | 0.5 | 300 | 6.5 | 1 | 1 | 0 | 5.54 |
| | 0.5 | 400 | 6.5 | 1 | 1 | 0 | 12.19 |
| | 0.5 | 500 | 6.5 | 1 | 1 | 0 | −27.86 |
| | 0.5 | 600 | 6.5 | 1 | 1 | 0 | −42.16 |
| | 0.5 | 700 | 6.5 | 1 | 1 | 0 | −50.78 |
| | 0.5 | 800 | 5 | 1 | 1 | 0 | −60.71 |
| [64] | 0.3 | 200 | 2.5 | 1 | 1 | 0 | −3.92 |
| | 0.3 | 400 | 2.5 | 1 | 1 | 0 | −10.50 |
| | 0.3 | 600 | 2.5 | 1 | 1 | 0 | −41.64 |
| | 0.3 | 800 | 2.5 | 1 | 1 | 0 | −77.24 |
| | 0.3 | 200 | 2.5 | 1 | 1 | 0 | 0.09 |
| | 0.3 | 400 | 2.5 | 1 | 1 | 0 | −6.35 |
| | 0.3 | 600 | 2.5 | 1 | 1 | 0 | −48.18 |
| | 0.3 | 800 | 2.5 | 1 | 1 | 0 | −78.08 |
| | 0.3 | 200 | 2.5 | 1 | 1 | 0 | −0.25 |
| | 0.3 | 400 | 2.5 | 1 | 1 | 0 | −1.41 |
| | 0.3 | 600 | 2.5 | 1 | 1 | 0 | −5.89 |
| | 0.3 | 800 | 2.5 | 1 | 1 | 0 | −8.73 |
| | 0.3 | 200 | 2.5 | 1 | 1 | 0 | 2.88 |
| | 0.3 | 400 | 2.5 | 1 | 1 | 0 | −6.47 |
| | 0.3 | 600 | 2.5 | 1 | 1 | 0 | −53.59 |
| | 0.3 | 800 | 2.5 | 1 | 1 | 0 | −80.77 |
| | 0.3 | 200 | 2.5 | 1 | 1 | 0 | 7.12 |

**Table A1.** *Cont.*

| REF. | Input Parameters | | | | | | Output Parameters |
|------|-----|-------|----------|--------|---|----------|-------|
| | W/B | T (°C) | V (°C/min) | MD (h) | C | RD (day) | P (%) |
| [64] | 0.3 | 400 | 2.5 | 1 | 1 | 0 | −9.14 |
| | 0.3 | 600 | 2.5 | 1 | 1 | 0 | −62.59 |
| | 0.3 | 800 | 2.5 | 1 | 1 | 0 | −86.22 |
| | 0.3 | 200 | 2.5 | 1 | 1 | 0 | 5.20 |
| | 0.3 | 400 | 2.5 | 1 | 1 | 0 | −13.45 |
| | 0.3 | 600 | 2.5 | 1 | 1 | 0 | −68.06 |
| | 0.3 | 800 | 2.5 | 1 | 1 | 0 | −91.24 |
| | 0.3 | 200 | 2.5 | 1 | 1 | 0 | 14.50 |
| | 0.3 | 400 | 2.5 | 1 | 1 | 0 | −3.19 |
| | 0.3 | 600 | 2.5 | 1 | 1 | 0 | −37.83 |
| | 0.3 | 800 | 2.5 | 1 | 1 | 0 | −71.62 |
| | 0.3 | 200 | 2.5 | 1 | 1 | 0 | −5.69 |
| | 0.3 | 400 | 2.5 | 1 | 1 | 0 | −26.14 |
| | 0.3 | 600 | 2.5 | 1 | 1 | 0 | −67.05 |
| | 0.3 | 800 | 2.5 | 1 | 1 | 0 | −92.05 |
| | 0.3 | 200 | 2.5 | 1 | 1 | 0 | −1.92 |
| | 0.3 | 400 | 2.5 | 1 | 1 | 0 | −9.61 |
| | 0.3 | 600 | 2.5 | 1 | 1 | 0 | −67.31 |
| | 0.3 | 800 | 2.5 | 1 | 1 | 0 | −87.50 |
| | 0.3 | 200 | 2.5 | 1 | 1 | 0 | 1.67 |
| | 0.3 | 400 | 2.5 | 1 | 1 | 0 | −14.99 |
| | 0.3 | 600 | 2.5 | 1 | 1 | 0 | −71.66 |
| | 0.3 | 800 | 2.5 | 1 | 1 | 0 | −90.00 |
| | 0.3 | 200 | 2.5 | 1 | 1 | 0 | 3.34 |
| | 0.3 | 400 | 2.5 | 1 | 1 | 0 | −21.33 |
| | 0.3 | 600 | 2.5 | 1 | 1 | 0 | −78.00 |
| | 0.3 | 800 | 2.5 | 1 | 1 | 0 | −92.67 |
| | 0.3 | 200 | 2.5 | 1 | 1 | 0 | 0.91 |
| | 0.3 | 400 | 2.5 | 1 | 1 | 0 | −18.18 |
| | 0.3 | 600 | 2.5 | 1 | 1 | 0 | −64.54 |
| | 0.3 | 800 | 2.5 | 1 | 1 | 0 | −84.55 |
| [65] | 0.5 | 200 | 1 | 1 | 1 | 0 | 8.04 |
| | 0.5 | 400 | 1 | 1 | 1 | 0 | −15.32 |
| | 0.5 | 600 | 1 | 1 | 1 | 0 | −48.90 |
| | 0.5 | 800 | 1 | 1 | 1 | 0 | −69.34 |
| | 0.5 | 200 | 1 | 1 | 1 | 0 | 12.83 |
| | 0.5 | 400 | 1 | 1 | 1 | 0 | −14.87 |
| | 0.5 | 600 | 1 | 1 | 1 | 0 | −46.62 |
| | 0.5 | 800 | 1 | 1 | 1 | 0 | −69.60 |
| | 0.5 | 200 | 1 | 1 | 1 | 0 | 14.91 |
| | 0.5 | 400 | 1 | 1 | 1 | 0 | 1.07 |
| | 0.5 | 600 | 1 | 1 | 1 | 0 | −32.97 |
| | 0.5 | 800 | 1 | 1 | 1 | 0 | −76.59 |
| | 0.3 | 200 | 1 | 1 | 1 | 0 | 7.74 |
| | 0.3 | 400 | 1 | 1 | 1 | 0 | −19.36 |
| | 0.3 | 600 | 1 | 1 | 1 | 0 | −47.42 |
| | 0.3 | 800 | 1 | 1 | 1 | 0 | −77.74 |
| | 0.3 | 200 | 1 | 1 | 1 | 0 | 14.49 |
| | 0.3 | 400 | 1 | 1 | 1 | 0 | −14.84 |
| | 0.3 | 600 | 1 | 1 | 1 | 0 | −44.17 |
| | 0.3 | 800 | 1 | 1 | 1 | 0 | −76.33 |
| | 0.3 | 200 | 1 | 1 | 1 | 0 | 23.76 |
| | 0.3 | 400 | 1 | 1 | 1 | 0 | −0.99 |
| | 0.3 | 600 | 1 | 1 | 1 | 0 | −34.16 |
| | 0.3 | 800 | 1 | 1 | 1 | 0 | −74.75 |
| [66] | 0.5 | 100 | 10 | 1 | 1 | 0 | −10.42 |
| | 0.5 | 300 | 10 | 1 | 1 | 0 | −22.66 |

**Table A1.** *Cont.*

| REF. | Input Parameters | | | | | | Output Parameters |
|---|---|---|---|---|---|---|---|
| | W/B | T (°C) | V (°C/min) | MD (h) | C | RD (day) | P (%) |
| [66] | 0.5 | 500 | 10 | 1 | 1 | 0 | −33.41 |
| | 0.5 | 800 | 10 | 1 | 1 | 0 | −78.34 |
| | 0.5 | 100 | 10 | 1 | 0 | 0 | −11.30 |
| | 0.5 | 300 | 10 | 1 | 0 | 0 | −35.71 |
| | 0.5 | 500 | 10 | 1 | 0 | 0 | −44.69 |
| | 0.5 | 800 | 10 | 1 | 0 | 0 | −76.85 |
| | 0.35 | 100 | 10 | 1 | 1 | 0 | −7.57 |
| | 0.35 | 300 | 10 | 1 | 1 | 0 | −3.72 |
| | 0.35 | 500 | 10 | 1 | 1 | 0 | 5.38 |
| | 0.35 | 800 | 10 | 1 | 1 | 0 | −52.45 |
| | 0.35 | 100 | 10 | 1 | 0 | 0 | −7.57 |
| | 0.35 | 300 | 10 | 1 | 0 | 0 | −14.21 |
| | 0.35 | 500 | 10 | 1 | 0 | 0 | −28.07 |
| | 0.35 | 800 | 10 | 1 | 0 | 0 | −64.58 |
| [67] | 0.6 | 100 | 1.5 | 2 | 1 | 0 | −0.11 |
| | 0.6 | 300 | 1.5 | 2 | 1 | 0 | 0.05 |
| | 0.6 | 600 | 1.5 | 2 | 1 | 0 | −0.52 |
| | 0.6 | 750 | 1.5 | 2 | 1 | 0 | −0.75 |
| | 0.6 | 100 | 1.5 | 2 | 1 | 0 | −0.05 |
| | 0.6 | 300 | 1.5 | 2 | 1 | 0 | 0.06 |
| | 0.6 | 600 | 1.5 | 2 | 1 | 0 | −0.50 |
| | 0.6 | 750 | 1.5 | 2 | 1 | 0 | −0.75 |
| | 0.6 | 100 | 1.5 | 2 | 1 | 0 | 0.19 |
| | 0.6 | 300 | 1.5 | 2 | 1 | 0 | 0.32 |
| | 0.6 | 600 | 1.5 | 2 | 1 | 0 | −0.51 |
| | 0.6 | 750 | 1.5 | 2 | 1 | 0 | −0.78 |
| | 0.6 | 100 | 1.5 | 2 | 1 | 0 | 0.13 |
| | 0.6 | 300 | 1.5 | 2 | 1 | 0 | 0.37 |
| | 0.6 | 600 | 1.5 | 2 | 1 | 0 | −0.60 |
| | 0.6 | 750 | 1.5 | 2 | 1 | 0 | −0.85 |
| | 0.6 | 100 | 1.5 | 2 | 1 | 0 | −0.03 |
| | 0.6 | 300 | 1.5 | 2 | 1 | 0 | 0.16 |
| | 0.6 | 600 | 1.5 | 2 | 1 | 0 | −0.68 |
| | 0.6 | 750 | 1.5 | 2 | 1 | 0 | −0.88 |
| | 0.6 | 100 | 1.5 | 2 | 1 | 0 | 0.13 |
| | 0.6 | 300 | 1.5 | 2 | 1 | 0 | 0.39 |
| | 0.6 | 600 | 1.5 | 2 | 1 | 0 | −0.55 |
| | 0.6 | 750 | 1.5 | 2 | 1 | 0 | −0.93 |
| | 0.6 | 100 | 1.5 | 2 | 1 | 0 | 0.24 |
| | 0.6 | 300 | 1.5 | 2 | 1 | 0 | 0.24 |
| | 0.6 | 600 | 1.5 | 2 | 1 | 0 | −0.53 |
| | 0.6 | 750 | 1.5 | 2 | 1 | 0 | −0.75 |
| | 0.6 | 100 | 1.5 | 2 | 1 | 0 | −0.01 |
| | 0.6 | 300 | 1.5 | 2 | 1 | 0 | −0.07 |
| | 0.6 | 600 | 1.5 | 2 | 1 | 0 | −0.48 |
| | 0.6 | 750 | 1.5 | 2 | 1 | 0 | −0.77 |
| | 0.6 | 100 | 1.5 | 2 | 1 | 0 | −0.09 |
| | 0.6 | 300 | 1.5 | 2 | 1 | 0 | −0.08 |
| | 0.6 | 600 | 1.5 | 2 | 1 | 0 | −0.47 |
| | 0.6 | 750 | 1.5 | 2 | 1 | 0 | −0.77 |
| | 0.6 | 100 | 1.5 | 2 | 1 | 0 | −0.05 |
| | 0.6 | 300 | 1.5 | 2 | 1 | 0 | −0.08 |
| | 0.6 | 600 | 1.5 | 2 | 1 | 0 | −0.59 |
| | 0.6 | 750 | 1.5 | 2 | 1 | 0 | −0.81 |
| | 0.6 | 100 | 1.5 | 2 | 1 | 0 | 0.06 |
| | 0.6 | 300 | 1.5 | 2 | 1 | 0 | 0.14 |
| | 0.6 | 600 | 1.5 | 2 | 1 | 0 | −0.56 |

**Table A1.** *Cont.*

| REF. | Input Parameters | | | | | | Output Parameters |
|------|------|------|------|------|------|------|------|
| | W/B | T (°C) | V (°C/min) | MD (h) | C | RD (day) | P (%) |
| [67] | 0.6 | 750 | 1.5 | 2 | 1 | 0 | −0.87 |
| | 0.6 | 100 | 1.5 | 2 | 1 | 0 | 0.05 |
| | 0.6 | 300 | 1.5 | 2 | 1 | 0 | 0.06 |
| | 0.6 | 600 | 1.5 | 2 | 1 | 0 | −0.60 |
| | 0.6 | 750 | 1.5 | 2 | 1 | 0 | −0.93 |
| | 0.6 | 100 | 1.5 | 2 | 1 | 0 | 0.08 |
| | 0.6 | 300 | 1.5 | 2 | 1 | 0 | 0.11 |
| | 0.6 | 600 | 1.5 | 2 | 1 | 0 | −0.58 |
| | 0.6 | 750 | 1.5 | 2 | 1 | 0 | −0.85 |
| | 0.6 | 100 | 1.5 | 2 | 1 | 0 | 0.03 |
| | 0.6 | 300 | 1.5 | 2 | 1 | 0 | 0.11 |
| | 0.6 | 600 | 1.5 | 2 | 1 | 0 | −0.66 |
| | 0.6 | 750 | 1.5 | 2 | 1 | 0 | −0.76 |
| [68] | 0.29 | 200 | 5 | 1 | 1 | 0 | 2.25 |
| | 0.29 | 400 | 5 | 1 | 1 | 0 | −29.21 |
| | 0.29 | 500 | 5 | 1 | 1 | 0 | −43.26 |
| | 0.29 | 600 | 5 | 1 | 1 | 0 | −63.48 |
| | 0.45 | 200 | 5 | 1 | 1 | 0 | 2.68 |
| | 0.45 | 400 | 5 | 1 | 1 | 0 | −16.07 |
| | 0.45 | 500 | 5 | 1 | 1 | 0 | −36.61 |
| | 0.45 | 600 | 5 | 1 | 1 | 0 | −59.82 |
| | 0.32 | 200 | 5 | 1 | 1 | 0 | −14.20 |
| | 0.32 | 400 | 5 | 1 | 1 | 0 | −42.61 |
| | 0.32 | 500 | 5 | 1 | 1 | 0 | −48.30 |
| | 0.32 | 600 | 5 | 1 | 1 | 0 | −68.18 |
| | 0.48 | 200 | 5 | 1 | 1 | 0 | −23.97 |
| | 0.48 | 400 | 5 | 1 | 1 | 0 | −19.83 |
| | 0.48 | 500 | 5 | 1 | 1 | 0 | −41.32 |
| | 0.48 | 600 | 5 | 1 | 1 | 0 | −62.81 |
| [69] | 0.54 | 100 | 5 | 1 | 1 | 0 | −18.74 |
| | 0.54 | 300 | 5 | 1 | 1 | 0 | −40.00 |
| | 0.54 | 500 | 5 | 1 | 1 | 0 | −46.25 |
| | 0.54 | 700 | 5 | 1 | 1 | 0 | −72.50 |
| | 0.6 | 100 | 5 | 1 | 1 | 0 | −10.12 |
| | 0.6 | 300 | 5 | 1 | 1 | 0 | −19.10 |
| | 0.6 | 500 | 5 | 1 | 1 | 0 | −29.22 |
| | 0.6 | 700 | 5 | 1 | 1 | 0 | −49.44 |
| | 0.57 | 100 | 5 | 1 | 1 | 0 | −27.91 |
| | 0.57 | 300 | 5 | 1 | 1 | 0 | −27.91 |
| | 0.57 | 500 | 5 | 1 | 1 | 0 | −52.33 |
| | 0.57 | 700 | 5 | 1 | 1 | 0 | −82.56 |
| | 0.57 | 100 | 5 | 1 | 1 | 0 | −11.77 |
| | 0.57 | 300 | 5 | 1 | 1 | 0 | −16.67 |
| | 0.57 | 500 | 5 | 1 | 1 | 0 | −16.67 |
| | 0.57 | 700 | 5 | 1 | 1 | 0 | −31.38 |
| | 0.46 | 100 | 5 | 1 | 1 | 0 | −28.26 |
| | 0.46 | 300 | 5 | 1 | 1 | 0 | −32.81 |
| | 0.46 | 500 | 5 | 1 | 1 | 0 | −39.34 |
| | 0.46 | 700 | 5 | 1 | 1 | 0 | −45.83 |
| | 0.46 | 100 | 5 | 1 | 1 | 0 | −16.14 |
| | 0.46 | 300 | 5 | 1 | 1 | 0 | −23.61 |
| | 0.46 | 500 | 5 | 1 | 1 | 0 | −30.21 |
| | 0.46 | 700 | 5 | 1 | 1 | 0 | −35.07 |
| | 0.43 | 100 | 5 | 1 | 1 | 0 | −18.63 |
| | 0.43 | 300 | 5 | 1 | 1 | 0 | −30.15 |
| | 0.45 | 100 | 5 | 1 | 1 | 0 | −19.47 |
| | 0.45 | 300 | 5 | 1 | 1 | 0 | −29.08 |

**Table A1.** *Cont.*

| REF. | Input Parameters | | | | | | Output Parameters |
|------|------|------|-------|-------|---|----------|-------|
| | W/B | T (°C) | V (°C/min) | MD (h) | C | RD (day) | P (%) |
| [70] | 0.5 | 200 | 5 | 2 | 1 | 3 | −4.00 |
| | 0.5 | 400 | 5 | 2 | 1 | 3 | −20.00 |
| | 0.5 | 600 | 5 | 2 | 1 | 3 | −36.00 |
| | 0.5 | 800 | 5 | 2 | 1 | 3 | −77.00 |
| | 0.5 | 200 | 5 | 2 | 0 | 3 | −16.00 |
| | 0.5 | 400 | 5 | 2 | 0 | 3 | −37.00 |
| | 0.5 | 600 | 5 | 2 | 0 | 3 | −53.00 |
| | 0.5 | 800 | 5 | 2 | 0 | 3 | −82.00 |
| [71] | 0.55 | 150 | 10 | 2 | 1 | 0 | 8.05 |
| | 0.55 | 500 | 10 | 2 | 1 | 0 | 25.12 |
| | 0.4 | 150 | 10 | 2 | 1 | 0 | 2.30 |
| | 0.4 | 500 | 10 | 2 | 1 | 0 | 46.93 |
| | 0.55 | 150 | 10 | 2 | 1 | 0 | −6.44 |
| | 0.55 | 500 | 10 | 2 | 1 | 0 | −7.92 |
| | 0.55 | 750 | 10 | 2 | 1 | 0 | −57.18 |
| | 0.4 | 150 | 10 | 2 | 1 | 0 | −9.70 |
| | 0.4 | 500 | 10 | 2 | 1 | 0 | −6.84 |
| | 0.4 | 750 | 10 | 2 | 1 | 0 | −65.97 |
| | 0.5 | 150 | 10 | 2 | 1 | 0 | −14.49 |
| | 0.5 | 500 | 10 | 2 | 1 | 0 | −21.74 |
| | 0.5 | 750 | 10 | 2 | 1 | 0 | −77.97 |
| | 0.5 | 1000 | 10 | 2 | 1 | 0 | −91.01 |
| | 0.4 | 150 | 10 | 2 | 1 | 0 | −8.07 |
| | 0.4 | 500 | 10 | 2 | 1 | 0 | −16.14 |
| | 0.4 | 750 | 10 | 2 | 1 | 0 | −67.37 |
| | 0.4 | 1000 | 10 | 2 | 1 | 0 | −89.65 |
| [72] | 0.33 | 200 | 1 | 3 | 1 | 0 | 4.42 |
| | 0.33 | 400 | 1 | 3 | 1 | 0 | −17.33 |
| | 0.33 | 600 | 1 | 3 | 1 | 0 | −49.08 |
| | 0.33 | 800 | 1 | 3 | 1 | 0 | −75.24 |
| | 0.33 | 200 | 1 | 3 | 1 | 0 | −0.92 |
| | 0.33 | 400 | 1 | 3 | 1 | 0 | −16.38 |
| | 0.33 | 600 | 1 | 3 | 1 | 0 | −52.87 |
| | 0.33 | 800 | 1 | 3 | 1 | 0 | −77.30 |
| | 0.33 | 200 | 1 | 3 | 1 | 0 | −1.33 |
| | 0.33 | 400 | 1 | 3 | 1 | 0 | −21.29 |
| | 0.33 | 600 | 1 | 3 | 1 | 0 | −48.69 |
| | 0.33 | 800 | 1 | 3 | 1 | 0 | −79.14 |
| | 0.33 | 200 | 1 | 3 | 1 | 0 | −3.56 |
| | 0.33 | 400 | 1 | 3 | 1 | 0 | −22.85 |
| | 0.33 | 600 | 1 | 3 | 1 | 0 | −51.80 |
| | 0.33 | 800 | 1 | 3 | 1 | 0 | −79.68 |
| | 0.33 | 200 | 1 | 3 | 1 | 0 | 1.22 |
| | 0.33 | 400 | 1 | 3 | 1 | 0 | −18.47 |
| | 0.33 | 600 | 1 | 3 | 1 | 0 | −49.37 |
| | 0.33 | 800 | 1 | 3 | 1 | 0 | −76.34 |
| | 0.33 | 200 | 1 | 3 | 1 | 0 | 0.60 |
| | 0.33 | 400 | 1 | 3 | 1 | 0 | −15.82 |
| | 0.33 | 600 | 1 | 3 | 1 | 0 | −47.50 |
| | 0.33 | 800 | 1 | 3 | 1 | 0 | −75.72 |
| | 0.33 | 200 | 1 | 3 | 1 | 0 | 3.57 |
| | 0.33 | 400 | 1 | 3 | 1 | 0 | −18.66 |
| | 0.33 | 600 | 1 | 3 | 1 | 0 | −47.10 |
| | 0.33 | 800 | 1 | 3 | 1 | 0 | −77.03 |
| | 0.33 | 200 | 1 | 3 | 1 | 0 | −3.77 |
| | 0.33 | 400 | 1 | 3 | 1 | 0 | −20.64 |
| | 0.33 | 600 | 1 | 3 | 1 | 0 | −50.32 |

**Table A1.** *Cont.*

| REF. | Input Parameters | | | | | | Output Parameters |
|---|---|---|---|---|---|---|---|
| | W/B | T (°C) | V (°C/min) | MD (h) | C | RD (day) | P (%) |
| | 0.33 | 800 | 1 | 3 | 1 | 0 | −78.25 |
| | 0.33 | 200 | 1 | 3 | 1 | 0 | −7.45 |
| | 0.33 | 400 | 1 | 3 | 1 | 0 | −21.48 |
| | 0.33 | 600 | 1 | 3 | 1 | 0 | −48.42 |
| | 0.33 | 800 | 1 | 3 | 1 | 0 | −78.80 |
| | 0.33 | 200 | 1 | 3 | 1 | 0 | −8.13 |
| | 0.33 | 400 | 1 | 3 | 1 | 0 | −20.35 |
| | 0.33 | 600 | 1 | 3 | 1 | 0 | −46.98 |
| | 0.33 | 800 | 1 | 3 | 1 | 0 | −79.88 |
| | 0.33 | 200 | 1 | 3 | 1 | 0 | −11.34 |
| | 0.33 | 400 | 1 | 3 | 1 | 0 | −24.47 |
| | 0.33 | 600 | 1 | 3 | 1 | 0 | −57.89 |
| | 0.33 | 800 | 1 | 3 | 1 | 0 | −82.37 |
| | 0.33 | 200 | 1 | 3 | 1 | 0 | −21.10 |
| | 0.33 | 400 | 1 | 3 | 1 | 0 | −30.10 |
| | 0.33 | 600 | 1 | 3 | 1 | 0 | −59.90 |
| | 0.33 | 800 | 1 | 3 | 1 | 0 | −80.70 |
| | 0.33 | 200 | 1 | 3 | 1 | 0 | −23.19 |
| | 0.33 | 400 | 1 | 3 | 1 | 0 | −33.41 |
| | 0.33 | 600 | 1 | 3 | 1 | 0 | −57.96 |
| | 0.33 | 800 | 1 | 3 | 1 | 0 | −83.53 |
| | 0.33 | 200 | 1 | 3 | 1 | 0 | −20.04 |
| | 0.33 | 400 | 1 | 3 | 1 | 0 | −35.07 |
| | 0.33 | 600 | 1 | 3 | 1 | 0 | −62.28 |
| [72] | 0.33 | 800 | 1 | 3 | 1 | 0 | −85.19 |
| | 0.33 | 200 | 1 | 3 | 1 | 0 | −16.13 |
| | 0.33 | 400 | 1 | 3 | 1 | 0 | −23.44 |
| | 0.33 | 600 | 1 | 3 | 1 | 0 | −56.93 |
| | 0.33 | 800 | 1 | 3 | 1 | 0 | −84.02 |
| | 0.33 | 200 | 1 | 3 | 1 | 0 | −21.02 |
| | 0.33 | 400 | 1 | 3 | 1 | 0 | −26.56 |
| | 0.33 | 600 | 1 | 3 | 1 | 0 | −57.50 |
| | 0.33 | 800 | 1 | 3 | 1 | 0 | −81.73 |
| | 0.33 | 200 | 1 | 3 | 1 | 0 | −21.84 |
| | 0.33 | 400 | 1 | 3 | 1 | 0 | −26.69 |
| | 0.33 | 600 | 1 | 3 | 1 | 0 | −59.45 |
| | 0.33 | 800 | 1 | 3 | 1 | 0 | −86.14 |
| | 0.33 | 200 | 1 | 3 | 1 | 0 | −17.50 |
| | 0.33 | 400 | 1 | 3 | 1 | 0 | −24.21 |
| | 0.33 | 600 | 1 | 3 | 1 | 0 | −59.22 |
| | 0.33 | 800 | 1 | 3 | 1 | 0 | −84.31 |
| | 0.33 | 200 | 1 | 3 | 1 | 0 | −19.78 |
| | 0.33 | 400 | 1 | 3 | 1 | 0 | −35.83 |
| | 0.33 | 600 | 1 | 3 | 1 | 0 | −62.77 |
| | 0.33 | 800 | 1 | 3 | 1 | 0 | −85.42 |
| | 0.33 | 200 | 1 | 3 | 1 | 0 | −24.77 |
| | 0.33 | 400 | 1 | 3 | 1 | 0 | −42.63 |
| | 0.33 | 600 | 1 | 3 | 1 | 0 | −62.38 |
| | 0.33 | 800 | 1 | 3 | 1 | 0 | −84.96 |
| | 0.61 | 150 | 1 | 1 | 1 | 0 | −0.13 |
| | 0.61 | 300 | 1 | 1 | 1 | 0 | −0.17 |
| | 0.61 | 450 | 1 | 1 | 1 | 0 | −0.51 |
| [73] | 0.61 | 600 | 1 | 1 | 1 | 0 | −0.86 |
| | 0.57 | 150 | 1 | 1 | 1 | 0 | −0.09 |
| | 0.57 | 300 | 1 | 1 | 1 | 0 | 0.17 |
| | 0.57 | 450 | 1 | 1 | 1 | 0 | −0.44 |
| | 0.57 | 600 | 1 | 1 | 1 | 0 | −0.80 |

Table A1. *Cont.*

| REF. | Input Parameters | | | | | | Output Parameters |
|---|---|---|---|---|---|---|---|
| | W/B | T (°C) | V (°C/min) | MD (h) | C | RD (day) | P (%) |
| [73] | 0.54 | 150 | 1 | 1 | 1 | 0 | −0.21 |
| | 0.54 | 300 | 1 | 1 | 1 | 0 | −0.04 |
| | 0.54 | 450 | 1 | 1 | 1 | 0 | −0.55 |
| | 0.54 | 600 | 1 | 1 | 1 | 0 | −0.83 |
| [74] | 0.35 | 300 | 5 | 4 | 1 | 0 | −19.55 |
| | 0.35 | 500 | 5 | 4 | 1 | 0 | −53.22 |
| | 0.35 | 600 | 5 | 4 | 1 | 0 | −65.49 |
| | 0.35 | 800 | 5 | 4 | 1 | 0 | −84.41 |
| | 0.35 | 300 | 5 | 4 | 1 | 0 | −10.25 |
| | 0.35 | 500 | 5 | 4 | 1 | 0 | −49.50 |
| | 0.35 | 600 | 5 | 4 | 1 | 0 | −65.00 |
| | 0.35 | 800 | 5 | 4 | 1 | 0 | −83.50 |
| | 0.35 | 300 | 5 | 4 | 1 | 0 | −15.75 |
| | 0.35 | 500 | 5 | 4 | 1 | 0 | −51.38 |
| | 0.35 | 600 | 5 | 4 | 1 | 0 | −66.58 |
| | 0.35 | 800 | 5 | 4 | 1 | 0 | −82.60 |
| | 0.35 | 300 | 5 | 4 | 1 | 0 | −17.52 |
| | 0.35 | 500 | 5 | 4 | 1 | 0 | −56.32 |
| | 0.35 | 600 | 5 | 4 | 1 | 0 | −66.66 |
| | 0.35 | 800 | 5 | 4 | 1 | 0 | −80.46 |
| | 0.35 | 300 | 5 | 4 | 1 | 0 | −18.60 |
| | 0.35 | 500 | 5 | 4 | 1 | 0 | −54.81 |
| | 0.35 | 600 | 5 | 4 | 1 | 0 | −66.67 |
| | 0.35 | 800 | 5 | 4 | 1 | 0 | −78.53 |
| | 0.35 | 300 | 5 | 4 | 1 | 0 | −7.84 |
| | 0.35 | 500 | 5 | 4 | 1 | 0 | −50.39 |
| | 0.35 | 600 | 5 | 4 | 1 | 0 | −61.88 |
| | 0.35 | 800 | 5 | 4 | 1 | 0 | −80.16 |
| | 0.35 | 300 | 5 | 4 | 1 | 0 | −6.87 |
| | 0.35 | 500 | 5 | 4 | 1 | 0 | −47.39 |
| | 0.35 | 600 | 5 | 4 | 1 | 0 | −59.15 |
| | 0.35 | 800 | 5 | 4 | 1 | 0 | −77.78 |
| | 0.35 | 300 | 5 | 4 | 1 | 0 | −5.08 |
| | 0.35 | 500 | 5 | 4 | 1 | 0 | −46.78 |
| | 0.35 | 600 | 5 | 4 | 1 | 0 | −61.02 |
| | 0.35 | 800 | 5 | 4 | 1 | 0 | −77.63 |
| | 0.35 | 300 | 5 | 4 | 1 | 0 | −7.92 |
| | 0.35 | 500 | 5 | 4 | 1 | 0 | −49.81 |
| | 0.35 | 600 | 5 | 4 | 1 | 0 | −56.60 |
| | 0.35 | 800 | 5 | 4 | 1 | 0 | −74.72 |
| | 0.35 | 300 | 5 | 4 | 1 | 0 | −15.29 |
| | 0.35 | 500 | 5 | 4 | 1 | 0 | −54.12 |
| | 0.35 | 600 | 5 | 4 | 1 | 0 | −65.88 |
| | 0.35 | 800 | 5 | 4 | 1 | 0 | −72.94 |
| [75] | 0.3 | 100 | 2 | 1 | 1 | 0 | −0.26 |
| | 0.3 | 200 | 2 | 1 | 1 | 0 | −8.74 |
| | 0.3 | 400 | 2 | 1 | 1 | 0 | −12.58 |
| | 0.3 | 600 | 2 | 1 | 1 | 0 | −53.11 |
| | 0.3 | 100 | 2 | 1 | 1 | 0 | −2.12 |
| | 0.3 | 200 | 2 | 1 | 1 | 0 | −2.66 |
| | 0.3 | 400 | 2 | 1 | 1 | 0 | −19.79 |
| | 0.3 | 600 | 2 | 1 | 1 | 0 | −47.94 |
| | 0.3 | 100 | 2 | 1 | 1 | 0 | −0.40 |
| | 0.3 | 200 | 2 | 1 | 1 | 0 | −3.04 |
| | 0.3 | 400 | 2 | 1 | 1 | 0 | −9.51 |
| | 0.3 | 600 | 2 | 1 | 1 | 0 | −54.82 |
| | 0.5 | 100 | 2 | 1 | 1 | 0 | 4.40 |

**Table A1.** *Cont.*

| REF. | Input Parameters | | | | | | Output Parameters |
|---|---|---|---|---|---|---|---|
| | W/B | T (°C) | V (°C/min) | MD (h) | C | RD (day) | P (%) |
| [75] | 0.5 | 200 | 2 | 1 | 1 | 0 | −7.33 |
| | 0.5 | 400 | 2 | 1 | 1 | 0 | −21.99 |
| | 0.5 | 600 | 2 | 1 | 1 | 0 | −50.73 |
| [76] | 0.4 | 500 | 4.17 | 1 | 1 | 0 | −21.77 |
| | 0.4 | 500 | 5 | 4 | 1 | 0 | −26.30 |
| | 0.4 | 500 | 4.17 | 1 | 1 | 0 | −9.98 |
| | 0.4 | 500 | 5 | 4 | 1 | 0 | −26.52 |
| | 0.55 | 500 | 4.17 | 1 | 1 | 0 | −10.14 |
| | 0.55 | 500 | 5 | 4 | 1 | 0 | −26.69 |
| | 0.55 | 500 | 4.17 | 1 | 1 | 0 | −15.97 |
| | 0.55 | 500 | 5 | 4 | 1 | 0 | −32.59 |
| | 0.7 | 500 | 4.17 | 1 | 1 | 0 | −15.00 |
| | 0.7 | 500 | 5 | 4 | 1 | 0 | −21.82 |
| | 0.7 | 500 | 4.17 | 1 | 1 | 0 | −17.54 |
| | 0.7 | 500 | 5 | 4 | 1 | 0 | −30.33 |
| [77] | 0.28 | 200 | 10 | 3 | 1 | 0 | 1.23 |
| | 0.28 | 400 | 10 | 3 | 1 | 0 | −54.30 |
| | 0.28 | 600 | 10 | 3 | 1 | 0 | −75.61 |
| | 0.28 | 800 | 10 | 3 | 1 | 0 | −88.93 |
| | 0.28 | 200 | 10 | 3 | 1 | 0 | 0.36 |
| | 0.28 | 400 | 10 | 3 | 1 | 0 | −14.92 |
| | 0.28 | 600 | 10 | 3 | 1 | 0 | −41.72 |
| | 0.28 | 800 | 10 | 3 | 1 | 0 | −67.80 |
| | 0.28 | 200 | 10 | 3 | 1 | 0 | −1.91 |
| | 0.28 | 400 | 10 | 3 | 1 | 0 | −13.69 |
| | 0.28 | 600 | 10 | 3 | 1 | 0 | −40.50 |
| | 0.28 | 800 | 10 | 3 | 1 | 0 | −64.26 |
| | 0.28 | 200 | 10 | 3 | 1 | 0 | −2.10 |
| | 0.28 | 400 | 10 | 3 | 1 | 0 | −48.01 |
| | 0.28 | 600 | 10 | 3 | 1 | 0 | −52.20 |
| | 0.28 | 800 | 10 | 3 | 1 | 0 | −70.65 |
| | 0.28 | 200 | 10 | 3 | 1 | 0 | −0.62 |
| | 0.28 | 400 | 10 | 3 | 1 | 0 | −16.54 |
| | 0.28 | 600 | 10 | 3 | 1 | 0 | −48.53 |
| | 0.28 | 800 | 10 | 3 | 1 | 0 | −72.49 |
| | 0.28 | 200 | 10 | 3 | 1 | 0 | −0.96 |
| | 0.28 | 400 | 10 | 3 | 1 | 0 | −15.66 |
| | 0.28 | 600 | 10 | 3 | 1 | 0 | −43.45 |
| | 0.28 | 800 | 10 | 3 | 1 | 0 | −65.02 |
| | 0.28 | 200 | 10 | 3 | 1 | 0 | −1.75 |
| | 0.28 | 400 | 10 | 3 | 1 | 0 | −14.79 |
| | 0.28 | 600 | 10 | 3 | 1 | 0 | −43.00 |
| | 0.28 | 800 | 10 | 3 | 1 | 0 | −59.73 |
| [78] | 0.6 | 400 | 5.2 | 1 | 1 | 0 | −25.37 |
| | 0.6 | 600 | 5.2 | 1 | 1 | 0 | −61.30 |
| | 0.6 | 800 | 4.14 | 1 | 1 | 0 | −82.60 |
| | 0.6 | 1000 | 3.5 | 1 | 1 | 0 | −90.08 |
| | 0.6 | 1200 | 2.96 | 1 | 1 | 0 | −83.74 |
| | 0.35 | 400 | 5.2 | 1 | 1 | 0 | 0.38 |
| | 0.35 | 600 | 5.2 | 1 | 1 | 0 | −45.05 |
| | 0.35 | 800 | 4.14 | 1 | 1 | 0 | −71.31 |
| | 0.35 | 1000 | 3.5 | 1 | 1 | 0 | −88.22 |
| | 0.35 | 1200 | 2.96 | 1 | 1 | 0 | −86.92 |
| | 0.28 | 400 | 5.2 | 1 | 1 | 0 | −0.39 |
| | 0.28 | 600 | 5.2 | 1 | 1 | 0 | −47.99 |
| | 0.28 | 800 | 4.14 | 1 | 1 | 0 | −74.46 |

**Table A1.** *Cont.*

| REF. | Input Parameters | | | | | | Output Parameters |
|---|---|---|---|---|---|---|---|
| | W/B | T (°C) | V (°C/min) | MD (h) | C | RD (day) | P (%) |
| [78] | 0.28 | 1000 | 3.5 | 1 | 1 | 0 | −88.05 |
| | 0.28 | 1200 | 2.96 | 1 | 1 | 0 | −88.12 |
| [79] | 0.4 | 100 | 3 | 3 | 1 | 0 | −14.01 |
| | 0.4 | 200 | 3 | 3 | 1 | 0 | −10.81 |
| | 0.4 | 300 | 3 | 3 | 1 | 0 | −25.21 |
| | 0.4 | 600 | 3 | 3 | 1 | 0 | −67.20 |
| | 0.35 | 100 | 3 | 3 | 1 | 0 | −14.29 |
| | 0.35 | 200 | 3 | 3 | 1 | 0 | −12.96 |
| | 0.35 | 300 | 3 | 3 | 1 | 0 | −22.93 |
| | 0.35 | 600 | 3 | 3 | 1 | 0 | −70.10 |
| | 0.3 | 100 | 3 | 3 | 1 | 0 | −16.28 |
| | 0.3 | 200 | 3 | 3 | 1 | 0 | −15.41 |
| | 0.3 | 300 | 3 | 3 | 1 | 0 | −31.10 |
| | 0.3 | 600 | 3 | 3 | 1 | 0 | −73.55 |
| | 0.3 | 100 | 3 | 3 | 1 | 0 | −14.49 |
| | 0.3 | 200 | 3 | 3 | 1 | 0 | −11.23 |
| | 0.3 | 300 | 3 | 3 | 1 | 0 | −27.54 |
| | 0.3 | 600 | 3 | 3 | 1 | 0 | −68.48 |
| [80] | 0.34 | 200 | 25 | 3 | 1 | 15 | −12.97 |
| | 0.34 | 400 | 25 | 2.5 | 1 | 15 | −38.59 |
| | 0.34 | 600 | 25 | 2 | 1 | 15 | −49.69 |
| | 0.34 | 800 | 25 | 2 | 1 | 15 | −84.06 |
| | 0.34 | 200 | 25 | 3 | 1 | 15 | −31.76 |
| | 0.34 | 400 | 25 | 2.5 | 1 | 15 | −56.44 |
| | 0.34 | 600 | 25 | 2 | 1 | 15 | −69.51 |
| | 0.34 | 800 | 25 | 2 | 1 | 15 | −90.38 |
| [81] | 0.5 | 100 | 5 | 1 | 1 | 0 | 0.29 |
| | 0.5 | 200 | 5 | 1 | 1 | 0 | −0.76 |
| | 0.5 | 300 | 5 | 1 | 1 | 0 | −10.18 |
| | 0.5 | 400 | 5 | 1 | 1 | 0 | −28.75 |
| | 0.5 | 500 | 5 | 1 | 1 | 0 | −51.10 |
| | 0.5 | 600 | 5 | 1 | 1 | 0 | −75.61 |
| | 0.5 | 700 | 5 | 1 | 1 | 0 | −81.51 |
| | 0.5 | 800 | 5 | 1 | 1 | 0 | −91.19 |
| | 0.5 | 1200 | 5 | 1 | 1 | 0 | −99.46 |
| | 0.5 | 100 | 5 | 1 | 1 | 0 | 3.80 |
| | 0.5 | 200 | 5 | 1 | 1 | 0 | 2.74 |
| | 0.5 | 300 | 5 | 1 | 1 | 0 | −6.67 |
| | 0.5 | 400 | 5 | 1 | 1 | 0 | −26.06 |
| | 0.5 | 500 | 5 | 1 | 1 | 0 | −43.28 |
| | 0.5 | 600 | 5 | 1 | 1 | 0 | −53.50 |
| | 0.5 | 700 | 5 | 1 | 1 | 0 | −68.85 |
| | 0.5 | 800 | 5 | 1 | 1 | 0 | −92.27 |
| [82] | 0.3 | 200 | 5 | 3 | 1 | 0 | −15.00 |
| | 0.3 | 400 | 5 | 3 | 1 | 0 | −20.00 |
| | 0.3 | 600 | 5 | 3 | 1 | 0 | −42.00 |
| [83] | 0.43 | 100 | 2 | 2 | 1 | 0 | −16.90 |
| | 0.43 | 150 | 2 | 2 | 1 | 0 | −11.34 |
| | 0.43 | 200 | 2 | 2 | 1 | 0 | −8.21 |
| | 0.43 | 250 | 2 | 2 | 1 | 0 | −0.12 |
| | 0.43 | 280 | 2 | 2 | 1 | 0 | 11.23 |
| | 0.43 | 100 | 2 | 2 | 1 | 0 | −8.77 |
| | 0.43 | 150 | 2 | 2 | 1 | 0 | −14.85 |
| | 0.43 | 200 | 2 | 2 | 1 | 0 | −17.98 |

**Table A1.** *Cont.*

| REF. | Input Parameters | | | | | | Output Parameters |
|---|---|---|---|---|---|---|---|
| | W/B | T (°C) | V (°C/min) | MD (h) | C | RD (day) | P (%) |
| [83] | 0.43 | 250 | 2 | 2 | 1 | 0 | −17.98 |
| | 0.43 | 280 | 2 | 2 | 1 | 0 | −11.89 |
| [84] | 0.5 | 100 | 2 | 0.75 | 1 | 0 | −8.59 |
| | 0.5 | 200 | 2 | 0.75 | 1 | 0 | −16.90 |
| | 0.5 | 400 | 2 | 0.75 | 1 | 0 | −34.06 |
| | 0.5 | 600 | 2 | 0.75 | 1 | 0 | −45.81 |
| | 0.5 | 800 | 2 | 0.75 | 1 | 0 | −71.56 |
| | 0.5 | 100 | 2 | 0.75 | 1 | 0 | −9.00 |
| | 0.5 | 200 | 2 | 0.75 | 1 | 0 | −17.64 |
| | 0.5 | 400 | 2 | 0.75 | 1 | 0 | −30.35 |
| | 0.5 | 600 | 2 | 0.75 | 1 | 0 | −50.17 |
| | 0.5 | 800 | 2 | 0.75 | 1 | 0 | −68.89 |
| [85] | 0.4 | 200 | 3 | 3 | 1 | 0 | −2.41 |
| | 0.4 | 400 | 3 | 2.5 | 1 | 0 | −13.74 |
| | 0.4 | 600 | 3 | 2 | 1 | 0 | −46.12 |
| | 0.4 | 800 | 3 | 2 | 1 | 0 | −80.24 |
| | 0.4 | 200 | 3 | 3 | 1 | 0 | −4.15 |
| | 0.4 | 400 | 3 | 2.5 | 1 | 0 | −9.78 |
| | 0.4 | 600 | 3 | 2 | 1 | 0 | −43.40 |
| | 0.4 | 800 | 3 | 2 | 1 | 0 | −74.79 |
| | 0.4 | 200 | 3 | 3 | 1 | 0 | −0.93 |
| | 0.4 | 400 | 3 | 2.5 | 1 | 0 | −6.81 |
| | 0.4 | 600 | 3 | 2 | 1 | 0 | −41.67 |
| | 0.4 | 800 | 3 | 2 | 1 | 0 | −80.74 |
| [86] | 0.3 | 200 | 3.3 | 2 | 1 | 0 | −7.28 |
| | 0.3 | 400 | 3.3 | 2 | 1 | 0 | −10.91 |
| | 0.3 | 600 | 3.3 | 2 | 1 | 0 | −28.32 |
| | 0.3 | 800 | 3.3 | 2 | 1 | 0 | −61.19 |
| | 0.3 | 200 | 3.3 | 2 | 1 | 0 | −3.52 |
| | 0.3 | 400 | 3.3 | 2 | 1 | 0 | −2.46 |
| | 0.3 | 600 | 3.3 | 2 | 1 | 0 | −23.39 |
| | 0.3 | 800 | 3.3 | 2 | 1 | 0 | −56.37 |
| [87] | 0.72 | 300 | 10 | 3 | 1 | 0 | 22.59 |
| | 0.72 | 600 | 10 | 3 | 1 | 0 | −4.17 |
| | 0.72 | 900 | 10 | 3 | 1 | 0 | −68.42 |
| | 0.71 | 300 | 10 | 3 | 1 | 0 | 34.31 |
| | 0.71 | 600 | 10 | 3 | 1 | 0 | 10.71 |
| | 0.71 | 900 | 10 | 3 | 1 | 0 | −72.26 |
| | 0.7 | 300 | 10 | 3 | 1 | 0 | 41.52 |
| | 0.7 | 600 | 10 | 3 | 1 | 0 | 16.71 |
| | 0.7 | 900 | 10 | 3 | 1 | 0 | −55.77 |
| | 0.7 | 300 | 10 | 3 | 1 | 0 | 43.13 |
| | 0.7 | 600 | 10 | 3 | 1 | 0 | 36.56 |
| | 0.7 | 900 | 10 | 3 | 1 | 0 | −48.44 |
| | 0.7 | 300 | 10 | 3 | 1 | 0 | 37.07 |
| | 0.7 | 600 | 10 | 3 | 1 | 0 | 24.88 |
| | 0.7 | 900 | 10 | 3 | 1 | 0 | −23.41 |
| | 0.72 | 300 | 20 | 3 | 1 | 0 | −18.86 |
| | 0.72 | 600 | 20 | 3 | 1 | 0 | −30.92 |
| | 0.72 | 900 | 20 | 3 | 1 | 0 | −71.93 |
| | 0.71 | 300 | 20 | 3 | 1 | 0 | −14.84 |
| | 0.71 | 600 | 20 | 3 | 1 | 0 | −30.90 |
| | 0.71 | 900 | 20 | 3 | 1 | 0 | −74.70 |
| | 0.7 | 300 | 20 | 3 | 1 | 0 | −13.02 |
| | 0.7 | 600 | 20 | 3 | 1 | 0 | −23.34 |
| | 0.7 | 900 | 20 | 3 | 1 | 0 | −60.20 |

**Table A1.** *Cont.*

| REF. | Input Parameters | | | | | | Output Parameters |
|------|-----|--------|-----------|--------|---|----------|-------|
| | W/B | T (°C) | V (°C/min) | MD (h) | C | RD (day) | P (%) |
| [87] | 0.7 | 300 | 20 | 3 | 1 | 0 | −14.69 |
| | 0.7 | 600 | 20 | 3 | 1 | 0 | −27.50 |
| | 0.7 | 900 | 20 | 3 | 1 | 0 | −53.44 |
| | 0.7 | 300 | 20 | 3 | 1 | 0 | −28.29 |
| | 0.7 | 600 | 20 | 3 | 1 | 0 | −36.10 |
| | 0.7 | 900 | 20 | 3 | 1 | 0 | −28.29 |
| [88] | 0.47 | 200 | 10 | 0 | 1 | 0 | −0.19 |
| | 0.47 | 400 | 10 | 0 | 1 | 0 | −15.39 |
| | 0.47 | 600 | 10 | 0 | 1 | 0 | −12.68 |
| | 0.47 | 800 | 10 | 0 | 1 | 0 | −55.18 |
| | 0.47 | 200 | 10 | 0 | 1 | 0 | −5.30 |
| | 0.47 | 400 | 10 | 0 | 1 | 0 | −25.88 |
| | 0.47 | 600 | 10 | 0 | 1 | 0 | −32.48 |
| | 0.47 | 800 | 10 | 0 | 1 | 0 | −47.68 |
| | 0.47 | 200 | 10 | 0 | 1 | 0 | 2.02 |
| | 0.47 | 400 | 10 | 0 | 1 | 0 | −14.72 |
| | 0.47 | 600 | 10 | 0 | 1 | 0 | −6.00 |
| | 0.47 | 800 | 10 | 0 | 1 | 0 | −30.94 |
| | 0.47 | 200 | 10 | 0 | 1 | 0 | −15.88 |
| | 0.47 | 400 | 10 | 0 | 1 | 0 | 4.28 |
| | 0.47 | 600 | 10 | 0 | 1 | 0 | −14.91 |
| | 0.47 | 800 | 10 | 0 | 1 | 0 | −34.43 |
| | 0.47 | 200 | 10 | 1 | 1 | 0 | −21.77 |
| | 0.47 | 400 | 10 | 1 | 1 | 0 | −3.38 |
| | 0.47 | 600 | 10 | 1 | 1 | 0 | −16.81 |
| | 0.47 | 800 | 10 | 1 | 1 | 0 | −57.13 |
| | 0.47 | 200 | 10 | 1 | 1 | 0 | 3.42 |
| | 0.47 | 400 | 10 | 1 | 1 | 0 | −19.59 |
| | 0.47 | 600 | 10 | 1 | 1 | 0 | −39.67 |
| | 0.47 | 800 | 10 | 1 | 1 | 0 | −51.77 |
| | 0.47 | 200 | 10 | 1 | 1 | 0 | −10.02 |
| | 0.47 | 400 | 10 | 1 | 1 | 0 | −3.74 |
| | 0.47 | 600 | 10 | 1 | 1 | 0 | −11.44 |
| | 0.47 | 800 | 10 | 1 | 1 | 0 | −51.09 |
| | 0.47 | 200 | 10 | 1 | 1 | 0 | −30.66 |
| | 0.47 | 400 | 10 | 1 | 1 | 0 | 2.13 |
| | 0.47 | 600 | 10 | 1 | 1 | 0 | −22.09 |
| | 0.47 | 800 | 10 | 1 | 1 | 0 | −43.47 |
| | 0.47 | 200 | 10 | 2 | 1 | 0 | −11.96 |
| | 0.47 | 400 | 10 | 2 | 1 | 0 | −5.09 |
| | 0.47 | 600 | 10 | 2 | 1 | 0 | −30.14 |
| | 0.47 | 800 | 10 | 2 | 1 | 0 | −60.54 |
| | 0.47 | 200 | 10 | 2 | 1 | 0 | −9.17 |
| | 0.47 | 400 | 10 | 2 | 1 | 0 | −16.20 |
| | 0.47 | 600 | 10 | 2 | 1 | 0 | −39.69 |
| | 0.47 | 800 | 10 | 2 | 1 | 0 | −67.80 |
| | 0.47 | 200 | 10 | 2 | 1 | 0 | −10.28 |
| | 0.47 | 400 | 10 | 2 | 1 | 0 | 7.51 |
| | 0.47 | 600 | 10 | 2 | 1 | 0 | −26.43 |
| | 0.47 | 800 | 10 | 2 | 1 | 0 | −53.69 |
| | 0.47 | 200 | 10 | 2 | 1 | 0 | −9.97 |
| | 0.47 | 400 | 10 | 2 | 1 | 0 | −12.12 |
| | 0.47 | 600 | 10 | 2 | 1 | 0 | −30.34 |
| | 0.47 | 800 | 10 | 2 | 1 | 0 | −49.06 |

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
