# Peer review of "Prediction of Compressive Strength Loss of Normal Concrete after Exposure to High Temperature"

_applsci, doi:10.3390/app122312237_

Round 1

Reviewer 1 Report

The corrections are noted in the attached file.

Reviewer 2 Report

The paper proposes statistical models to predict compressive strength loss of normal concrete after exposure to high temperatures.

The topic of the paper is interesting and deserves to be published after major revisions. Nevertheless, the paper is defective in many of its parts. Some suggestions are provided below to improve its quality. Moreover, it is not clear how the proposed methodology could be applied in the real practice, and what advantages could have. Authors are suggest to add a part in which they provide strategies to be adopted by technicians in real applications, according to the AI methods discussed in their paper and providing more detailed and practical suggestions.

1.       In the abstract, Authors are suggested to give less detailed outcomes (that are well described in the conclusions) and more details about their work, what they have done, and why they decided to use AI algorithms to perform their work.

2.       The Introduction must be improved. The context in which the problem is treated is just mentioned. Authors wrote: “In recent years, there has been an increasing number of fires in buildings”. Is it true? In which part of the world is this a main concern? Authors need to further discuss this concern also adding some references discussing this problem. Moreover, please add more information about the work is developed, just to better describe the framework of the paper.

3.       The Authors use the term “room temperature” throughout the paper. Please, define what it means at the first time it appears in the text.

4.       In Section 2.2, please add more information about the considered data set. Indeed, it is better for the reader to be aware about the types of data used in the paper, from which kind of structure they were achieved. The long list of references is not enough and could disappoint the reader.

5.       Table 2 contains acronyms that are defined after in the paper. Please, correct this inconsistency.

6.       The content of Figure 4, 5 and 6 must be better explain, given the reader a key to read the content.

7.       In Section 4, it is not clear the difference between the contents of Sections 4.1 and 4.2. Is the first Section inherent to the literature data set? Please, add more information.

8.       The Conclusions are very short and they not explain the potentialities and advantages that this work could be add to the scientific community.

In addition, the reviewer has other additional comments about minor typos:

9.       Equation 1 is quoted twice.

1.   Some references are not quoted in the text (2-3-89-90-92)

Reviewer 3 Report

The article titled Prediction of Compressive Strength Loss of Normal Concrete after Exposure to High Temperature by Xiaoyu Qin, Qianmin Ma, Rongxin Guo, Shaoen Tan is a correct research paper. The introduction is very general and too laconic. Information on the behavior of concrete at high temperatures was missing. The temperature-time curves used by various researchers and included in the standards were not shown. This information was also missing from the description of experimental studies. In the introduction, the mathematical models used for the study were signaled. Based on these models, a very large amount of data obtained from numerous articles (mostly in Chinese) was analyzed. The amount of data analyzed (shown in the appendix) is impressive. As a result of the analyses, it turned out that all the mathematical models used are usable with very similar accuracy. The experimental studies conducted are interesting. It is a pity that, again, the information about them is very short. At least a few photos of the tests could be shown here, the temperature - time relationship could be described, a picture of the destruction of the samples without and after exposure to high temperature could be shown. A good summary comes from Table 6, where the P-parameter is compared for different mathematic models and experiment. It is interesting to try to rank the importance of input parameters on the loss of compressive strength of plain concrete after exposure to high temperature. The conclusions are very poor and essentially repeat the information shown earlier. Several comments and questions arose during the analysis of the article:

1) The water/binder (W/B) ratio, expressed as a percentage, appears among the input parameters. Why was it not called the water/cement ratio (W/C)? Was a binder other than cement used?

2) Why such large standard deviations for temperature T and heating rate V obtained for the literature data?

3) What does the information on resting time RD result 0 in Table 4 mean? Was the natural cooling method C sufficient to test the compressive strength of concrete after a rest time RD of 0?

4) There is a lack of explanation of the axes in the graphs and the presented results shown in Figures 4, 5 and 6. The description presented to this point is very general. How do we know that the correlation coefficient R > 0.83?

5) It is not fully explained why, in the study, blend 1 is so different from the others (Figure 7), regardless of the mathematical model used.

6) The appendix includes the result of the calculation of the output parameter P. Is this result from the cited works, or is it the result of the analyses carried out by the authors of this article?

7) Is it possible to show on graphs the relationship, for example, temperature T - parameter P (other relationships may also arise)? Is it possible to relate this information (T-P) to, for example, standard results describing the decrease in compressive strength of concrete as a function of increasing temperature?

Round 2

Reviewer 2 Report

Please, revise some typos still present in the manuscript.

Author Response

The corrections have been marked in yellow, please see the attachment.
